



# Pelagic primary production in the coastal Mediterranean Sea: variability, trends and contribution to basin scale budgets

Paula Maria Salgado-Hernanz[1,2], Aurore Regaudie de Gioux[3], David Antoine[4,5], Gotzon Basterretxea[1]

[1]Department of Marine Ecology, IMEDEA (UIB-CSIC), Miquel Marquès 21, 07190 Esporles, Spain
[2]Centro Oceanográfico de Baleares, Instituto Español Oceanografia (COB - IEO), Muelle de Poniente s/n, 07015 Palma de Mallorca, Spain
[3]ODE/DYNECO/Pelagos, Centre de Bretagne, IFREMER, I. Technopôle Brest-Iroise, Pointe du Diable BP70 29280 Plouzané, France
[4]Remote Sensing and Satellite Research Group, School of Earth and Planetary Sciences, Curtin University, Perth, WA 6845,
Australia
[5]Sorbonne Université, CNRS, Laboratoire d'Océanographie de Villefranche, LOV, F-06230 Villefranche-sur-Mer, France

*Correspondence to*: : Paula M. Salgado-Hernanz (pmsalgadohernanz@gmail.com)

**Abstract.** We estimated pelagic primary production (PP) in the coastal (<200 m depth) Mediterranean Sea from satellite-borne data, its contribution to basin-scale carbon fixation, its variability and long-term trends during the period 2002-2016. Annual
coastal PP was estimated at 0.041 Gt C, which approximately represents 12 % of total carbon fixation in the Mediterranean Sea. About 50 % of this production occurs in the eastern basin, whereas the western and Adriatic shelves contribute with 25% each of total coastal production. Strong regional variability is revealed, from high-production areas (>300 g C m$^{-2}$) associated with major river discharges, to less productive provinces (<50 g C m$^{-2}$) located in the southeastern Mediterranean. PP variability in the Mediterranean Sea is dominated by interannual variations but overall trend during the study period shows
notable decrease (17%) since 2012 concurring with a period of increasing sea surface temperatures in the Mediterranean Sea and positive North Atlantic Oscillation and the Mediterranean Oscillation climate indices. PP declines in most coastal areas (-0.05 to -0.1 g C m$^{-2}$ per decade) except in the Adriatic where PP increases at +0.1 g C m$^{-2}$ per decade. Regionalization of coastal waters based on PP seasonal patterns reveals the importance of river effluents in determining PP in coastal waters that can regionally increase in up to five-fold. Our study provides insight on the contribution of coastal waters to basin scale carbon
balances in the Mediterranean Sea while highlighting the importance of the different temporal and spatial scales of variability.

## 1 Introduction

Coastal ocean waters (i.e., < 200 m depth) are an important link between the land and the open ocean. They act as a buffer between terrestrial and human influences and the open ocean (Liu et al., 2000). Despite their relatively reduced extension (~7% of ocean surface area; Gattuso et al., 1998), they behold some of the most productive habitats on the planet. Therefore, they
have a disproportionate importance in many basin-scale biogeochemical and ecological processes, including carbon and nitrogen cycling, and in the maintenance of marine diversity (Cebrian, 2002; Coll et al., 2010; Dunne et al., 2007). Besides,





biological production of continental shelves supports over 90 % of global fish catches (Pauly et al., 2002; Pauly and Christensen, 1995).

Coastal seawaters support high primary production (PP), contributing to some 10 % of global ocean PP and up to 30
% if estuarine and benthic production is considered (Ducklow et al., 2001; Muller-Karger et al., 2005). These high rates of organic productivity occur in the coastal oceans due to the rapid turnover of the large inputs of nutrients and organic carbon from land. PP drives a significant carbon sink in the ocean (Field et al., 1998; Laws et al., 2000), and is a key regulator of ecological processes such as elemental cycling, trophic structure variabilities and climate change (Bauer et al., 2013; Chavez et al., 2011). In coastal waters, physical and biological processes enhance the carbon transport out of the continental margins
into the deep layers of the oceans, thus connecting terrestrial with deep oceanic systems (Cai, 2011; Carlson et al., 2001; Cole et al., 2007) The productivity of coastal sea areas is also of strategic socio-economic importance for many countries considering that PP constrains the amount of fish and invertebrates available to expanding fisheries, a primary resource for many coastal human communities (Chassot et al., 2010). The estimation and understanding of PP evolution and trends in the coastal seas is therefore essential to improve our knowledge of the oceanic carbon cycle.

Scaling up local measurements to estimate the contribution of coastal regions to global carbon fluxes has been hindered by the high spatial and temporal heterogeneity of these waters. Global models of oceanic systems produce carbon fixation estimates with a high degree of uncertainty in coastal regions (Muller-Karger et al., 2005). Coastal waters are complex because of the tight connection between terrestrial and oceanic systems. Terrestrial uploads of nutrients and organic matter originating from groundwater discharges, flash floods or river runoff as well as exchanges with seafloor strongly control the
productivity of these waters (Woodson and Litvin, 2014). The amplitude of seasonal variation of surface chlorophyll (Chl) and surface temperature is often higher on coastal waters compared to the open ocean (Cloern and Jassby, 2008). Furthermore, coastal topography and its interaction with winds, waves and currents generates a high variety of physicochemical niches for phytoplankton growth. Likewise, benthic-pelagic coupling allows the remineralization of nutrients present in shelf sediments during most intense storms. These episodic variations may constitute an important contribution to the overall productivity of
shelf waters. Because of the high spatiotemporal heterogeneity in the main coastal subsystems and the concomitant lack of data, most estimated carbon fluxes in these subsystems have relatively high uncertainties (Bauer et al., 2013). In addition, direct human activities and climate change lead to a long-term variation in terrestrial fluxes and coastal biogeochemistry that can potentially have important consequences for the global carbon cycle (Gregg et al., 2003).

In the Mediterranean Sea, coastal and shelf areas represent about 21 % of the global basin (259,000 km$^2$), which is a
higher contribution than for the global ocean (Pinardi et al., 2006). Although the Mediterranean Sea includes amongst the most oligotrophic areas of the world oceans, it can display marked spatial productivity variations related to the variety of regional climate and oceanographic conditions as well as to the multiple land-derived fluxes that locally fertilize the coastal waters (Goffart et al., 2002). Nutrient inputs from human activities in the coast and river discharges affect continental shelf productivity in this sea, sustaining locally enhanced pelagic and benthic biomass. Nevertheless, the influence of some river
flows has been notably reduced by damming affecting water chemistry and sediment loads and, thereby, the productivity of





coastal waters at local and regional scales (Ludwig et al., 2009; Tovar-Sánchez et al., 2016). Moreover, intensive agricultural practices and urbanization have brought unprecedented use and contamination of coastal groundwater (Basterretxea et al., 2010; Tovar-Sánchez et al., 2014). For example, the use of fertilizers has resulted in higher nutrient flowing into the Adriatic and in the lagoons of the Nile river, which has led to eutrophication (Turley, 1999a). However, the impact of this anthropogenic

nutrient enrichment may vary between regions, and modelling projections suggest spatial variations in PP as results of climate change.

Accurate quantification of the coastal PP is fundamental for assessment of global carbon cycling in the Mediterranean Sea. Changes in PP have important effects on fish stocks that are socially relevant because of the economical dependency of many Mediterranean coastal communities on marine food products. Several studies have assessed PP at the scale of the entire

Mediterranean Sea from satellite remote sensing data (Bosc et al., 2004; Bricaud et al., 2002b; Lazzari et al., 2012). However, coastal areas were generally ignored in such studies, so that their contribution to basin scale budgets is still largely unknown. Most coastal studies have a focus on specific regions and/or times (Estrada, 1996; Marty et al., 2002; Moutin and Raimbault, 2002; Rahav et al., 2013). Observed rates of climate change in the Mediterranean basin exceed global trends (Cramer et al., 2018) and future warming in the Mediterranean region is expected to be above global rates by 25 % (Lionello and Scarascia,

2018). Long-term responses of PP in coastal areas to climate forcing remain however uncertain because of the scarcity of adequate field datasets (Gasol et al., 2016).

In this study, we present major characteristics of pelagic PP in Mediterranean coastal waters based on satellite-borne observations for the period 2002–2016. First, we provide global estimations of PP in coastal waters and we assess their contribution to basin scale budgets, their interannual variability and long-term trends. Then, we regionalize the coastal waters

based on their temporal patterns of pelagic PP using Self-Organizing Maps (SOM) and we analyse the contribution of each region to total coastal PP.

## 2 Materials and Methods

### 2.1 Remote sensing data

We used the Mediterranean Sea Level-3 reprocessed surface chlorophyll concentration product (Chl L3) from multi satellite

observations, obtained from the EU Copernicus Marine Environment Monitoring Service (CMEMS) available at 1-day and 1-km resolution, from the website http://marine.copernicus.eu/ (OCEANCOLOUR_MED_CHL_L3_REP_OBSERVATIONS_009_073). Chl L3 dataset is derived by means of the Mediterranean Ocean Colour regional algorithms: an updated version of the regional algorithm MedOC4 (Mediterranean Ocean-Colour 4 bands MedOC4, Volpe et al., 2019) for Case-1 waters (deep pelagic waters) and the AD4 algorithm (ADriatic 4 band; Berthon and Zibordi, 2004;

D'Alimonte and Zibordi, 2003) for Case-2 waters (coastal shallow waters).

Level-2 Sea Surface Temperature (SST, ºC) at 1-day and 1-km from every available orbit from Moderate Resolution Imaging Spectroradiometer (MODIS) aboard satellites Terra and Aqua, downloaded from the National Aeronautics and Space



Administration (NASA) archive website (http://oceancolor.gsfc.nasa.gov/), were used for model calculations. Only night-time orbits were selected to avoid problems with skin temperature during daylight. Orbits with quality flag 2 in SST were included

after checking their validity and accuracy in order to have a more complete dataset. Daily (24-hour averaged) Photosynthetically Active Radiation (PAR, in E m$^{-2}$) was obtained as a Level 3 product at 9 km, the best available resolution from the NASA archive from both MODIS and Medium Resolution Imaging Spectrometer (MERIS).

All satellite-derived variables were remapped onto a regular 1-km spatial grid over the study area, by averaging all available pixels within each grid cell. For each parameter, outliers were removed whenever they exceeded about 3-times the

mean ± SD of the time series. For the purpose of this study, coastal areas were defined as the waters lying between 5 and 200 m depth. Only values at depths exceeding 5 m depth were considered in order to avoid any chlorophyll (Chl, mg m$^{-3}$) bias due to the bottom seagrass reflectance. The analysed time series covers the period from January 2002 to December 2016 for the Mediterranean Sea (30 to 46°N and 6°W to 37°E, Fig. 1).

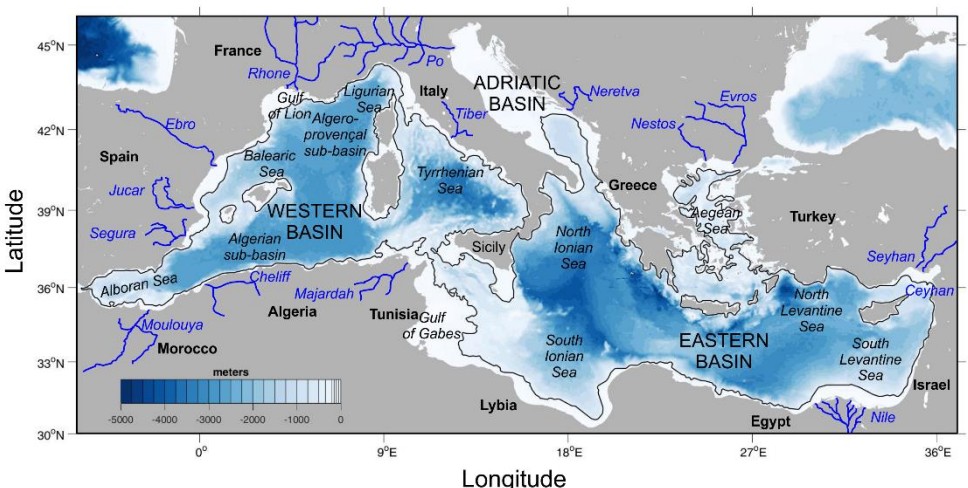

Figure 1: Map of the Mediterranean Sea showing the main basins, sea regions, surrounding countries and major rivers. Bathymetric data were obtained from ETOPO1 (Amante and Eakins, 2009). The black contour indicates the 200m isobath, the limit of coastal waters as defined in the present study.

**2.2 Primary production estimates**

PP was estimated from satellite-derived Chl, SST and PAR values using the time, depth, and wavelength-resolved light-

photosynthesis model of Morel (1991). This model was previously used for estimating PP in the Mediterranean sea (Antoine and André, 1995) and at global scale (Antoine et al., 1996; Antoine and Morel, 1996) and performs well when compared to *in situ* measurements (Campbell et al., 2002; Friedrichs et al., 2009) or when compared to other similar algorithms designed for





use with satellite observations (Carr et al., 2006; Saba et al., 2011). Instantaneous production at depth $z$ (m) of the water column, time $t$ of the day, and for absorption of irradiance at wavelength $\lambda$, P($\lambda$, z, t), is calculated as:


$$P\ (\lambda,\ z,\ t) = E\ (\lambda,\ z,\ t)\ Chl\ (z)\ a^*(\lambda,\ z)\ \Phi \qquad (mol\ C\ m^{-3}\ s^{-1}), \tag{1}$$

where $E(\lambda)$ is the spectral scalar irradiance (mol photons $m^{-2}$ $s^{-1}$), $a^*(\lambda)$ is the spectral chlorophyll-specific absorption coefficient of phytoplankton ($m^2$ mg $Chl^{-1}$), and $\Phi$ is the quantum yield of photosynthesis for carbon fixation (mol C mol

$photons^{-1}$; its possible spectral changes are ignored). Note that neither Chl, $a^*$ and $\Phi$ are made variable with time.

The triple integration of (1) w.r.t. wavelength, depth and time gives the daily column-integrated primary production, PP:

$$daily\ PP = 12\ \int_0^D \int_0^{\min(Z_p/Z_b)} \int_{400}^{700} P(\lambda, z, t)\ d\lambda\ dz\ dt (g\ C\ m^{-2}), \tag{2}$$

where the factor 12 is the conversion from moles to grams of carbon, $D$ is the day length (h), $Z_p$ (m) is the depth where the photosynthetically available radiation (PAR) falls to 0.1% of its value just below the sea surface (so approximately 1.5 times the euphotic depth), and $Z_b$ is the bottom depth taken from the ETOPO1 data base (Amante and Eakins, 2009). The time

integration used intervals equal to 1/30 of the day length (about 20 to 30 min depending on season). The depth integration used intervals equal to 1/50 of $Z_p$ and goes down to whichever is shallower between $Z_p$ and $Z_b$. The spectral integration is performed over the visible range (400 to 700 nm) with a 5 nm resolution.

The spectral irradiance at a given depth $z$, $E(\lambda, z, t)$, is calculated as (starting from just below the sea surface):


$$E(\lambda, z, t) = E(\lambda, z - dz, t) e^{[-K_d(\lambda, z)\ dz]}, \tag{3}$$

where the diffuse attenuation for downward irradiance, $K_d\ (\lambda,\ z)$ ($m^{-1}$), is computed as a function of chlorophyll following Morel and Maritorena (2001):


$$K_d(\lambda,\ z) = K_w(\lambda,\ z) + \chi(\lambda)\ Chl(z)^{e(\lambda)} \tag{4}$$

Details about how values are assigned to the parameters $a^*$ and $\Phi$, their dependence on temperature, and other features of this model, are to be found in Morel (1991) and Morel et al. (1996).





The model was operated both for clear sky conditions and for the actual MODIS PAR values, in which case a reduction of the clear-sky irradiance is uniformly applied across the entire day, as being the ratio of the satellite to clear-sky PAR daily values. Chl is assumed to be uniformly distributed with depth, and equal to the satellite-derived value. This simplification was considered more appropriate for the generally shallow and well-mixed waters of coastal areas than the use of global parameterization of the shape of the vertical profile as a function of the surface Chl value (e.g. Morel and Berthon, 155    1989; Uitz et al., 2006), whose validity outside of open ocean waters is not established.

From PP estimates, new ($PP_{new}$) and regenerated ($PP_{reg}$) production were calculated using the ratio of export production to total production (i.e., *ef*-ratio) (Laws et al., 2000; 2011). Indeed, assuming a steady state, the export production must equal the new production fuelled by new nutrients brought to the surface layers. The *ef*-ratio as a function of satellite-derived temperature and production can be obtained from the empirical relationship obtained by Laws et al. (2011):


$$ef = \frac{(0.5857 - 0.0165\,T)\,PP}{(51.7 + PP)} \tag{5}$$

$$PP_{exp} = PP_{new} = PP \times ef \tag{6}$$

$$PP_{reg} = PP - PP_{new} \tag{7}$$

where T is temperature in degrees Celsius (°C) and PP is the daily production (mg C m$^{-2}$).

We report annual PP estimates (Gt C) for the entire Mediterranean coastal areas ($\Sigma PP_{coast}$) and separately for the 170    Western, Eastern and Adriatic basins ($\Sigma PP_{basin}$). While some authors include the Adriatic in the eastern basin (e.g. Bosc et al., 2004), we treated it separately because its peculiarities (i.e. bathymetry, influence of rivers, eutrophic character) differentiate it from the rest of the Mediterranean Sea (Cushman-Roisin et al., 2001). Most of the Adriatic has a shallow (<200 m) bathymetry and it collects some 30% of the freshwater flowing into the Mediterranean, acting as a dilution basin for the nutrients discharged by the Po and other Adriatic rivers and becoming one of the most human-impacted regions of the 175    Mediterranean Sea (Ludwig et al., 2009; Micheli et al., 2013; Raicich et al., 2013).

### 2.3 Coastal regionalization

We used a two-step classification procedure to define coastal regions along the Mediterranean based on their temporal PP patterns. First, 9 regions (R1 to R9) were identified using a classification technique based on an unsupervised learning neural network (Self-Organizing Maps or SOM; Kohonen, 1982; 2001). Then, 18 alongshore marine ecoregions were obtained 180    considering the most relevant cross-shore limits of the SOM-derived regions (Z1 to Z18).



SOM is an unsupervised neural network method that reduces the high dimensional feature space of the input data to a lower dimensional network of units called neurons. SOM is especially suited to extract patterns in large datasets of satellite data (Ben Mustapha et al., 2014; Charantonis et al., 2015; Farikou et al., 2015). Unlike other classification methods, like *k*-means, SOM tends to preserve data topology (i.e. preserves neighbouring regions) and, therefore, it is particularly suited for pattern recognition (Liu and Weisberg, 2005). It allows adequate classification of areas with high spatial complexity and strong gradients. Similar neurons are mapped close together on the network facilitating the visualization of patterns and a topological ordination of the classified areas and the relative distance among neurons is obtained as results of the analysis.

For typical satellite imagery, SOM can be applied to both space and time domains. Here, we have addressed the analysis in the time domain of the datasets, which allows regionalizing the studied area on the basis of similitudes in the time variation of PP. We chose a map size of (3 x 3), with 9 neurons (for further details, see Basterretxea et al. 2018). We used a hexagonal map lattice in order to have equidistant neighbours and to avoid introducing anisotropy artefacts. For the algorithm initialization, we opted for linear mode, batch training algorithm, and '*ep*' type neighbourhood function since this parameter configuration produces the lower quantitative and topological error and computational cost (Liu et al., 2006). These SOM computations were performed using the MATLAB toolbox of SOM v.2.0 (Vesanto et al., 2000a, 2000b) provided by the Helsinki University of Technology (http://www.cis.hut.fi/somtoolbox/).

### 2.4 Climate data

To identify possible drivers of long-term PP variability we searched for correlations with two climate indices, the North Atlantic Oscillation index (NAO) and the Mediterranean Oscillation Index (MOI). The corresponding data were downloaded from the Climate Research Unit at the University of East Anglia (https://crudata.uea.ac.uk/cru/data/). Climate indices are defined either as anomalies of a climate variable, using the difference between two geographical points, or as principal components (Hurrell, 1995).

NAO is the central mode of climate variability of the Northern Hemisphere atmosphere. It is based on the pressure difference between the middle of the North Atlantic Ocean and Iceland, which affects winter conditions in the North Hemisphere (Hurrell and Van Loon, 1997; Marshall et al., 2001). Positive NAO results in a relatively dry winter in the Mediterranean but a warmer and wetter winter in northern Europe, and *vice versa*. Because of its influence on precipitation, Mediterranean river inflows are generally anti-correlated with the NAO (Trigo et al., 2006).

MOI is the most widely used teleconnection index for the Mediterranean basin. It reflects differences in temperature, precipitation, circulation, evaporation and other parameters between the eastern and western basin. There are different versions depending on the points of reference (Criado-Aldeanueva and Soto-Navarro, 2013). We used the version obtained as the normalized pressure difference between Gibraltar and Israel (Palutikof, 2003). Positive MOI phases are associated with increased atmospheric pressure over the Mediterranean Sea that promotes a shift of the wind trajectories toward lower latitudes leading to milder winters (Criado-Aldeanueva and Soto-Navarro, 2013). Under these conditions, reduced precipitation is observed in the southeastern Mediterranean region (Törnros, 2013). With some regional differences, NAO and MOI express





relatively similar climate patterns over the Mediterranean Sea. They are highly positively correlated in winter, and weakly but
still significantly correlated in summer  (Efthymiadis et al., 2011; Martínez-Asensio et al., 2014).

## 2.5 Statistical analyses

Linear temporal trends in the PP series were calculated using Theil-Sen slope adjustment (Sen, 1968) of the residuals of the
deseasonalized series. Only pixels with a trend statistically significant at the 95% level were considered. Correlation analyses
were performed using the Pearson Product Moment correlation. Differences between means were tested using the
Kolmogorov-Smirnov test (Massey et al., 1951).

## 3 Results

### 3.1 Coastal primary production

Annual primary production in coastal waters of the Mediterranean Sea ($\Sigma PP_{Coast}$) is estimated to be 0.041±0.004 Gt C, which
represents some 12% of total carbon fixation in the Mediterranean Sea (see Table 1 and 2). Approximately, 80% of this $\Sigma PP_{Coast}$
is sustained by recycling processes and, the rest, $PP_{new}$, is exported to the seafloor or to nearby areas. Although average surface
Chl concentration is 3-fold higher in coastal areas (0.3 mg m$^{-3}$) than in open areas (0.11 mg m$^{-3}$), the annual carbon fixation
per surface area ($PP_{annual}$) over the shelf is, on average, 26% lower than in the open ocean (100±91 and 136±40 g C m$^{-2}$
respectively; see Table 1 and 2). We would have expected that Mediterranean coastal annual PP would also be higher (about
1.7-fold) than oceanic annual PP. This hypothesis would have been observed if depth integration in coastal areas would have
been down to $Z_p$. However, depth integration in coastal areas is quite often stopped at a much shallower depth, i.e. $Z_b$, hence
the lower PP per unit area. The surface volumetric PP has been also estimated with a mean value of 2.93±9.60 g C m$^{-3}$ (Table
1).  This mean value of volumetric productivity could not have been compared with oceanic productivity in the Mediterranean
Sea considering that previous works presented only integrated production (e.g. Azov, 1986; Bosc et al., 2004; Bricaud et al.,
2002a; Coll et al., 2010; Krom et al., 1991; Macias et al., 2015).
235        Some differences in PP are observed between the more productive shelf waters in the western basin and those in the
eastern basin ($PP_{annual}$ =98±55 g C m$^{-2}$ and 92±96 g C m$^{-2}$, p<0.001), the Adriatic shelf being by far the most productive
($PP_{annual}$= 123±106 g C m$^{-2}$, Table 1). Annual carbon fixation is 97% higher in the eastern ($\Sigma PP_{east}$=0.021±0.002 Gt C) than in
the western shelf ($\Sigma PP_{west}$=0.011±0.001 Gt C), which is due to greater eastern surface area (about twice the western surface
area; Table 1). $PP_{annual}$ varies spatially from 90 to 250 g C m$^{-2}$ in the western shelf, and from 50 to 400 g C m$^{-2}$ in the eastern
basin where lowest values (<75 g C m$^{-2}$) are found mainly along the Gulf of Sirte. Contrastingly, $PP_{annual}$ exceeds 100 g C m$^{-2}$
in the Adriatic basin reaching values above 400 g C m$^{-2}$ in the north western coast (Fig. 2b). The most productive coastal
regions (>150 g C m$^{-2}$) are mainly located along the European coasts and seem to be related with the outflow regions of major
rivers. Indeed, the highest values of the coefficient of variation of primary production ($CV_{PP}$) are observed in the mouth of the
Ebro, Rhone, Tiber, Po, Neretva or Nestos/Evros rivers. Along the western African coast, $PP_{annual}$ displays values >150 g C m$^{-}$





$^2$; however, since the shelf is narrow, its contribution to $\Sigma PP_{Coast}$ is marginal (Fig. 1 and Fig. 2). The annual volumetric productivity follows a similar pattern than the annual integrated production with most values varying between 1-3 g C m$^{-3}$ but reaching up to > 10 g C m$^{-3}$ in the most productive coastal regions of the Adriatic Sea and the Gulf of Gabes (Fig. 3a).

**Table 1.** Primary production for the Mediterranean Sea, open ocean waters, and coastal waters during the period 2002–2016.
Surface area, annual mean surface Chl, annual average PP (PP$_{annual}$), annual integrated PP ($\Sigma PP$) and annual average productivity per unit volume PP (PPVOL$_{annual}$).

| | Surface area (10$^3$ km$^2$) | (%) | Chl (mg m$^{-3}$) | PP$_{annual}$ (g C m$^{-2}$) | $\Sigma PP$ (Gt C) | (%) | PPVOL$_{annual}$ (g C m$^{-3}$) | |
|---|---|---|---|---|---|---|---|---|
| Mediterranean Sea | 2,504 | | 0.19±0.78* | 140±40** | 0.349±0.118*** | | | |
| Open ocean waters | 1,975 | | 0.11±0.18* | 136±40** | 0.308±0.118 | | | |
| Coastal waters | 529 | 100 | 0.30±0.17 | 100±91 | 0.041±0.004 | 100 | 2.93±9.60 | 100 |
| Western shelf | 141 | 27 | 0.21±0.14 | 98±55 | 0.011±0.001 | 25.5 | 1.59±2.61 | 14 |
| Eastern shelf | 287 | 54 | 0.30±0.16 | 92±96 | 0.021±0.002 | 50.5 | 3.34±11.84 | 64 |
| Adriatic shelf | 101 | 19 | 0.39±0.23 | 123±106 | 0.010±0.001 | 24 | 3.27±7.23 | 22 |

* Mean surface Chl values obtained by averaging the 8-days and 4-km resolution of surface satellite Chl values obtained from CMEMS.
** PP estimated by averaging published satellite data shown in Table 2.
*** $\Sigma PP$ estimated from published data for open ocean waters and values from the present study.








**Table 2.** Compilation of published values of annual PP (PP$_{annual}$) and spatially integrated PP (ΣPP) estimations for the different Mediterranean basins.

| Region | Period (years) | PP$_{annual}$ (gC m$^{-2}$) | ΣPP (Gt C) | Method | Reference |
|---|---|---|---|---|---|
| Global estimation | - | 80 - 90 | | *In situ* ($^{14}$C method) | (Sournia, 1973) |
| | 1981 | 94±60 - 117.5±75[a] | | Satellite (CZCS) | (Morel and André, 1991b) |
| | 1979–1983 | 125 –-156[a] | 0.308-0.385[a] | Satellite (CZCS) | (Antoine and André, 1995) |
| | 1997–1998 | 190 | | Satellite (SeaWiFS) | (Bricaud et al., 2002a) |
| | 1998–2001 | 79.1 - 88.4 | | Satellite (SeaWiFS) | (Colella et al., 2003) |
| | 1998-2001 | 130 - 140 | | Satellite (SeaWiFS) | (Bosc et al., 2004) |
| | 1998-2007 | | 0.5 | Satellite (SeaWiFS) | (Uitz et al., 2010) |
| | 1998-2013 | 116 | | Satellite (5 sensors) | (O'Reilly and Sherman, 2016) |
| Western basin | 1980-1985 | 120 | | *In situ* (Oxygen) | (Bethoux, 1989) |
| | 1981 | 157,7 | | Satellite (CZCS) | (Morel and André, 1991b) |
| | 1979-1983 | 157 - 197[a] | | Satellite (CZCS) | (Antoine and André, 1995) |
| | 1996 | 140 - 150 | | In situ ($^{14}$C data) | (Conan et al., 1998) |
| | 1997-1998 | 198 | | Satellite (SeaWiFS) | (Bricaud et al., 2002a) |
| | 1991-1999 | 83 - 235 | | *In situ* ($^{14}$C method) | (Marty et al., 2002) |
| | 1996 | 175 | | *In situ* ($^{14}$C method) | (Moutin and Raimbault, 2002) |
| | 1997-1998 | 123 | | *In situ* ($^{14}$C method) | (Van Wambeke et al., 2004) |
| | 1997-1998 | 152 | | *In situ* ($^{14}$C method) | (Lefèvre et al., PANGEA 2001) |
| | 1998-2001 | 93.8 – 98.8 | | Satellite (SeaWiFS) | (Colella et al., 2003) |
| | 1998-2001 | 163±7 | | Satellite (SeaWiFS) | (Bosc et al., 2004) |
| | 2006-2007 | 858[c,d] | | *In situ* (dark-light method) | (Regaudie-De-Gioux et al., 2009) |
| Eastern basin (including Adriatic) | 1980-1985 | 137 - 150[b] | | *In situ* (Phosphorous) | (Béthoux et al., 1998) |
| | 1981 | 109.4 | | Satellite (CZCS) | (Morel and André, 1991b) |
| | 1979-1983 | 110 - 137[a] | | Satellite (CZCS) | (Antoine and André, 1995) |
| | 1997-1998 | 183 | | Satellite (SeaWiFS) | (Bricaud et al., 2002a) |
| | 1996 | 96 | | *In situ* ($^{14}$C data) | (Moutin and Raimbault, 2002) |
| | 1998-2001 | 69.1 - 81.5 | | Satellite (SeaWiFS) | (Colella et al., 2003) |
| | 1998-2001 | 121±5 | | Satellite (SeaWiFS) | (Bosc et al., 2004) |
| | 2006-2007 | 521[c,d] | | *In situ* (dark-light method) | (Regaudie-De-Gioux et al., 2009) |
| Adriatic basin | 1978-1983 | 241 - 301[a] | 0.0235 | Satellite (CZCS) | (Antoine and André, 1995) |
| | 1998-2001 | 92.4 - 104.4 | | Satellite (SeaWiFS) | (Colella et al., 2003) |

[a]The estimates from Antoine et al. (1995) and Morel and André (1991) have been corrected by a factor of 1.25 as recommended by Morel et al. (1996).
[b]From Colella et al. (2003), who estimated it using *f*-ratios (the ratio between new and total PP) obtained from Boldrin et al., (2002).
[c]Daily PP (mgC m$^{-2}$ d$^{-1}$) converted to annual PP (mgC m$^{-2}$ y$^{-1}$) multiply by 365.
[d]Conversion to carbon unit using photosynthetic quotient PQ =1.




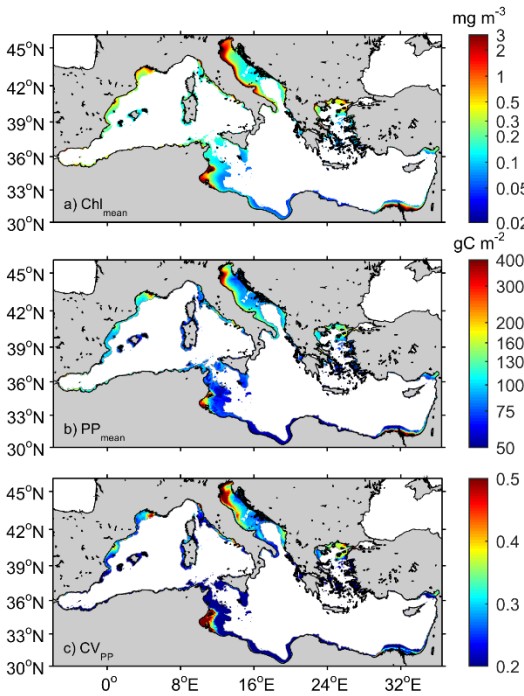

Figure 2: Mean distribution of a) chlorophyll, b) primary production (in g C m$^{-2}$) and, c) coefficient of variation of PP values (CV$_{PP}$).

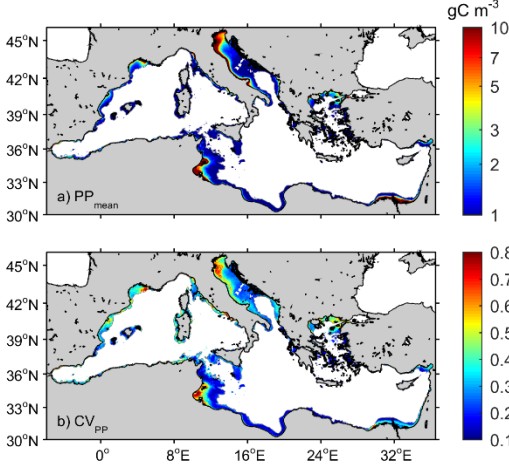

Figure 3: Mean distribution of a) primary productivity (in g C m$^{-3}$) and b) coefficient of variation of productivity values values (CV$_{PP}$).



## 3.2 Long-term variability and trends

As shown in Fig. 4, variability in annual PP is dominated by short-scale variations (i.e. subdecadal). $\Sigma PP_{Coastal}$ exhibits moderate
interannual variability (up to 25%) whereas basin scale interannual variations range from 26% in the Adriatic basin, up to 28%
in the western basin and 29% in the eastern basin. When considering the whole basin, positive anomalies in coastal PP extended
between 2004 and 2011 (mean $0.044\pm0.001$ Gt C y$^{-1}$; Fig. 4a). Conversely, year 2012 was particularly unproductive in all three
basins (specific annual mean PP for 2012 were 0.037 Gt C y$^{-1}$ for the whole basin, 0.010 Gt C y$^{-1}$ for the western, 0.019 Gt C y$^{-1}$ for the eastern and 0.009 Gt C y$^{-1}$ for the Adriatic basin). This negative anomaly marked the beginning of a less productive
period, particularly noticeable in the eastern basin (Fig. 4c).

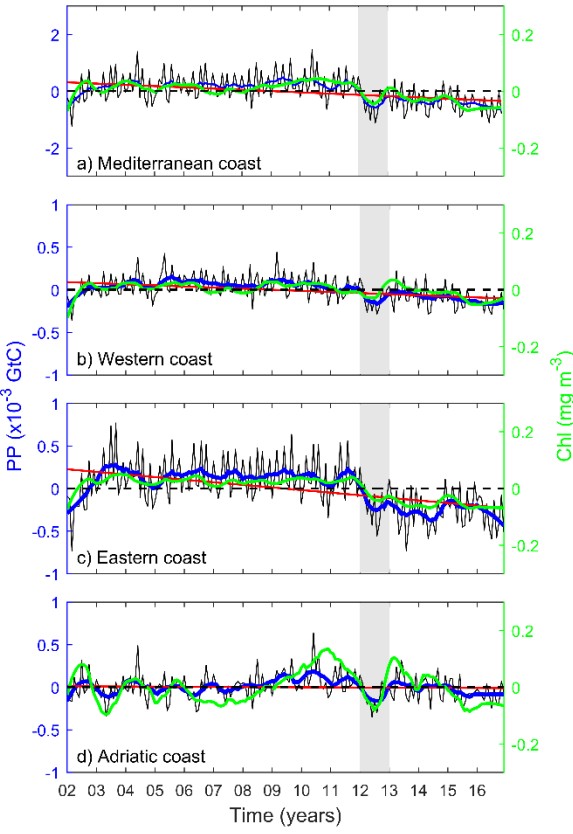

Figure 4:  PP variability and trends for coastal waters in a) the whole Mediterranean, b) western basin, c) eastern basin and d)
the Adriatic coast. Black solid lines indicate the original monthly $\Sigma PP_{Coastal}$ anomalies and the filtered low frequency signal is
overlaid in blue. Green solid lines indicate the filtered low frequency signal for Chl anomalies (mg m$^{-3}$). The red line
300              indicates the PP trend during the analysed period (2002–2016) and the grey band indicates year 2012.



Long-term trends in PP at 95% of confidence level are significant at basin scale and also in the western and in the eastern basins (p<0.05; Fig. 4a-c) whereas the Adriatic Sea does not display significant trends. However, while a slight negative tendency in seen in the western coast (-10.73 TC per decade. Fig. 4b), a more dramatic tendency, driven by a shift in year 305   2012, is observed in the eastern basin (-25.39 TC per decade; Fig. 4c). As revealed by Fig. 5a, some regionally coherent patches of significant trend in PP are observed along the coast. Most of these regions presented declining PP trends, particularly along the north African coast where SST temperature is increasing at a higher rate (Fig. 5b). Typical PP trend magnitudes observed along the Spanish Mediterranean and the North African coast from the Gulf of Gabes range from -0.05 to +0.05 g C m$^{-2}$ per decade. Some positive PP trends, exceeding +0.1 g C m$^{-2}$ per decade, can be determined in some coastal regions of the north 310   of the Adriatic Sea.

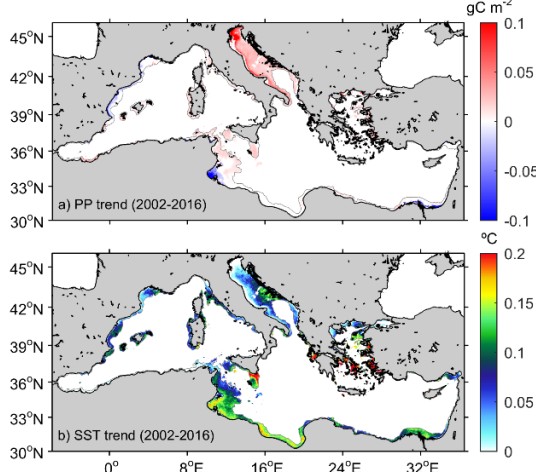

Figure 5: Trends in primary production and sea surface temperature. Values correspond to the change per decade. a) Theil-Sen trend in pelagic primary production estimated from daily values for the 2002–2016 period. b) Trend in SST temperature. Only significant trends (p < 0.05) are shown.

A significant negative correlation was observed between coastal ΣPP and SST  (r=-0.63, p< 0.001; Fig. 6a) showing that the important decrease of Chl over the years was able to compensate the effect of temperature increase. In addition, we observed evidence of inverse relationship between PP variability and the phase of the climate indices NAO and MOI (r=-0.45, p < 0.001 and r=-0.22, p < 0.001 respectively; Fig. 6b-c). The response of PP to climate variations varied seasonally. Indeed, NAO influenced coastal PP in summer, both in the western and in the eastern basin (r=0.25, p=0.06; r=0.22, p=0.08 320   respectively) and MOI variations were better correlated with PP global variations in spring (r=0.28, p=0.04), showing a higher impact in the Adriatic basin (r=0.37, p=0.02). No significant correlation was found during the winter nor fall season for any index.



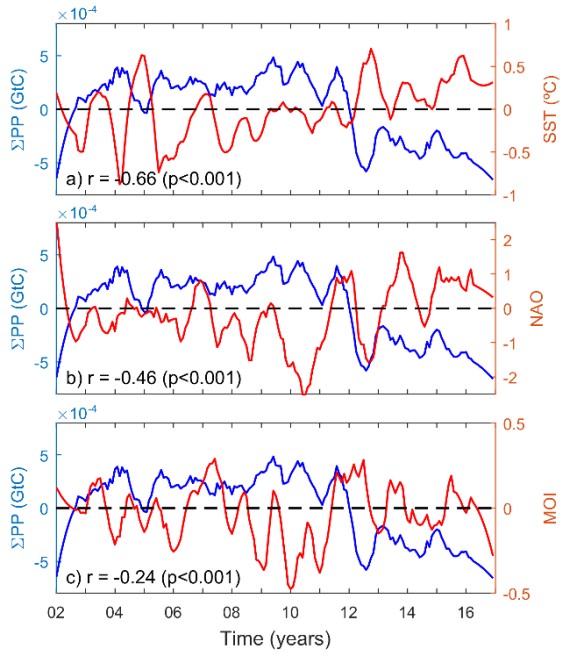

Figure 6: Relationship between coastal pelagic primary production (ΣPP, blue lines) and a) SST anomalies, b) NAO index
and c) MOI index (red lines).

## 3.3 Coastal regionalization

The nine characteristic temporal PP patterns, their corresponding spatial distribution obtained from the SOM analysis and the
18 zones in which the coastal region was classified are shown in Fig. 7. Generally, wider shelves present higher spatial
complexity manifested as a larger number of SOM patterns. About 78% of the shelf waters include R1, R2, R3 and R4 patterns.
In particular, R1 and R2 characterize areas of low production with scarce seasonality ($PP_{annual}$=44±17 and 69±22 g C m$^{-2}$,
respectively; Table 3) typically occurring in the southern and eastern Mediterranean (12 and 29% of the total surface area).
They are representative of the productivity patterns in vast shelf regions in the Gulf of Gabes and Sirte and in the central
Aegean (Z17, Z16, Z14 and Z11). R3 and R4 correspond to higher production and a wider range of variation ($PP_{annual}$=90±32
and 98±39 g C m$^{-2}$; Table 3 and Fig. 7). While R3 (18.6% of total coastal surface) is frequent in shelf regions of the western
basin (Z2, Z3, Z5 and Z7), R4 extends over the deepest areas of the Adriatic Sea (Z9 and Z10) Both patterns are representative
of variations observed in 36.9% of the coastal waters.

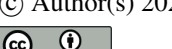

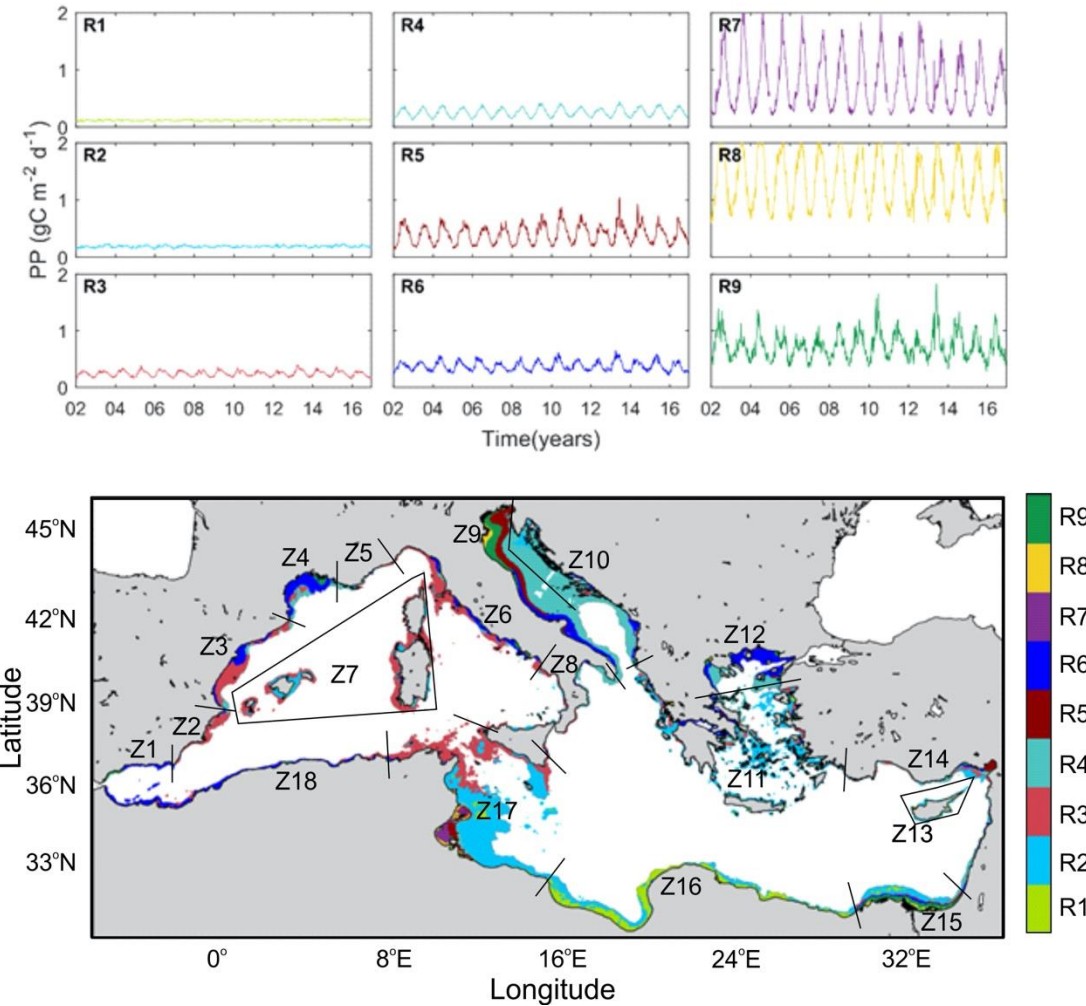

Figure 7: Regionalization of the coastal waters in the Mediterranean Sea based on their temporal patterns of pelagic primary
340      production. a) Characteristic temporal patterns of PP obtained from SOM classification (R1 to R9) and b) coastal regions
defined from alongshore variations of the SOM – regions (Z1 to Z18).

345



**Table 3.** Primary production for each of the SOM-defined regions. Coastal surface area and its relative contribution to total coastal water surface (%). Mean annual PP (PP$_{annual}$), annually integrated PP ($\Sigma$PP) and its contribution in each SOM-defined region to the total coastal PP.

350

|  | Area | | PP$_{annual}$ | $\Sigma$PP | |
| --- | --- | --- | --- | --- | --- |
|  | (km$^2$) | (%) | (g C m$^{-2}$) | ($10^{-3}$ Gt C) | (%) |
| R1 | 61,052 | 12.2 | 44±17 | 1.98±0.24 | 4.8 |
| R2 | 149,766 | 29.0 | 69±22 | 8.89±0.66 | 21.5 |
| R3 | 94,562 | 18.6 | 90±32 | 7.31±0.54 | 17.6 |
| R4 | 93,103 | 18.3 | 98±39 | 7.81±0.45 | 18.8 |
| R5 | 22,566 | 5.0 | 154±102 | 2.95±0.27 | 7.1 |
| R6 | 47,977 | 9.8 | 140±70 | 5.50±0.58 | 13.3 |
| R7 | 5,816 | 1.8 | 288±217 | 1.41±0.24 | 3.4 |
| R8 | 7,822 | 2.2 | 508±283 | 2.67±0.67 | 6.5 |
| R9 | 13,181 | 3.2 | 281±175 | 2.95±0.38 | 7.1 |

As shown in Table 3, areas of low production with seasonal patterns R1 to R4 contribute to more than 62% of total pelagic carbon fixation in Mediterranean coastal areas. In contrast, systems of higher production (PP$_{annual}$>280 g C m$^{-2}$ d$^{-1}$) 355 barely contribute to 17% of total production. These regions of enhanced production are generally constrained to Regions of Freshwater Influence (ROFIs; Simpson, 1997) where terrestrial nutrients fuel coastal production. Indeed, R8 is almost exclusively restricted to the river mouths and it presents elevated PP values (1.29±0.50 gC m$^{-2}$ d$^{-1}$) and a wide range of variation of 0.67 to 2.14 gC m$^{-2}$ d$^{-1}$. R7 pattern is exclusively located in the shallowest inner shelf of the Gulf of Gabes and it is bounded by R5, a transition region between the inner and outer shelf. Unlike the other regions, where PP peaks in late winter-spring, 360 maximum PP in R7 occurs in fall. Finally, R5 and R6 patterns correspond to transition regions accounting for 20.4% of the total production and 14.8% of the Mediterranean coast. While R5 mainly occurs in deltas areas, R6 is characteristic of the western Mediterranean shelf, including the North African coast (0.36-0.08 gC m$^{-2}$ d$^{-1}$; Fig. 7).

The SOM-based regionalization reveals two groups of coastal waters: those with low cross-shore variability and including only one or two SOM regions (i.e. Z1, Z13, Z16 and Z18) and those with strong cross-shore gradients including 365 several SOM-regions (i.e. Z4, Z9, Z12, Z15). The first pattern is typically observed in narrow continental shelf areas with low influence of river inputs whereas the second group is found in regions with wider continental shelf such as ROFIs (the Rhone delta, the north and western coastline of the Adriatic Sea and the Nile Delta) and in the Gulf of Gabes. The western Adriatic (Z9) and the Gulf of Gabes (Z17) are the largest contributors to $\Sigma$PP$_{Coast}$, contributing together to 35.9% of shelf production in the Mediterranean Sea but, in the case of Z17, it is mainly due to its large extension (Table 4). PP is also high in the northern 370 Alboran Sea (Z1), Nile delta (Z15), the western Adriatic (Z9), and Gulf of Lions (Z4; Table 4). With the exception of Z1, influenced by the entrance of waters from the Atlantic Ocean and by local coastal upwelling, these zones receive important riverine fluxes ($Q$).





**Table 4.** Surface, river discharge flow ($Q$), annual mean PP (PP$_{annual}$), annual integrated PP ($\Sigma$PP) and its contribution respect to the total coastal Mediterranean Sea PP for each of the 18 alongshore zones characterized in the Mediterranean Sea.


| | Area | | Q | PP$_{annual}$ | $\Sigma$PP | |
|---|---|---|---|---|---|---|
| | (km$^2$) | (%) | (km$^3$ y$^{-1}$) | (g C m$^{-2}$) | (10$^{-3}$ Gt C) | (%) |
| Z1 | 1,869 | 0.4 | 0.5 | 215 ±124 | 0.22±0.05 | 0.4 |
| Z2 | 7,226 | 1.4 | 1.2 | 107±58 | 0.62±0.08 | 1.4 |
| Z3 | 18,870 | 3.6 | 21.4 | 104±47 | 1.71±0.16 | 3.6 |
| Z4 | 15,196 | 2.9 | 57.7 | 128±72 | 1.61±0.12 | 2.9 |
| Z5 | 878 | 0.2 | 1.9 | 84±33 | 0.04±0.02 | 0.2 |
| Z6 | 20,392 | 3.9 | 14.6 | 101±64 | 1.62±0.20 | 3.9 |
| Z7 | 29,666 | 5.7 | 0.5 | 74±26 | 1.66±0.20 | 5.7 |
| Z8 | 8,178 | 1.6 | 3.7 | 81±34 | 0.40±0.09 | 1.6 |
| Z9 | 64,780 | 12.4 | 70.5 | 140±124 | 7.63±0.66 | 12.4 |
| Z10 | 40,997 | 7.8 | 35.8 | 89±37 | 2.81±0.25 | 7.8 |
| Z11 | 58,252 | 11.1 | 21.5 | 81±59 | 2.95±0.67 | 11.1 |
| Z12 | 25,720 | 4.9 | 21.2 | 123±76 | 2.25±0.33 | 4.9 |
| Z13 | 30,71 | 0.6 | 0 | 53±18 | 0.09±0.02 | 0.6 |
| Z14 | 16,814 | 3.2 | 21.3 | 97±61 | 1.21±0.14 | 3.2 |
| Z15 | 28,544 | 5.5 | 17 | 170±182 | 4.02±0.50 | 5.5 |
| Z16 | 46,065 | 8.8 | 0 | 48±17 | 1.85±0.12 | 8.8 |
| Z17 | 123,071 | 23.5 | 1.1 | 90±87 | 9.72±0.80 | 23.5 |
| Z18 | 13,411 | 2.6 | 6.1 | 125±56 | 1.24±0.18 | 2.6 |

## 4 Discussion

### 4.1 Coastal primary production

To our knowledge, this is the first study focused on the contribution of coastal waters to the overall pelagic PP in the

Mediterranean Sea. While the mean coastal values for the Mediterranean (100±91 g C m$^{-2}$) are somewhat lower that the mean values over the continental shelves of the World ocean (160±40 g C m$^{-2}$; Smith and Hollibaugh, 1993), the impact of coastal pelagic PP to total basin production (12%) is in the high range of the estimations for other Seas (Muller-Karger et al., 2005). This estimation is subject to the uncertainties inherent to using satellite ocean colour, which is limited to the upper ocean (down to 20 m at best in clear waters) and has poor performance in some areas (i.e. Case-2 waters). It nevertheless provides an

assessment of net rates of carbon fixation in coastal areas that is consistent with global estimations of the contribution of coastal areas to oceanic production (Gattuso et al., 1998; Ducklow and McCallister, 2004; Muller-Karger et al., 2005). Bias in coastal Chl estimations is mainly due to the presence of non-phytoplankton components such as coloured dissolved organic matter (CDOM) or other terrestrial substances (Morel et al., 2006). These compounds originate from coastal erosion, resuspension in shallow areas, river inputs or anthropogenic effluents. Likewise, they affect the propagation of photosynthetic radiation through

the water column (Morel, 1991). However, the possible uncertainties and biases caused by Chl estimation through satellite





data might have alter very weakly our estimation of coastal PP. Indeed, Case-1 waters are largely predominant in the coastal Mediterranean regions whereas Case-2 waters are reduced to less than 5% of the whole basin. In particular, they are confined to the north Adriatic Sea, Gulf of Gabes and around Nile delta (Antoine et al., 1995; Morel and André, 1991; Bosc et al., 2004) where our PP estimations may present larger uncertainties. However, $PP_{annual}$ values off the Nile river delta, >100 g C m$^{-2}$

estimated here, are only slightly higher than those reported by Antoine et al. (1995) (80-100 g C m$^{-2}$). Highest values have been reported for this region (>300 g C m$^{-2}$) but, as shown in Fig. 2, they are restricted to a narrow coastal band. In the case of the Adriatic Sea, Umani (1996) reported values of PP from 50 to 200 g C m$^{-2}$ y$^{-1}$, while Zoppini et al. (1995) estimated PP rates from 210 to 260 g C m$^{-2}$ y$^{-1}$ in the northern coastal areas. Our estimations range between 100 and >350 (with mean values of 123±106 g C m$^{-2}$).

Because of its extension, the eastern basin contributes more than the western basin to overall coastal production (50% and 25% respectively; Table 1). Furthermore, carbon fixation from the Adriatic Sea represents 24% of total coastal production, which is significant considering the area of this Sea (19% of Mediterranean coastal waters). The relevance of the contribution of the Adriatic Sea in overall coastal PP relies in two main characteristics; (1) coastal waters (<200 m) constitute a large part of the Adriatic Sea and (2) about one third of the river discharge in the Mediterranean is concentrated in the Adriatic Sea (see

Table 4). Indeed, patterns in the northern Adriatic Sea reflect a variation in the drivers of PP with respect to other regions. For example, while internal processes (i.e. vertical diffusion and mixing) and, less so, atmospheric deposition, drive PP in most coastal waters, production in the north Adriatic would be mainly driven by fluvial sources of carbon and regeneration through bacterial pathways (Umani et al., 2007). Moreover, distinctive dynamics in this sea is driven by the influence of river outflows on stratification and general circulation patterns (Djakovac et al., 2012; Giani et al., 2012).

Here, from *ef*-ratios, we estimated that on average only 22±20 % of the production in the coastal Mediterranean Sea is new and the rest is sustained *via* regenerated sources (Fig. 8a-c). This $PP_{new}$ value (Fig. 8b) is comparable to the mean organic carbon that sinks to the sea floor (28%) estimated from Muller-Karger et al. (2005) and Pace et al. (1987) but higher than $PP_{new}$ estimations provided by Vidussi et al. (2001) for oceanic waters in the eastern basin (15% of total production). Contrarily to Vidussi et al. (2001) who estimated $PP_{new}$ in the eastern basin, the coastal $PP_{new}$ average here includes both eastern

and western basins of the Mediterranean, but also the highly productive areas in the northern Adriatic. This could explain that the $PP_{new}$ observed here is higher than the one observed for oceanic waters in the eastern basins. Additionally, high *ef*-ratios (> 0.3) are observed in our case in the areas where nutrient inputs from the Atlantic and river effluents significantly enhance $PP_{new}$ (Fig. 8a). Furthermore, *ef*-ratios present significant seasonality, varying between 0.26±0.04 in the most productive winter-spring season and 0.15±0.02 in summer, when the water column is strongly stratified and food web shifts to a more

recycling dominated system.


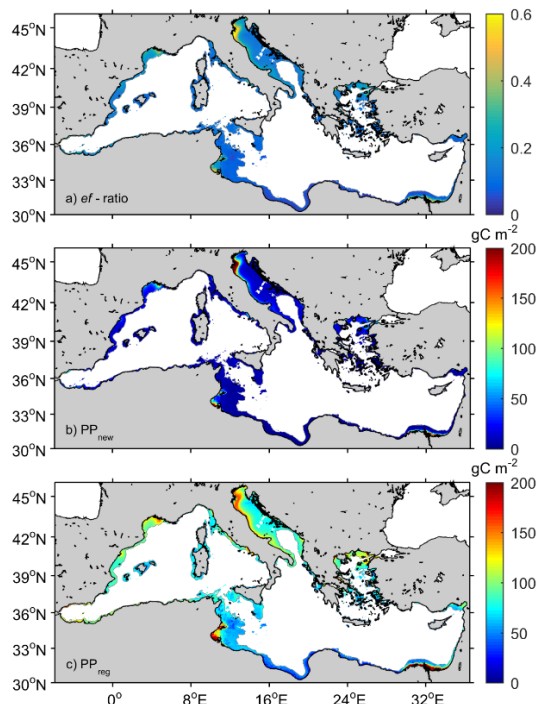

Figure 8. Mean seasonal PP over the 2002–2016 period for coastal Mediterranean areas (< 200 m isobath). Quarterly composites: winter (Dec-Jan-Feb), spring (Mar-Apr-May), summer (Jun-Jul-Aug) and autumn (Sep-Oct-Nov).

## 4.2 Long-term variability and trends

Available satellite ocean colour data span about 20 years, so that temporal trends derived from their analysis are highly depending on decadal variability (Henson et al., 2010). Despite these limitations, satellite observations of ocean colour over the past two decades suggest relationship between warming and reduced productivity in permanently stratified areas (Behrenfeld et al., 2006). Since clear tendencies of warming are observed in the Mediterranean Sea (Nykjaer, 2009; Pastor et al., 2018), intensification of stratification would decrease nutrient supply to phytoplankton and, thus, decrease PP (Behrenfeld 430 et al., 2006; Stambler, 2014).

Barale et al. (2008) observed a general decrease in Chl biomass in the Mediterranean Sea over the period 1998–2003. However, some coastal areas in their study displayed the opposite tendency. Macias et al. (2015) anticipated no future global changes of integrated PP in the Mediterranean Sea from modelling results. They predicted a tendency to oligotrophication in the western basin and increase in the productivity of the eastern basin. Our study reveals that $\Sigma PP_{Coastal}$ in the Mediterranean 435 Sea varies nonlinearly and a reduction of carbon fixation rates is observed since 2012 (Fig. 4a). Overall negative trends are reported here in both the Western and in the Eastern basin (-10.70 and -25.39 TC per decade; Fig. 4b-c). A spatial analysis of the long-term decadal variability reveals weak but spatially coherent and significant tendencies ($p<0.05$; Fig. 5). In particular, PP declines along the coasts of Spain and Africa. Conversely, trends in some areas of the Adriatic Sea are markedly positive





(> 0.1 gC m$^{-2}$ per decade) mainly in the proximity of the Po river. While negative tendencies seem to fit with the assumed

model of PP limitation associated with increasing temperatures, the origin of the positive trend in the Adriatic basin is more uncertain. A plausible explanation is the variation in the flux or/and loads of the northern Adriatic rivers. For example, Giani et al. (2012) observed an increase of the Po River flow increasing phosphate and dissolved nitrogen concentrations in the Po's delta and its surrounding shelf waters that could enhance the coastal PP of this region. Alternatively, changes occurring between 2004 and 2006 in the deep structure of the Mediterranean Sea could have affected mass and nutrient exchanges

between the Adriatic and the north Ionian Sea (Font et al., 2007; Schroeder et al., 2008; Šolić et al., 2008; Vilićić et al., 2012).

Long-term decadal variations in the eastern and western basins are mostly coupled, suggesting that they share the same PP drivers at this basin scale (Fig. 4b-c). A major feature in the interannual pattern is a global decrease in production in 2012 that is extended to the following years in the eastern basin. Durrieu de Madron et al. (2013) reported peculiar atmospheric conditions in the Mediterranean Sea during 2012 that triggered a massive formation of dense water on the continental shelf

and in the deep basin of the Gulf of Lions. A similar anomaly was described in the Adriatic shelf where unprecedented dense water generation was preconditioned by a dry and warm year resulting in a significant reduction of coastal freshwaters and basin-wide salinity increase (Mihanović et al., 2013; Raicich et al., 2013). Additionally, Pastor et al. (2018) observed an anomalously temperature increase in the Mediterranean Sea during summer 2012. From our analysis, we infer that this climate-related event had strong influence on the global coastal PP of the Mediterranean Sea.

Several studies have reported influence of climate variations in the coast (Belgrano et al., 2008; Cloern et al., 2007; Tiselius et al., 2016). In agreement, we observed an influence of climate scale variability on coastal productivity as suggested by the inverse correlations between ΣPP and SST and, more loosely, with NAO and MOI (Fig. 6). While these correlations emphasize the pre-eminent role of climate variability in the regulation of interannual to decadal scale coastal productivity, the pathways through which this control of the atmosphere over coastal productivity is exerted are complex and may regionally

differ (Grbec et al., 2009). Climate can influence phytoplankton growth by the direct effect of temperature on algal metabolism, by changes in basin scale circulation (including exchanges with adjacent seas), by regulating nutrient supply through variations in the thermocline intensity, by changes in wind patterns affecting mixing and dust deposition pathways or through changes in precipitation that have direct influence on wet deposition and on river runoff. These effects are modulated by changes in the biota and in the interaction between organisms (e.g. Molinero et al., 2005). The relative importance of climate-driven processes

relative to other productivity enhancing processes depends on regional characteristics and may be seasonally varying. For example, variations in dust deposition, which may sustain up to 50% of new production in the Levantine basin ( Kress and Herut, 2001; Herut et al., 2002), are expected to be more important in the eastern and southern Mediterranean coasts because of their proximity to the Saharan dust sources. Likewise, variations in cooling and vertical mixing are expected to be more effective during late winter when PP peaks and when diatoms dominate in the Mediterranean Sea (Lacroix and Nival, 1998;

Marty, 2002; Marty and Chiavérini, 2010).

Our results reveal that, in contrast to other regions like the North Sea (Capuzzo et al., 2018) or the Arctic Ocean (Gregg et al., 2003), the coastal Mediterranean Sea did not globally display a marked decline in PP during the last decades.





We suggest that in some coastal areas, a decrease in vertical nutrient supply though the thermocline may be compensated by other nutrient sources. Variations in atmospheric deposition, groundwater and river outflows together with the influence of
human activities through changes in landscape use and nutrient management are important sources of nutrient in the ecosystem and thus, act as major drivers of PP in these waters (e.g. Paerl et al., 1999). As a consequence of human activities, both terrestrial and coastal ecosystems have experienced progressive nutrient enrichment (Conley et al., 2009; Deegan et al., 2012). However, while this effect is evidenced in shallow nearshore waters, its influence in the ocean is estimated to be minimal (Wang et al., 2018). In the Mediterranean Sea, high coast population growth rates and concomitant food demand have resulted
in dramatic increase of water demand for irrigation farming and fertilizer use (Ryan, 2008). Indeed, while the freshwater discharge of Mediterranean rivers has significantly reduced during recent decades (~20%), the corresponding total nitrogen inputs to coastal seas are estimated to have increased by a factor up to 5, fuelling PP in river influenced areas (Ludwig et al., 2009). While the importance of groundwater in the Mediterranean Sea could be comparable to that of rivers (Rodellas et al., 2015) and generalized nitrification of Mediterranean coastal aquifers is acknowledged (EEA, 1995; Zalidis et al., 2002),
general trends in groundwater discharges remain largely unknown.

### 4.3 Coastal regionalization

Coastal regionalization reveals marked differences in coastal waters PP in the Mediterranean Sea. Annual values range from $215\pm124$ g C m$^{-2}$ in the north Alboran Sea (Z1) to $48\pm17$ g C m$^{-2}$ along the coasts of Egypt and Libya (Z16). These values are globally lower than published data; yet, literature values in coastal waters are highly variable depending on methodology,
depth and/or sampling date. For example, García-Gorriz and Carr (2001) estimated annual PP of 300-900 g C m$^{-2}$ for Z1 but Morán and Estrada (2001) narrowed this range to mean values between 121 and 366 g C m$^{-2}$ depending on distance from the coast. Pugnetti et al. (2008) reported mean values of 150 g C m$^{-2}$ that are almost twice higher than our values at Z8. In the lower range, Sournia (1973) estimated 30–60 g C m$^{-2}$ in Z16, which is in accordance with our values for this zone. In this sense, despite the limitations inherent to satellite data, the present work provides estimations based on a long data record (14
years) and a homogeneous methodology.

The nine characteristic temporal patterns obtained from the SOM analysis (Fig. 7) reveal small differences in PP among the different regions. Most variations are due to changes in the magnitude of annual carbon fixation, although seasonality little varies. Exceptions are R7, R8 and R9 which represent the dynamics in coastal regions regulated by terrestrial inputs. Likewise, interannual variations are highly coherent among regions, following the basin-scale pattern shown in Fig. 4,
including the remarkable decline in productivity during 2012. Exceptions are R1, representing the dynamics in the Gulf of Sirte and R7 in Gulf of Gabes where a different interannual variability suggests alternative sources of PP variability in this region. Indeed, the Gulf of Gabes is a peculiar region displaying consistently high Chl and PP in most studies (e.g. Bosc et al., 2004; Barale et al., 2008). Drira et al. (2008) reported high biomass and toxic dinoflagellate blooms in the inner shelf of the Gulf of Gabes where surface nitrate concentration often exceeded 1µM. This enrichment is associated with degradation of the
water quality attributed to industrial and urban activities (Hamza-Chaffai et al., 1997; Zairi and Rouis, 1999). However, even





though these waters may suffer from eutrophication, satellite-borne data overestimates Chl within these waters, as revealed by Katlane et al. (2011) who observed constant high turbidity and suspended matter of industrial origin affecting these waters but also, reflection from the bottom affecting MODIS data. This suggests that general Chl algorithms may be particularly inaccurate in this region.

510       The magnitude of coastal PP has been often related to both shelf width and magnitude of river discharge (Liu et al., 2010). Our data does not display a general relationship between shelf width ($Q$) and $PP_{annual}$ (Table 4 and Fig. 9). Indeed, wide shelves with important river discharge flux from the Po, the Rhone and the Nile rivers display high productivity (Z4, Z9 and Z15 > 170 g C m$^{-2}$) whereas production is low in narrowest shelves like Z2, Z5 and Z16. However, PP in some regions with important river inflows, like Z10, are significantly lower (89±37 g C m$^{-2}$). In other regions like Z1 and Z8 PP is high despite

the lack of important freshwater sources.

      The role of river discharges depends both on $Q$ and on nutrient loads. Most Mediterranean rivers have lost their natural flows and their discharges to the sea are strongly regulated by dams and water abstractions. Consequently, their outflow to the Mediterranean Sea is highly uncoupled from weather and climate variability. For instance, some rivers flowing into the Adriatic and Ionian Seas, like the Acheloos, Nestos or Aliakmon nowadays present high to maximum discharge in July due to peak

hydropower production (Skoulikidis et al., 2009).

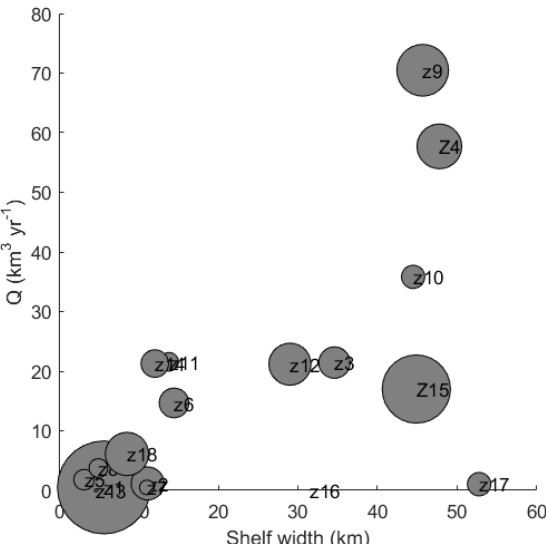

Figure 9: Relation between primary production, shelf width and river discharge flow ($Q$). Bubbles sizes are proportional to the annual mean PP (PPannual) of each of the 18 defined zones (see Fig. 8 and Table 4).

525       Nutrient and organic matter loads have globally increased during the last century (Beusen et al., 2016). Concentrations exported to nearby seas depend on the combined effects of lithology, urban effluents, industry and agriculture in catchment basins that are often difficult to quantify. The Land Use and Land Cover (LULC) data collection provide indices of the threat


of potential development for setting land and water quality policies. Rivers like the Rhone and Po with important influence on coastal productivity flow through extensive areas and therefore accumulate the impact from anthropogenic activities.

Agricultural practices and urban effluents can strongly determine the concentration and molar ratios of the nutrients flowing into coastal waters. For example, despite the flux of the Nile river has been drastically reduced after the operation of the Aswan Dam (from 47 to 17 km$^3$ y$^{-1}$, Ludwig et al. (2009)), the coastal region is still highly productive. A remarkable increase in the concentrations of nitrate derived from fertilizers and sewage is responsible for this sustained productivity (Turley, 1999; Nixon, 2003, 2004). Conversely, pollution pressures in the western Balkan basins are relatively low and the Neretva (Z10), running

through a karstic region in Croatia, displays low nutrient levels (Ludwig et al., 2009; Skoulikidis et al., 2009).

Finally, other oceanographic processes determine the productivity of coastal regions. In particular, Z1 and Z18, in the Alboran Sea are comparatively more productive than other areas. The influence of winds and circulation patterns favouring subsurface water upwelling higher productivity in the northern Alboran Sea where described by García-Gorriz and Carr (2001). Also, localized patterns of relatively high primary production were found in persistent deep water density fronts resulting from the

interaction of MAW and Mediterranean water by Lohrenz et al. (1988).

## 5 Conclusions

In summary, pelagic PP in coastal shelves of the Mediterranean Sea during the period 2002-2016 was estimated in this study for the first time using available satellite ocean colour product. We estimated that 12 % of PP of the Mediterranean Sea is attributable to coastal pelagic production and from that, about 80% of this carbon fixation is sustained by regenerated pathways.

High PP spatial variations were observed among the different regions, as mainly driven by major river effluents. Our analysis also reveals a weak negative PP trend, which cannot be qualified as climate-driven because most of the temporal variability is dominated by interannual or sub-decadal variations and the satellite record is only 14-year long. Finally, we identify 18 along-shelf zones based on their temporal PP patterns. Two main PP groups were observed: zones with strong cross-shore gradients, typically found in wider estuarine regions and homogeneous zones within narrow continental shelf areas.

**Competing interests**

The authors declare that they have no conflict of interest.

**Author contribution**

AR and GB designed the study in collaboration with DA. PMS and DA conducted most of the analyses. PMS wrote the manuscript, with substantial contributions from all co-authors.



**Acknowledgments**

We are grateful to National Aeronautics and Space Administration, NASA (https://oceancolour.gsfc.nasa.gov/) and EU Copernicus Marine Environment Monitoring Service, CMEMS (http://marine.copernicus.eu/) for the freely available ocean-color remotely-sensed data. Climate indices were obtained from the Climate Research Unit at the University of East Anglia (https://crudata.uea.ac.uk/cru/data/).

**Financial support.**

This article is a result of the Ministry of Economy and Competitiveness (MINECO) of Spain Project Fine-scale structure of cross-shore GRADIENTS along the Mediterranean coast (CTM2012-39476) and SifoMED (CTM2017-83774-P). P.M. Salgado-Hernanz, was supported by a Ph.D. Doctoral research fellowship FPI (*Formación Personal Investigación*) fellowship BES-2013-067305 from MINECO.

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
