# Peer review of "Pelagic primary production in the coastal Mediterranean Sea: variability, trends and contribution to basin scale budgets"

_Biogeosciences, 2020_

## Referee Comment (RC1)

**Review of "Pelagic primary production in the coastal Mediterranean Sea: variability, trends and contribution to basin scale budgets" by Salgado-Hernanz et al.**

**Detailed comments**

Title: I suggest removing pelagic as I currently feel the title is an oxymoron. I don't consider areas>5m deep to be pelagic?

Line 76: 'Coastal areas were generally ignored in such studies'. Following my statement above, both Bricaud et al. and Bosc et al. masked areas of high turbidity where data is uncertain. Please comment on what improvements have been made to the CMEMS data to make it relevant in this study (if improvements have indeed been made).

Line 152: How does this assumption impact your results? Are waters in the Mediterranean well mixed to 200m deep?

Line 224: As already mentioned, considering that some of these authors exclude the productive areas that you are including is this a fair comparison? The analysis would be a lot stronger if the same dataset was used to compare coastal production vs total production in the Mediterranean

Equation 1: Considering you assign a uniform chlorophyll concentration it is not really dependant on depth?

Line 236-238: The authors mention that the Eastern Mediterranean has twice the amount of coastal primar productivity than the western basin due to its size. However, primary productivity per unit volume is also twice the amount of western shelf and is also higher than that observed in the Adriatic? Why is this? This is not what I would expect, especially given the little river inputs along the coast of the Eastern Mediterranean,

Line 241-242: What about the Nile delta and Gulf of Gabes – these stand out to me as high areas of primary production based on Figure 1.

Table 1: What are the uncertainties? Standard deviation? Please state this in the caption

Table1: Why did you use a different product to estimate chlorophyll in the whole Mediterranean Sea or open ocean water rather than the same one as coastal waters? Why couldn't you also estimate primary productivity using the whole dataset? Then it is a coherent analysis and you are comparing like for like. It would then enable comparison of the coastal ocean vs the entire Med Sea in the temporal trend analysis too.

Figure2/3. What is the difference between Fig 2c and fig 3b?

Figure 5/Line 303: The authors say there is no significant trend in primary productivity in the Adriatic based on Figure 4. Why then does the Adriatic actually show the largest trend in Figure 5 with almost the entire 'coastal' Adriatic showing a positive trend? Likewise I can't really see any trends in the Western basin despite the authors saying there was a slight significant negative trend in the Western basin based on Figure4.

Figure 7: Are the alongshore (Z areas) also based on the temporal patterns as indicated by the main caption to the figure?

Line 358: The authors suggest enhanced production occurs in regions of freshwater influence. I would argue R7 is not. What other factors lead to high R7? Possibly domestic and industrial wastewater inputs?

Line 369-372: Interestingly Macias et al. (2018) use model simulations to show that primary production in the coastal region of the Western basin (including Gulf of Lions) is mostly influenced by circulation patterns, not river inputs . I suggest the authors include this reference somewhere in this manuscript.

Line 400: But the eastern Mediterranean also had higher values m3 that the western basin so it is not purely due to the bigger surface are of the eastern basin?

Lines 400-410: What about the influence of wastewater inputs (Powley et al., 2016) and submarine groundwater discharge (Rodellas et al., 2015)? It is mentioned again later in the discussion but I think it should be introduced earlier.

Line 431: What method did Barale et al., use? Is this also from satellites?

Line 443-445: Are you referring to the Biomodal Osciallation System (BIOS; i.e., Civitarese et al. 2010) here? If yes, I suggest you refer to it explicitly.

Lines 475-485: What about domestic and industrial wastewater inputs into the sea? Powley et al. (2016) show they may be significant and certainly are likely to contribute to primary production in some areas of the Mediterranean coastline.

Table 4: Please state how the errors are calculated.

Figure 8: The figure caption and figure do not seem to match to me. There appears to be nothing about seasonality in the figure

Figure 9: What unit is annual PP in? Does it make a difference if you use m-2 vs m-3 vs total?

Line 546-547– "Our analysis also reveals a weak negative PP trend which cannot be classed as climate driven" – but on lines 456 you say *"we observed an influence of climate scale variability on coastal productivity as suggested by the inverse correlations between ΣPP and SST and, more loosely, with NAO and MO?"* I don't agree/understand this conclusion based upon what is mentioned in the discussion

Conclusion: It would be nice if the authors could speculate how a dataset like this could be useful to the Mediterranean/scientific community. For example, could it be used to highlight coastal areas where additional monitoring should take place (Note the authors don't have to use this particular example)

**Minor edits**

Line 84: rather than basin scale budgets I suggest the authors be specific and either say basin scale PP or basin scale carbon fixation.

Line 104: 'whenever they exceeded about 3-times the mean'. Using "about" in this sentence makes it seem not very precise. Do you really mean to include this here?

Line 131: when you say day length do you mean hours of daylight?

Line 170 : For clarity I suggest adding coastline before Western, Eastern and Adriatic.

Line 240 Gulf of Sirte – I suggest if places are mentioned , they are included in the map in figure 1.

Line 241: add north before western African

Table 2: Suggest using 'Mediterranean' rather than 'Global' .

Line 316: Please rephrase as I don't understand what you are trying to say,

Lines 318-323: Are these results shown anywhere: Perhaps they can be included in supplementary material?

Line 550: MAW – This acronym is not defined in the text so please use full term.

Line 610 Bricaud reference – please provide full reference/link that works

Figures: I suggest to avoid using the rainbow colour scheme as it can emphasize unrealistic patterns.

**References**

Civitarese, G., Gačić, M., Lipizer, M., & Eusebi Borzelli, G. L. (2010). On the impact of the Bimodal Oscillating System (BiOS) on the biogeochemistry and biology of the Adriatic and Ionian Seas (Eastern Mediterranean). *Biogeosciences*, *7*(12), 3987–3997. https://doi.org/10.5194/bg-7-3987-2010

Macias, D., Garcia-Gorriz, E., & Stips, A. (2018). Major fertilization sources and mechanisms for Mediterranean Sea coastal ecosystems. *Limnology and Oceanography*, *63*(2), 897–914. https://doi.org/10.1002/lno.10677

Powley, H. R., Dürr, H. H., Lima, A. T., Krom, M. D., & Van Cappellen, P. (2016). Direct Discharges of Domestic Wastewater are a Major Source of Phosphorus and Nitrogen to the Mediterranean Sea. *Environmental Science & Technology*, *50*(16), 8722–8730. https://doi.org/10.1021/acs.est.6b01742

Rodellas, V., Garcia-Orellana, J., Masque, P., Feldman, M., & Weinstein, Y. (2015). Submarine groundwater discharge as a major source of nutrients to the Mediterranean Sea. *Proceedings of the National Academy of Sciences of the United States of America*, *112*(13), 3926–3930. https://doi.org/10.1073/pnas.1419049112

---

## Referee Comment (RC2)

Review of the manuscript "Pelagic primary production in the coastal Mediterranean Sea: variability, trends and contribution to basin scale budgets" by Paula Maria Salgado-Hernanz et al.

Authors present an analysis of the spatial and temporal variability of primary production in the coastal areas of the Mediterranean Sea. The analysis of the marine coastal areas is often neglected in experimental and modelling studies due to the inherent complexity of the processes. Therefore this work is extremely important since coastal areas are the most impacted and most impacting on anthropogenic activity. The scope of the manuscript is well presented and the methodology is correctly described. The analyses performed support the conclusions presented.

I suggest publication of the present manuscript after minor revisions detailed below.

**Minor comments**:

1. PG 2 Line 41 missing full stop: 2007). The productivity

2. PG 5 line 125 I would rephrase "Note that neither Chl, a* and \phi are made variable with time." With " Note that Chl, a* and \phi are considered time independent parameters."

3. PG 5 In Equation 4, in order to compute light attenuation is it necessary to consider the normalization on cosines to account for Solar Declination ?

4. PG 5 lines 148,149 the empirical formula, Morel (1991) and Morel et al. (1996), are valid also for coastal waters, the modelled primary production correspond to Gross Primary Production or Net Primary Production?

5. PG 6 lines 156,159 The studies by Laws 2000,2011 to derive *ef-ratio* are calibrated on open ocean conditions, could Authors comments on the applicability of such empirical relations in the coastal areas?

6. PG 6 lines 169,170 "We report annual PP estimates (Gt C) for the entire Mediterranean coastal areas (ΣPPcoast) and separately for the Western, Eastern and Adriatic basins (ΣPPbasin)." Here Authors mean Western and Eastern coastal basins or open ocean Basins?

7. PG 7 Table 1: The total values of PP for the Mediterranean Sea are obtained combining literature data for the open ocean water summed to the coastal estimates derived in this manuscript? Please explain.

8. PG11: Does Figure 3 show surface PP values or vertically averaged values or they coincide because chlorophyll vertical distribution is constant?

9. PG12 Figure 4 In the caption I would specify "whole Mediterranean **coast**, b) western **coast** basin, c) eastern **coast** basin" otherwise it can be confusing.

10. PG 13 lines 315,316"A significant negative correlation was observed between coastal ΣPP and SST (r=-0.63, p< 0.001; Fig. 6a) showing that the important decrease

of Chl over the years was able to compensate the effect of temperature increase."
Could authors elaborate a bit more the expected correlation between ΣPP, SST and
Chl and the corresponding compensation?

11. PG 14 Figure 6: It would be nice to see also the chlorophyll trend and how it
    correlates with SST, NAO and MOI.

12. Pg 18 lines 391-392 "Indeed, Case-1 waters are largely predominant in the coastal
    Mediterranean regions whereas Case-2 waters are reduced to less than 5% of the
    whole basin." The 5% is related to the coastal basin or to the total Basin? It would be
    important to report the Case-2 water fraction of the coastal basin to evaluate the
    relative importance.

13. Pg 20 Lines 439,441" While negative tendencies seem to fit with the assumed model
    of PP limitation associated with increasing temperatures, the origin of the positive
    trend in the Adriatic basin is more uncertain". Also chlorophyll exhibits a reduction
    starting from 2012 and  being an independent variable it could be the responsible, or
    a concurring responsible, for such trend.

14. Pg 22 line 511. "Our data does not display a general relationship between shelf
    width (Q) and PPannual" from this sentence it seems that Q is the symbol to indicate
    shelf width instead in figure 9 Q refers to river discharge.

15. Pg 22 Figure 9. The bubble are a bit superimposed and it is not easy to understand
    what's going on especially near the origin axis. Would it be possible to use a color
    bar with fixed size bubbles to reduce overlapping, use a log scale for x and y axis, or
    to arrange the plot to increase readability

---

## Referee Comment (RC3)

The manuscript by Salgado-Hernanz et al. describes the costal primary production of the Mediterranean Sea, estimated using satellite data, deepening on its spatial and temporal distribution and analyzing trends and possible link with NAO and/or MOI indexes. In general, I consider primary production an interesting topic, especially in the Mediterranean Sea, but in this paper is particularly interesting since it is analyzed in those areas highly impacted by anthropogenic pressure.

The paper is well organized, quite clear and English is good.

I admit I was amazed to observe that the primary production of the eastern compart of the Mediterranean Sea is higher than the western compart and to the Adriatic Sea. This result it is not only linked to the greater surface of the eastern sub-basin since it is also evident in the productivity per unit of volume. This result should be analyzed in depth by the authors.

About the trends estimated in the paper, I found some inconsistencies and I think the authors should better explain their observations. In general, it should be better defined how the quantities are calculated (i.e. $\Sigma PP$, $PP_{annual}$, $PPVOL_{annual}$, $CV_{PP}$, etc.) in such a way as to put the reader in the best conditions to understand the obtained results.

Once these issues I underlined are solved, I think the paper could be considered for the publication.

**Other Comments**

Lines 89-102: Are satellite data (chl, sst etc.) used at daily frequency? In the paper the authors cited monthly or 8days means but in this paragraph there is not any reference. Could you clarify?

Line 106: "Only values at depths exceeding 5 m depth were considered…" is there any reference for this assumption? As you know the layer that could influence remote sensing measurements depends on the sea water bio-chemical conditions. Based on my experience I believe that satellite measurements could be influenced by the bottom seagrass also for depths greater than 5 m. Maybe the authors could investigate, in some way, in order to give to the reader an idea of how much final results could be influenced by this issue.

Line 130: Is PP estimated on daily satellite images? Graphics and images in the paper show monthly data. Could the authors describe the exact technique used? Did they average input satellite data (i.e. CHL, sst, PAR etc.) and then compute PP? Or did they computed PP on daily satellite data and then averaged PP data?

Line 152-155: this point could also be investigated analyzing the mixing layer depth of the study area. Probably the chlorophyll uniform profile assumption may not be wrong in many cases, but having an idea of where this assumption is wrong could help to better understand the results of the study.

Line 169-170: Please define exactly how you computed annual PP ($\Sigma PP$). Afterwards, in the text, the authors analyze the results for $\Sigma PP$, $PP_{annual}$, $PPVOL_{annual}$, etc. but I cannot find any definition of these parameters. I think it is crucial to define exactly the quantities used in the analysis.

Line 179-180: has this alongshore regionalization been done with SOM (or other technique), or has it been done by the authors observing the results of the SOM regionalization?

Line 217-218: How did the authors deseasonalize the data?

Line 226: Is annual carbon fixation per surface area the PP daily average multiplied by 365? If no, how is it estimated?

Line 237-239: from tab 1, annual carbon fixation is quite similar between eastern and western sub-basins. Since the area of east shelf is about twice of west shelf, it is obvious that the eastern annual integrated PP is approximately double of the west sub-basin. On the other hand, it is not absolutely obvious why the "productivity per unit volume" of eastern compart is more than double of the western one (and even higher than the Adriatic Sea). I'm a little bit surprised…

Line 239-241: I suggest starting the sentence by citing the figure you are referring to instead of citing it at the end.

Line 243. "…the coefficient of variation of primary production ($CV_{PP}$)…" how did you calculate this coefficient? Defining this "coefficient of variation of primary production" would also clarify why there are 2 different "coefficient of variation of primary production", one in figure 2c and another in figure 3b.

Tab 1: Could you please insert the exact reference (product ID as for satellite daily data described above) for the 8-days e 4km resolution data taken from CMEMS? I cannot find them.
Why is *** only for "Mediterranean Sea"? shouldn't it also be on "Open ocean waters"?

Line 290-291: How did you estimate this interannual variability?

Line 298-299: How did you calculate the "the filtered low frequency signal" for PP and CHL?

Line 306-307: "Most of these regions presented declining PP trends…". This sentence does not seem so evident observing fig 5a. The only evident negative trend is in the Gulf of Gabes as underlined by the authors. Moreover I believe that fig 5a and graphs in fig4 are quite inconsistent. In fig 4 trends are negative for Mediterranean Sea, west and east sub-basin, while for Adriatic Sea there is no evident trend. From fig 5a I'd say that on average Mediterranean Sea trend is quite positive (red areas are greater than blue ones). For west and east sub-basin the negative trend shown in fig 4a is not so evident, especially for the eastern compart. About Adriatic Sea fig 5a shows a clear positive trend. Could the authors explain this apparently discrepancy and how a reader should interpret it?

Fig 6: line 324: it is not specified (here or in the text) what blue lines meaning. Are they annual PP anomalies?

Line 327-336: R1 to R9 are represented in fig 7 as PP in gC m$^{-2}$ d$^{-1}$, but in tab 3 there are mean annual PP values. Again, it should be defined how you estimate mean annual PP starting from daily PP.

Line 392: "…whereas Case-2 waters are reduced to less than 5% of the whole basin.". Is there any reference for this statement?

Fig 8: Caption of the figure refers to another type of figure (seasonal PP). Fig 8a has a different color palette (and range) with respect to other 2 (b and c) images.

---

## Author Comment (AC1)

**Manuscript Reference No**.: bg-2020-457
**Biogeosciences Discuss**., https://doi.org/10.5194/bg-2020-457-RC1, 2021
**Title:** "Pelagic primary production in the coastal Mediterranean Sea: variability, trends and contribution to basin scale budgets" by Paula Maria Salgado-Hernanz et al. 2021.

==============================

**Response to Interactive Discussion Referee #1. 1ˢᵗrevision (01-02-2021):**

Dear reviewer, we would like to thank you the interest you have shown in our study and therefore consider it as suitable for publication in Biogeosciences. We then proceed to disaggregate and answer your comments:

**Response to main comments Referee #1:**

**#1.1:** One of the things that struck me was the higher primary productivity (per $m^3$) and chlorophyll data in the Eastern basin than Western basin which I found very surprising. Due to low river inputs in the Eastern Mediterranean compared to the Western Mediterranean I naturally would expect the Western Mediterranean coastal area to be more productive. I would like the authors to discuss this in more detail – is this due to the uncertainty/overestimation of chlorophyll in the Gulf of Gabes as mentioned by the authors or is there observational data to back up the high productivity here.

We agree. Averages are in this case misleading because distributions are not normal. In some shallow and highly productive regions this is particularly notable. In the case of PP (gC $m^{-2}$, see herein Fig. 1a) this compensated by the integration depth and, thus, the weight of these pixel, although relevant, is less critical than in PP (gC $m^{-3}$, see herein Fig. 1b) where the influence of vertically averaging just over few surface values exacerbates the differences with overall values. As shown in herein Fig. 1b, due to the lack of large shallow and productive areas, there are few values above 30 (gC $m^3$) in the western Mediterranean, whereas high PP is more frequent in the Adriatic Sea (red) and in the eastern Mediterranean (blue). If the pixels with values >30 gC $m^3$ are plotted (see next Fig. 2) it becomes evident that most of them are located in shallow waters of the Gulf of Gabes and in the Nile Delta, and less so, in the northern Adriatic. To avoid these problems, we refer now to median values in Table 2, yet mean values are still provided as a reference.

[Figure]

Figure 1. Frequency histograms for a) Integrated and b) vertically averaged PP estimations. Blue (East Med.) green (West Med), red (Adriatic). Note that Y-axis in Fig 1b is logarithmic.

[Figure]

Figure 2. Map showing the location of the pixels with values >30 gC m$^{-3}$ in blue.

**#1.2:** Can the authors put any error estimates on this or give a lower bound on the Eastern value.

We have maintained de standard deviation (S.D.) instead of the standard error (S.E.) because most published PP use this deviation descriptor in their studies.

**#1.3:** Likewise, can you really give a contribution of total primary production to the Mediterranean for coastal areas if the studies that estimated these total Mediterranean values exclude the highly productive coastal areas (i.e. North Adriatic, Gulf of Gabes) as mentioned by the authors on Line 76 due to the high turbidity and thus inaccurate values.

We use an improved Chl algorithm that is regionally tuned to consider the characteristics of the different Mediterranean regions. Certainly, this not exempt from inaccuracies due to turbidity, but it is much more accurate than previous algorithms used for PP estimations. See, also the following question #1.4:

***#1.4:*** Alternatively is the CMEMS chlorophyll data corrected for these high turbidity areas, reducing the uncertainty in your estimates compared to previous studies like Bosc et al. 2004 and Bricaud et al., 2002 where these areas were excluded?

In the present study, we used the most adequate regional Chl product available up to date for the Mediterranean Sea. This ocean-colour data record is a regionally-tuned reprocessing of the climate-quality, error-characterized, and bias-corrected merged product of multi satellite observations (SeaWiFS, MODIS-Aqua and MERIS sensors) initially developed by the European Space Agency Ocean- Colour Climate Change Initiative Program (ESA OC-CCI) (Sathyendranath et al., 2017; Sathyendranath and Krasemann, 2014). Then, the Chl product available from CMEMS has been tailored to the Mediterranean region by using the regional algorithm MedOC4 (Mediterranean Ocean-Colour 4 bands, Volpe et al., 2007) for Case-1 waters and the AD4 algorithm (ADriatic 4 band, Berthon and Zibordi, 2004; D'Alimonte and Zibordi, 2003) for Case-2 waters. In this product, the merging of Case-1 and Case-2 information was performed following D'Alimonte et al. (2003). In practice, the CMEMS processor ingests the OC-CCI remote sensing reflectance, which is the result of a merging procedure that accounts for the inter-sensor bias among different sensors and then applies the specific regional algorithm. In the studies mentioned by the reviewer, no specific regional Chl algorithm was used. Both Bricaud et al. (2002) and Bosc et al. (2004) used Chl resulting from reprocessing #4, provided in July 2002 (see http://seawifs.gsfc.nasa.gov/SEAWIFS/RECAL/Repro4). The bio-optical algorithm was the ''OC4v4''algorithm proposed by O'Reilly et al. (1998).

***#1.5:*** Following on from this I would like to ask the authors whether they have considered doing the analysis (with small adjustments) for the whole Mediterranean Sea so that comparison for coastal primary against the whole Mediterranean is coherent using data that has been prepared in the same way. This would enhance their conclusions on the contribution of the coastal zone to primary productivity in the Mediterranean.

The focus of the present study is the coastal zone of the Mediterranean Sea at a reasonable resolution to be able to identify the main coastal PP features and with the aim of defining different coastal regions that are oversaw in more general PP estimations. Therefore, we decided to exclude open ocean waters. We agree that running the entire Mediterranean Sea would be more coherent but, since there are plenty of studies providing this information (we do in fact review all of them in Table 2), we do not feel that this is a major drawback.

***#1.6:*** Generally, the manuscript is well written and English is good. I do feel that the conclusions can be strengthened and it would be nice if the authors could specifically say how this dataset/analysis will be useful to the Mediterranean science community. If the authors address the comments I have made, I think this manuscript can be considered for publication in Biogeosciences. The attached supplement provides my detailed comments on the manuscript

We are thankful for the positive comment of the reviewer. We have now enriched the Conclusion section with several sentences explaining the importance of understanding coastal production and its long-term variability in the Mediterranean Sea.  It now reads "*In summary, pelagic PP in coastal shelves of the Mediterranean Sea during the period 2002-2016 was estimated in this study for the first time using available satellite ocean colour product. We estimated that 12% of PP of the Mediterranean Sea is attributable to coastal pelagic production and from that, about 80% of*

*this carbon fixation is sustained by regenerated pathways. High PP spatial variations were observed among the different regions, as mainly driven by major river effluents, exchanges with nearby seas (i.e. Black Sea and the Atlantic Ocean) and by local processes. Our study shows that some coastal areas are indeed highly productive (>400 g C m-2) and sustain a large percentage of overall coastal production. Indeed, their temporal variability could be of paramount importance to understand variations in higher trophic levels (e.g. Piroddi et al., 2017). Despite that temporal variability is dominated by interannual and sub-decadal variations, our analysis reveals a weak global negative PP trend in the Mediterranean Sea related to climate drive patters (i.e temperature increase).Our analysis also reveals a weak negative PP trend, which cannot be qualified as climate-driven because most of the temporal variability is dominated by interannual or sub-decadal variations and the satellite record is only 14-year long. Nevertheless, long-terms effects can be regionally variable (i.e. PP trends in the Adriatic Sea are positive) and variations inof decreases fluvial nutrient inputs, together with other processes such as ocean warming in coastal regions, including heat waves, deserve a closer look as longer ocean colour database becomes available. Finally, we identify 18 along-shelf zones based on their temporal PP patterns. Two main PP groups were observed: zones with strong cross-shore gradients, typically found in wider estuarine regions and homogeneous zones within narrow continental shelf areas. These two types of coastal waters clearly characterize the coastal area of a sea were coastal waters are otherwise strongly influenced by ocean conditions*".

**Response to attached detailed comments from Reviewer # 1:**

*#1.7:* Title: I suggest removing pelagic as I currently feel the title is an oxymoron. I don't consider areas>5m deep to be pelagic?

We intended to clarify that we are not estimating the contribution of the benthic PP in the coast. We agree that, in some contexts, the term may be confusing in this context since pelagic often refers to open waters. Nevertheless, pelagic is also commonly used as opposed to benthic in coastal studies (ie, pelagic, bentho-pelagic and benthic fish classification). Some other authors (i.e. Macias et al., 2017) also use the term pelagic as opposed to benthic in coastal PP studies.

*#1.8:* Line 76: 'Coastal areas were generally ignored in such studies'. Following my statement above, both Bricaud et al. and Bosc et al. masked areas of high turbidity where data is uncertain. Please comment on what improvements have been made to the CMEMS data to make it relevant in this study (if improvements have indeed been made).

Answer to this comment has been provided in the response to main comment *#1.4.*

*#1.9:* Line 152: How does this assumption impact your results? Are waters in the Mediterranean well mixed to 200m deep?

Given the variability in coastal waters, we consider that this is a better assumption that using global parameterization of the shape of the vertical profile as a function of the surface Chl obtained from deep open ocean waters. Assuming a homogeneous profile will bias the calculations in offshore boundary of the shelf, particularly during summer conditions when the contribution of the DCM is not accounted for. We state this now in the M&M section.

***#1.10:*** Line 224: As already mentioned, considering that some of these authors exclude the productive areas that you are including is this a fair comparison? The analysis would be a lot stronger if the same dataset was used to compare coastal production vs total production in the Mediterranean.

Answer to this comment is addressed in comment ***#1.5.***

***#1.11:*** Equation 1: Considering you assign a uniform chlorophyll concentration it is not really dependent on depth?

Yes, the chlorophyll concentration is uniform with depth. Irradiance, hence PP, are however varying with depth by virtue of how light propagates in the water column

***#1.12:*** Line 236-238: The authors mention that the Eastern Mediterranean has twice the amount of coastal primary productivity than the western basin due to its size. However, primary productivity per unit volume is also twice the amount of western shelf and is also higher than that observed in the Adriatic? Why is this? This is not what I would expect, especially given the little river inputs along the coast of the Eastern Mediterranean.

As we explained in ***#1.1***, mean primary production per unit volume exaggerates the production in shallow areas. We consider that mean values are highly affected by the production in these areas and, therefore, we refer to median values which reveal that coastal median PP per unit volume is 16% lower in the Eastern that in the western Mediterranean. Please, refer to Table 1 for the updated values.

***#1.13:*** Line 241-242: What about the Nile delta and Gulf of Gabes – these stand out to me as high areas of primary production based on Figure 1.

The reviewer is right. It now reads '*However, in some coastal regions of the eastern basin like the Gulf of Gabes and the Nile Estuary primary production is outstandingly high (>300 g C m$^{-2}$)*'

***#1.14:*** Table 1: What are the uncertainties? Standard deviation? Please state this in the caption

Yes, the uncertainties indicate the Standard Deviation (S.D). It is now specified in the caption.

***#1.15:*** Table1: Why did you use a different product to estimate chlorophyll in the whole Mediterranean Sea or open ocean water rather than the same one as coastal waters? Why couldn't you also estimate primary productivity using the whole dataset? Then it is a coherent analysis and you are comparing like for like. It would then enable comparison of the coastal ocean vs the entire Med Sea in the temporal trend analysis too.

A detailed answer to this comment is addressed in comment ***#1.5.***

***#1.16:*** Figure2/3. What is the difference between Fig 2c and Fig 3b?

Figure 2c shows the coefficient of variation (CV) of the mean primary production per unit area, in g C m$^{-2}$, of the surface waters. Figure 3b shows the CV of the mean productivity per unit volume, in g C m$^{-3}$.

***#1.17:*** Figure 5/Line 303: The authors say there is no significant trend in primary productivity in the Adriatic based on Figure 4. Why then does the Adriatic actually show the largest trend in Figure 5 with almost the entire 'coastal' Adriatic showing a positive trend? Likewise, I can't really see any trends in the Western basin despite the authors saying there was a slight significant negative trend in the Western basin based on Figure4.

The reviewer is right and Figure 4 and Figure 5 could bring misunderstanding. Figure 4 showed a regional trend resulting from 15 points (one mean value per year). Moreover, from 2012 a reduction in PP is shown at every region with the exception of the Adriatic. The Adriatic region presented positive PP values for years 2013 and 2014 (see Supplementary figure 1) and this could change the PP trend when only 15 points are considered (i.e Figure 4). For that reason, now we only provide trends obtained calculated with the complete time series (as shown in fig 5).

***#1.18:*** Figure 7: Are the alongshore (Z areas) also based on the temporal patterns as indicated by the main caption to the figure?

Yes, that is correct. SOM aggregates the characteristic temporal patterns according to their similarities. In section 2.3 Coastal regionalization we quote, line 18-190 "*Then, 18 alongshore marine ecoregions were obtained considering the most relevant cross-shore limits of the SOM-derived regions (Z1 to Z18)."*

***#1.19:*** Line 358: The authors suggest enhanced production occurs in regions of freshwater influence. I would argue R7 is not. What other factors lead to high R7? Possibly domestic and industrial wastewater inputs?

We have rephrased this sentence. Now it reads, '*An exception is R7 pattern, which is exclusively located in the shallowest inner shelf of the Gulf of Gabes,..*' . In the discussion section, line 502, we suggest that the PP enrichment in that area may be associated with degradation of the water quality attributed to industrial and urban activities (Hamza-Chaffai et al., 1997; Zairi and Rouis, 1999).

***#1.20:*** Line 369-372: Interestingly Macias et al. (2018) use model simulations to show that primary production in the coastal region of the Western basin (including Gulf of Lions) is mostly influenced by circulation patterns, not river inputs. I suggest the authors include this reference somewhere in this manuscript.

We now make reference to the paper of Macias et al (2017) L&O in the discussion section 4.1. '*Mediterranean coastal production is also supported by other sources such as local mesoscale processes (Macias et al., 2017).*'

***#1.21:*** Line 400: But the eastern Mediterranean also had higher values m3 that the western basin so it is not purely due to the bigger surface are of the eastern basin?

The reviewer is right. It now reads *'Because of its extension, due to the increased productivity in regions like Gabes, the Nile and the northern Aegean Sea'*.

*#1.22:* Lines 400-410: What about the influence of wastewater inputs (Powley et al., 2016) and submarine groundwater discharge (Rodellas et al., 2015)? It is mentioned again later in the discussion but I think it should be introduced earlier.

These references have been added. We have also emphasized the importance of groundwater and nutrient-rich effluents from human activities in the 4$^{th}$ paragraph of the introduction.

*#1.23:* Line 431: What method did Barale et al., use? Is this also from satellites?

The sentence has been rephrase by "*Barale et al. (2008), using Chl anomalies derived from SeaWiFS data, observed a general decrease in Chl biomass in the Mediterranean Sea over the period 1998–2003*".

*#1.24:* Line 443-445: Are you referring to the Biomodal Osciallation System (BIOS; i.e., Civitarese et al. 2010) here? If yes, I suggest you refer to it explicitly.

We have added explicit reference to the BIOS. Now it reads' *Alternatively, the Bimodal Oscillating System i.e. the feedback mechanism between the Adriatic and Ionian (Civitarese et al., 2010) peaking between 2004 and 2006 could have affected mass and nutrient exchanges between the Adriatic and the north Ionian.*'

*#1.25:* Lines 475-485: What about domestic and industrial wastewater inputs into the sea? Powley et al. (2016) show they may be significant and certainly are likely to contribute to primary production in some areas of the Mediterranean coastline.

We have included an explicit reference to domestic wastewater (Powley et al. 2016) in this paragraph now.

*#1.26:* Table 4: Please state how the errors are calculated.

The caption now reads '*Mean and standard deviation (S.D.) are calculated from 14 year averages is calculated from 15-year averages (2002-2016)*'.

*#1.27:* Figure 8: The figure caption and figure do not seem to match to me. There appears to be nothing about seasonality in the figure

There was an error in the caption. The correct one is now included.

*#1.28:* Figure 9: What unit is annual PP in? Does it make a difference if you use m-2 vs m-3 vs total?

g C m$^{-2}$. It is indicated now in the figure caption.

*#1.29:* Line 546-547– "Our analysis also reveals a weak negative PP trend which cannot be classed as climate driven" – but on lines 456 you say "we observed an influence of climate scale variability on coastal productivity as suggested by the inverse correlations between ΣPP and SST and, more loosely, with NAO and MO?" I don't agree/understand this conclusion based upon

Note: Line 546-547 refers to actual lines 579-580 in the conclusion section while line 456, it refers to actual line 491 in the discussion 4.2 section. We thank the referee for this inconsistency. It has been corrected.

*#1.30:* Conclusion: It would be nice if the authors could speculate how a dataset like this could be useful to the Mediterranean/scientific community. For example, could it be used to highlight coastal areas where additional monitoring should take place (Note the authors don't have to use this particular example)

The detailed answer to this comment is also addressed in comment *#1.6.* We have extended the conclusion section.

**Response to attached minor edits from Reviewer # 1:**

*#1.31:* Line 84: rather than basin scale budgets I suggest the authors be specific and either say basin scale PP or basin scale carbon fixation.

We have changed it accordingly.

*#1.32:* Line 104: 'whenever they exceeded about 3-times the mean'. Using "about" in this sentence makes it seem not very precise. Do you really mean to include this here?

We agree. The term "about" has been deleted. Thank you for the advice.

*#1.33:* Line 131: when you say day length do you mean hours of daylight?

It has been specified: "D is the day length or hours of daylight (h)".

*#1.34:* Line 170 : For clarity I suggest adding coastline before Western, Eastern and Adriatic.

Done.

*#1.35:* Line 240 Gulf of Sirte – I suggest if places are mentioned, they are included in the map in figure 1.

The location of Gulf of Sirte has been added in Figure 1.

*#1.36:* Line 241: add north before western African.

Done.

***#1.37:*** Table 2: Suggest using 'Mediterranean' rather than 'Global' .

Done.

***#1.38:*** Line 316: Please rephrase as I don't understand what you are trying to say,

We agree. It has been rephrased to '*A significant negative correlation was observed between coastal ΣPP and SST (r=-0.63, p< 0.001; Fig. 6a) revealing a decrease in phytoplankton biomass as the sea warms up*'.

***#1.39:*** Lines 318-323: Are these results shown anywhere: Perhaps they can be included in supplementary material?

They are not included in the manuscript. We run the analysis and in the manuscript we mention the results that were more relevant for the discussion.

***#1.40:*** Line 550: MAW – This acronym is not defined in the text so please use full term.

Thank you. It has been added "Modified Atlantic Water (MAW)".

***#1.41:*** Line 610 Bricaud reference – please provide full reference/link that works

Corrected.

***#1.42:*** Figures: I suggest to avoid using the rainbow colour scheme as it can emphasize unrealistic patterns.

Thank you for the advice. The color has been changed.

**References**

Barale, V., Jaquet, J. M. and Ndiaye, M.: Algal blooming patterns and anomalies in the Mediterranean Sea as derived from the SeaWiFS data set (1998-2003), Remote Sens. Environ., 112(8), 3300–3313, doi:10.1016/j.rse.2007.10.014, 2008.

Macias, D., Garcia-Gorriz, E. and Stips, A.: Major fertilization sources and mechanisms for Mediterranean Sea coastal ecosystems, Limnol. Oceanogr., 63(2), 897–914, doi:10.1002/lno.10677, 2017.

---

## Author Comment (AC2)

**Manuscript Reference No**.: bg-2020-457
**Biogeosciences Discuss**., https://doi.org/10.5194/bg-2020-457-RC1, 2021
**Title:** "Pelagic primary production in the coastal Mediterranean Sea: variability, trends and contribution to basin scale budgets" by Paula Maria Salgado-Hernanz et al. 2021.

============================

**Response to Interactive Discussion Referee #2. 1ˢᵗrevision (16-02-2021):**

Dear referee, we would like to thank you the interest you have shown in our study and we appreciate the point you made about the importance of studying coastal areas. We thank you for consider our manuscript as suitable for publication in Biogeosciences with minor changes. We then proceed to reply to your comments:

**Response to supplement minor comments from Referee #2:**

*#2.1:* PG 2 Line 41 missing full stop: 2007). The productivity.

Corrected.

*#2.2:* PG 5 line 125 I would rephrase "*Note that neither Chl, a\* and \phi are made variable with time.*" With " Note that Chl, a\* and \phi are considered time independent parameters."

It has been changed accordingly. We appreciate the reviewer's suggestion.

*#2.3:* PG 5 In Equation 4, in order to compute light attenuation is it necessary to consider the normalization on cosines to account for Solar Declination?

The $\chi$ factor and the exponent "e" were derived from a large dataset of in situ measurements taken over a range of solar zenith angle. As such, the sun zenith angle is not explicitly considered in this formalism, yet it is taken into account on average.

*#2.4:* PG 5 lines 148,149 the empirical formula, Morel (1991) and Morel et al. (1996), are valid also for coastal waters, the modelled primary production corresponds to Gross Primary Production or Net Primary Production?

This is Net Primary Production (NPP).

*#2.5:* PG 6 lines 156,159 The studies by Laws 2000,2011 to derive *ef*-ratio are calibrated on open ocean conditions, could Authors comments on the applicability of such empirical relations in the coastal areas?

The temperature dependent *ef*-ratio is certainly an empirical approximation and a global algorithm that may present local biases. According to Laws et al, It provides a good approximation to experiments of N-15 labeled uptake, explaining 87% of the variance in the observed ratios. As shown in the original paper (Laws et al., 2000), values from highly different systems were

considered in this relationship. As pointed out by the reviewer, most of these regions were oceanic, but also included coastal upwelling areas such as Peru. Even though the algorithm is a coarse generalization of the relationship between total production, export production, and environmental variables (T) that may not be particular systems like river discharge regions we believe it yields reasonable results in the Mediterranean and provides a good approximation of export production.

***#2.6:*** PG 6 lines 169,170 "*We report annual PP estimates (Gt C) for the entire Mediterranean coastal areas (ΣPPcoast) and separately for the Western, Eastern and Adriatic basins (ΣPPbasin).*" Here Authors mean Western and Eastern coastal basins or open ocean Basins?

We refer to the coastal basins. It has been now specified in the manuscript.

***#2.7:*** PG 7 Table 1: The total values of PP for the Mediterranean Sea are obtained combining literature data for the open ocean water summed to the coastal estimates derived in this manuscript? Please explain.

Our coastal values are added to open ocean estimations to estimate total basin PP.

***#2.8:*** PG11: Does Figure 3 show surface PP values or vertically averaged values or they coincide because chlorophyll vertical distribution is constant?

Figure 3 shows the vertically average PP values registered in the water column. $PP_{VOL}$ is simply the integrated PP divided by whichever is shallower of the bottom depth or the productive layer.

***#2.9:*** PG12 Figure 4 In the caption I would specify "whole Mediterranean coast, b) western coast basin, c) eastern coast basin" otherwise it can be confusing.

Thank you for the suggestion. We decided to specify "whole Mediterranean Sea" in the figure caption. Now it reads '*Figure 4: PP variability and trends for coastal waters in a) the whole Mediterranean Sea, b) western basin, c) eastern basin and d) the Adriatic coast.*'

***#2.10:*** PG 13 lines 315,316"*A significant negative correlation was observed between coastal ΣPP and SST (r=-0.63, p< 0.001; Fig. 6a) showing that the important decrease of Chl over the years was able to compensate the effect of temperature increase.*" Could authors elaborate a bit more the expected correlation between ΣPP, SST and Chl and the corresponding compensation?

We agree this was confusing. The sentence has been rephrased to "*A significant negative correlation was observed between coastal ΣPP and SST (r=-0.63, p< 0.001; Fig. 6a) revealing a decrease in phytoplankton biomass as the sea warms up*".

***#2.11:*** PG 14 Figure 6: It would be nice to see also the chlorophyll trend and how it correlates with SST, NAO and MOI.

The figure with Chl provides similar results to the one provided with PP. This relationship has now been included in Supp. Figure 3.

***#2.12:*** Pg 18 lines 391-392 "*Indeed, Case-1 waters are largely predominant in the coastal Mediterranean regions whereas Case-2 waters are reduced to less than 5% of the whole basin.*" The 5% is related to the coastal basin or to the total Basin? It would be important to report the Case-2 water fraction of the coastal basin to evaluate the relative importance.

It refers to the whole basin. In order to clarify this point, the text now reads '*This constitutes some 23% of the coastal waters with prevalence in the north Adriatic Sea, Gulf of Gabes and around Nile delta*".

***#2.13:*** Pg 20 Lines 439,441 "While *negative tendencies seem to fit with the assumed model of PP limitation associated with increasing temperatures, the origin of the positive trend in the Adriatic basin is more uncertain*". Also chlorophyll exhibits a reduction starting from 2012 and being an independent variable it could be the responsible, or a concurring responsible, for such trend.

We appreciate your contribution. We add now it in the manuscript as a plausible explanation.

***#2.14:*** Pg 22 line 511. "*Our data does not display a general relationship between shelf width (Q) and PPannual*" from this sentence it seems that Q is the symbol to indicate shelf width instead in figure 9 Q refers to river discharge.

We agree. This error has been corrected. Figure caption now reads "*Our data does not display a general relationship between shelf width (Q) and PPannual (Table 4 and Fig. 9)*".

***#2.15:*** Pg 22 Figure 9. The bubble are a bit superimposed and it is not easy to understand what's going on especially near the origin axis. Would it be possible to use a color bar with fixed size bubbles to reduce overlapping, use a log scale for x and y axis, or to arrange the plot to increase readability.

Figure 9 has been changes following reviewer's suggestion.

---

## Author Comment (AC3)

**Manuscript Reference No**.: bg-2020-457
**Biogeosciences Discuss**., https://doi.org/10.5194/bg-2020-457-RC1, 2021
**Title:** "Pelagic primary production in the coastal Mediterranean Sea:
variability, trends and contribution to basin scale budgets" by Paula Maria Salgado-
Hernanz et al. 2021.

==============================

**Response to Interactive Discussion Referee #3. 1ˢᵗrevision (20-02-2021):**

Dear reviewer, we would like to thank you the interest you have shown in our study and therefore consider it as suitable for publication in Biogeosciences with some modifications. We then proceed to and answer your comments:

**Response to comments Referee #3:**

*#3.1:* Lines 89-102: Are satellite data (chl, sst etc.) used at daily frequency? In the paper the authors cited monthly or 8days means but in this paragraph there is not any reference. Could you clarify?

We recognize the text was misleading regarding the resolution of the different products. We have added further details about the remote sensing data and its processing methodology in section 2.1 Remote sensing data in the Materials and Methods section.

- Surface chlorophyll (**Chl,** in mg C $m^{-2}$) data, was downloaded for the specific Mediterranean Sea reprocessed Level-3 surface chlorophyll concentration 1-day and 1-km product obtained from the EU Copernicus Marine Environment Monitoring Service - CMEMS (OCEANCOLOUR_MED_CHL_L3_REP_OBSERVATIONS_009_073, (http://marine.copernicus.eu/services-portfolio/access-to-products/?option=com_csw&view=details&product_id=OCEANCOLOUR_MED_CHL_L3_REP_OBSERVATIONS_009_073 - CMEMS (copernicus.eu)). The specific dataset name was 'dataset-oc-med-chl-multi-l3-chl_1km_daily-rep-v02' and the variable used was 'mass_concentration_of_chlorophyll_a_in_sea_water (CHL)' in a NetCDF-4 Classic model file format.
- Sea Surface Temperature (**SST**, in ºC) data, we used the Level-2 SST at 1-day and 1-km from every available orbit from Moderate Resolution Imaging Spectroradiometer (MODIS) aboard satellites Terra and Aqua, downloaded from the National Aeronautics and Space Administration (NASA). Then, every orbit was averaged and projected onto the 1km² working grid (https://oceancolor.gsfc.nasa.gov/cgi/browse.pl?sub=level3&per=CU&day=18693&set=10&ndx=0&mon=18659&sen=amod&rad=0&frc=0&dnm=D@M&prm=SST).
- Photosynthetically Active Radiation (**PAR**, in E $m^{-2}$) was obtained as a Level 3 product at 9 km, the best available resolution from the NASA archive from both MODIS and Medium Resolution Imaging Spectrometer (MERIS). Data was downloaded by the browser (https://oceancolor.gsfc.nasa.gov/l3/).

All satellite-derived variables were remapped onto a **regular 1-km spatial grid** over the Mediterranean Sea area (-6 to 37º E et 30 to 46º N) by averaging all available pixels within each grid cell. This was made at daily resolution for the time period 2002-2016. This lead to a

considerable huge matrix of 529,226 grid points or pixels x 5,479 time steps. The time resolution resulted in 15 years * 365 days = 5475. As there were 4 leap years (2004, 2008, 2012, 2016), time resolution was 5479.

Further explanations regarding calculations performed to run the PP model are given in *#3.3:*

*#3.2:* Line 106: "*Only values at depths exceeding 5 m depth were considered…*" is there any reference for this assumption? As you know the layer that could influence remote sensing measurements depends on the sea water bio-chemical conditions. Based on my experience I believe that satellite measurements could be influenced by the bottom seagrass also for depths greater than 5 m. Maybe the authors could investigate, in some way, in order to give to the reader an idea of how much final results could be influenced by this issue.

We considered that 5m depth was a reasonable value to avoid interference by bottom seagrasses reflectance or extreme coastal regions. We recognize that some pixels at the bottom boundary may be affected by seafloor reflectance but, in most cases, this should only represent a small fraction of the 1x1 km pixel signal.

*#3.3:* Line 130: Is PP estimated on daily satellite images? Graphics and images in the paper show monthly data. Could the authors describe the exact technique used? Did they average input satellite data (i.e. CHL, sst, PAR etc.) and then compute PP? Or did they computed PP on daily satellite data and then averaged PP data?

We acknowledge that this was not clear. PP calculation was performed every 7 days (starting at day 4), using the averaged Chl for a 7-day window from day-3 to day+3 and for 1 grid point out of 3. Therefore, PP model was run with 8-days resolution and for 1 pixel out of 3 pixels. Later, PP data was interpolated to 1-day 1-km grid. In some cases -e.g. Fig 6-, monthly PP data and its anomalies were derived from the above mentioned dataset to be able to correlate PP values with the climatic indices NAO and MOI, and SST values.

*#3.4:* Line 152-155: this point could also be investigated analyzing the mixing layer depth of the study area. Probably the chlorophyll uniform profile assumption may not be wrong in many cases, but having an idea of where this assumption is wrong could help to better understand the results of the study.

Given the variability in coastal waters, we consider that this is a better assumption that using global parameterization of the shape of the vertical profile as a function of the surface Chl obtained from deep open ocean waters. Assuming a homogeneous profile will bias the calculations in offshore boundary of the shelf, particularly during summer conditions when the contribution of the DCM is not accounted for. We state this now in the M&M section.

*#3.5:* Line 169-170: Please define exactly how you computed annual PP (PP). Afterwards, in the text, the authors analyze the results for ΣPP, PPannual, PPVOLannual, etc. but I cannot find any definition of these parameters. I think it is crucial to define exactly the quantities used in the analysis.

A new Table 1 has been added clarifying the different definitions of PP that we use

**Table 1**. Primary production acronyms used in this study, their units and definitions.

| Variable | Units | Definition |
|---|---|---|
| P | mol C $m^{-3}$ $s^{-1}$ | Instantaneous production at each depth ($z$) of the water column. |
| PP | gC $m^{-2}$ $d^{-1}$ | Daily primary production per surface unit. Integration of P over depth and daylength. |
| $PP_{annual}$ | gC $m^{-2}$ | Annual mean production per surface unit. |
| ΣPP | GtC | Total carbon fixation per year within a basin or specific region. |
| $PP_{VOL}$ | gC $m^{-3}$ | Mean volumetric PP. Averaged form the surface to the bottom depth or productive layer |
| PPnew | gC $m^{-2}$ | New production (i.e from alochtonous sources) |
| PPreg | gC $m^{-2}$ | Regenerated production. |

***#3.6:*** Line 179-180: has this alongshore regionalization been done with SOM (or other technique), or has it been done by the authors observing the results of the SOM regionalization?

This alongshore regionalization in 18 zones (Z1-Z18) has been performed observing the results of the temporal patterns observed father the SOM regionalization (R1-R9). SOM aggregates the characteristic temporal patterns according to their similarities. In section 2.3 Coastal regionalization we quote, line 18-190 "*Then, 18 alongshore marine ecoregions were obtained considering the most relevant cross-shore limits of the SOM-derived regions (Z1 to Z18).*"

***#3.7:*** Line 217-218: How did the authors deseasonalize the data?

Time series were de-seasonalized by removing the 8-days means for the original 8-days time series. It has been added to the manuscript section 2.5. Statistical analysis as ". *Time series were de-seasonalized by removing the 8-days climatological mean for the original time series*".

***#3.8:*** Line 226: Is annual carbon fixation per surface area the PP daily average multiplied by 365? If no, how is it estimated?

That is correct. The original PP data was in day units, therefore we multiply it by 365 to obtain annual values.

***#3.9:*** Line 237-239: from tab 1, annual carbon fixation is quite similar between eastern and western subbasins. Since the area of east shelf is about twice of west shelf, it is obvious that the eastern annual integrated PP is approximately double of the west sub-basin. On the other hand, it is not absolutely obvious why the "productivity per unit volume" of eastern compart is more than double of the western one (and even higher than the Adriatic Sea). I'm a little bit surprised…

We agree. Averages are in this case misleading because distributions are not normal. In some shallow and highly productive regions this is particularly notable. In the case of PP (gC $m^{-2}$; see figure 1a) this compensated by the integration depth and, thus, the weight of these pixel, although relevant, is less critical than in PP(gC $m^{-3}$, Fig 1b) where the influence of vertically averaging just over few surface values exacerbates the differences with overall values. As shown in Fig. 1b, due to the lack of large shallow and productive areas, there are few values above 30 (gC $m^3$) in the Western Med, whereas high PP is more frequent in the Adriatic (red) and in the Eastern Med (blue). If the pixels with values >30 gC $m^3$ are plotted (Fig 2.) it becomes evident that most of

them are located in shallow waters of the Gulf of Gabes and in the Nile delta, and less so, in the northern Adriatic. To avoid these problems, we refer now to median values in Table 2, yet mean values are still provided as a reference.

[Figure]

Fig. 1 Frequency histograms for a) Integrated and b) vertically averaged PP estimations. Blue (East Med.) green (West Med), red (Adriatic). Y-axis in fig 1b is logarithmic.

[Figure]

Figure 2. Map showing the location of the pixels with values >30 gC m$^{-3}$ in blue.

**#3.10:** Line 239-241: I suggest starting the sentence by citing the figure you are referring to instead of citing it at the end.

This change has been performed.

**#3.11:** Line 243. "…*the coefficient of variation of primary production (CVPP)…*" how did you calculate this coefficient? Defining this "coefficient of variation of primary production" would also clarify why there are 2 different "coefficient of variation of primary production", one in figure 2c and another in figure 3b.

The Coefficient of Variation is a typical dispersion measurement defined as Standard Deviation / Mean.

***#3.12:*** Tab 1: Could you please insert the exact reference (product ID as for satellite daily data described above) for the 8-days e 4km resolution data taken from CMEMS? I cannot find them. Why is \*\*\* only for "Mediterranean Sea"? shouldn't it also be on "Open ocean waters"?

It has been added now in the manuscript the specific product ID and variable used for the analysis. In In section 2.1 – Remote sensing data, it stands: "*The dataset name was 'dataset-oc-med-chl-multi-l3-chl_1km_daily-rep-v02' and the variable used was 'mass_concentration_of_chlorophyll_a_in_sea_water (Chl)', obtainable in a NetCDF-4 file format*."

As for the second question, we used \*\* for data that we took from Table 2. We used \*\*\* for data that we estimated using our coastal results added to the open waters results that we obtained in the literature. We clarify now:

*\*\* PP estimated by averaging published satellite data shown in Table 2.*

*\*\*\* ΣPP estimated adding coastal waters data from this study to open ocean waters data taken from Table 2.*

***#3.13:*** Line 290-291: How did you estimate this interannual variability?

This value of interannual variability was calculated using the formula (PP max – PP min/ PP average) \* 100. A PPtot value was calculated for every year. Then, the minimum value is subtracted from the maximum value and divided by the average.

***#3.14:*** Line 298-299: How did you calculate the "the filtered low frequency signal" for PP and CHL?

First, we calculated the anomalies of the total monthly PP (ΣPP) for each of the 180 months between January 2002 – Dec 2015. Then, we used the *smooth* function of Matlab applying the *sgolay* filter with a span degree of 17 to the anomalies of the total PP. This sgolay filter uses the Savitzky-Golay method with the polynomial degree specified by degree. In the case of Figure 4, we were using monthly data (in total 180 time steps) for each region (a-whole Mediterranean coast, b-western coast, c-eastern coast, d- Adriatic coast) to visualize PP variability and trends. To filter the low frequency signal, we used a span degree of 17. This degree was therefore filtering about 8 months before and after every time step (about 1.5 years). Some proofs were doing using a degree of 29 (2.5 years low signal) or 59 (5 years low signal).

***#3.15:*** Line 306-307: "Most of these regions presented declining PP trends…". This sentence does not seem so evident observing fig 5a. The only evident negative trend is in the Gulf of Gabes as underlined by the authors. Moreover I believe that fig 5a and graphs in fig4 are quite inconsistent. In fig 4 trends are negative for Mediterranean Sea, west and east sub-basin, while for Adriatic Sea there is no evident trend. From fig 5a I'd say that on average Mediterranean Sea trend is quite positive (red areas are greater than blue ones). For west and east sub-basin the negative trend shown in fig 4a is not so evident, especially for the eastern compart. About Adriatic Sea fig 5a shows a clear positive trend. Could the authors explain this apparently discrepancy and how a reader should interpret it?

We recognize that, despite being different estimations, Figure 4 and Figure 5 could be misleading. Figure 4 showed a regional trend based on annual data (one mean value per year). This was higly influenced by the decline in PP observed in year 2012. Hence, we decided to provide only the

trend obtained with the complete time series, as shown in Fig. 5. This issue was also addressed in *#1.17*.

*#3.16:* Fig 6: line 324: it is not specified (here or in the text) what blue lines meaning. Are they annual PP anomalies?

This figure shows the relationship between coastal pelagic PP and SST, NAO and MOI. Blue lines indicate the global PP monthly anomalies for the whole Mediterranean Sea, whereas red lines indicate the SST monthly anomalies, NAO or MOI indices respectively. Blue lines are the same in Fig 6a-c. It has been indicated in Figure 6a caption: "*Figure 6: Relationship between coastal pelagic primary production (ΣPP anomalies, blue lines) and a) SST anomalies, b) NAO index and c) MOI index (red lines)*")

*#3.17:* Line 327-336: R1 to R9 are represented in fig 7 as PP in gC m$^{-2}$ d$^{-1}$, but in tab 3 there are mean annual PP values. Again, it should be defined how you estimate mean annual PP starting from daily PP.

The SOM analysis was performed using the original PP time series in gC m$^{-2}$ d$^{-1}$. The patterns that we obtained are shown in Fig. 7. In order to provide mean values for the different regions, we give information of the mean annual values in gC m$^{-2}$ y$^{-1}$. To be consistent with the rest of the analysis and to obtain mean values of PP at every area.

*#3.18:* Line 392: "…whereas Case-2 waters are reduced to less than 5% of the whole basin.". Is there any reference for this statement?

In the manuscript we provided some references "*Indeed, previous studies agreed that Indeed, Case-1 waters are largely predominant in the coastal Mediterranean regions whereas Case-2 waters are reduced to less than 5% of the whole basin (Antoine and André, 1995; Bosc et al., 2004; Bricaud et al., 2002). In particular, they are confined to the north Adriatic Sea, Gulf of Gabes and around Nile delta where our PP estimations may present larger uncertainties (Antoine and André, 1995).*".

We obtained that value of about 5% from the literature. Bricaud et al. (2002) *explained in section 2.1. Computation of PP: "On these maps, Case 2 waters were identified by discarding the pixels where R(555) was > 0.025 (see Bricaud & Morel, 1987), and an ''average mask'' was selected. The corresponding area was 5% of the total area of the Basin, which is close to the estimate (4%) provided by Antoine et al. (1995). Although it is acknowledged that the extension of Case 2 waters may vary throughout the year, the use of such a constant mask (instead of a temporally variable mask) allows the spatial means of chlorophyll concentration (or primary production) to be computed over a fixed area, and therefore to be comparable from month to month. The same average mask was used for".*

Bosc et al. (2004) also use this value: "*The Mediterranean is particularly well adapted to ocean color studies as the conditions of observation are optimal (low cloudiness), and also because turbid-case 2 waters, where the interpretation of ocean color is complex, are here of marginal areal extent, covering less than 5% of the whole basin [Antoine et al., 1995].*"

Antoine and André (1995) specify "*Turbid case 2 waters are particularly extensive in the Adriatic Sea, covering about 25% of the basin area. Moreover*". Also: "*This procedure also allows turbid case 2 waters to be identified and thereafter discarded (black mask in images). They cover about 3.5% of the entire eastern Mediterranean, mainly located in the north Adriatic Sea, in the Gulf of Gabes, and around the Nile delta.*". In this article, they published a Table where they provide

information about the area excluding case 2 waters for the subprovinces of the Adriatic, Aegean, Ionian, North Levantine, South Levantine and Total Mediterranean Sea.

**Table 2.** Sea Surface Covered by Each of the Five Provinces Defined in the Present Work

| | Subprovince | | | | | |
|---|---|---|---|---|---|---|
| | Adriatic | Aegean | Ionian | North Levantine | South Levantine | Total |
| Area, $10^{12}$ m$^2$ | 0.134 | 0.193 | 0.773 | 0.244 | 0.336 | 1.68 |
| Percentage of whole eastern Mediterranean | 8 | 11.5 | 46 | 14.5 | 20 | 100* |
| Area excluding case 2 waters, $10^{12}$ m$^2$ | 0.100 | 0.192 | 0.760 | 0.241 | 0.330 | 1.62 |

*The eastern basin contributes to 67.4% of the whole Mediterranean area.

*#3.10:* Fig 8: Caption of the figure refers to another type of figure (seasonal PP). Fig 8a has a different color palette (and range) with respect to other 2 (b and c) images.

We apologize for the mistake. The figure caption has been corrected by "*Figure 8. a) ef-ratio in coastal waters (<200 m) of the Mediterranean Sea and estimated values of b) new (PPnew) and c) regenerated production (PPreg). Mean values for the period 2002-2016.*"

---

## Author Response (AR1)

**Manuscript Reference No**.: bg-2020-457
**Biogeosciences Discuss**., https://doi.org/10.5194/bg-2020-457-RC1, 2021
**Title:** "Pelagic primary production in the coastal Mediterranean Sea:
variability, trends and contribution to basin scale budgets" by Paula Maria Salgado-
Hernanz et al. 2021.

================================

**Response to Associate Editor Decision**

We would like first to thank the associated editor for his comments and his interest in our work. The main concern of the associated editor and the reviewers was about our finding of higher PP in eastern coast than in western coast of the Mediterranean Sea. Although it is surprising to observe higher coastal PP for the eastern than for the western basin, our results are consistent for both integrated and volumetric values. Higher coastal integrated PP in the eastern basin can be explained by its great coastal area compared to the western basin (twice larger). For the volumetric PP values, considering the lack of large shallow and productive areas in the western Mediterranean, there were few mean annual PP values above 30 gC m$^{-3}$ in this basin whereas high volumetric PP were more frequent in the Adriatic Sea and in the eastern Mediterranean. Pixels with PP values >30 gC m$^{-3}$ were located in shallow waters of the Gulf of Gabes and in the Nile Delta, and less so, in the northern Adriatic. Coastal values are highly dependent on local enrichment processes. Apart from the influence of rivers (mainly the Rhone in the western side, the Po in the Adriatic and the Nile in the eastern side), major influence in shelf production generally comes from other sources such as the inputs of the Black Sea in the northern Aegean, and from local processes in the Gulf of Gabes. This would highly increase the productivity in the eastern coastal areas. While some overestimation of PP may occur in these waters due to the distinct optical conditions of these waters (Bosc et al., 2004), the Gulf of Gabes is considered one of the most productive coastal areas of the Mediterranean (e.g. D'Ortenzio and Alcalà, 2009). Its shallowness (< 50 m <at 110 km off the coast), unique tidal range (maximum >2 m) and the lack of summer nutrient exhaustion undoubtedly contribute to the high productivity found in the coastal areas of the eastern basin (Béjaoui et al., 2019).

Furthermore, the use of the CMEMS Chl data allows having more accurate PP results because specific regional algorithms are used contrary to the Chl data generated by a single algorithm for the open and coastal areas together (Bosc et al., 2004; Bricaud et al., 2002a). The improved Chl algorithm that we used is regionally tuned to consider the characteristics of the different Mediterranean regions. Certainly, this is not exempt from inaccuracies due to turbidity, but it is a definitive improvement as compared to algorithms used for PP estimations. Additionally, the coastal PP obtained here for highly turbid regions such as Nile river delta or North Adriatic Sea were in accordance with previous published works that estimated PP (e.g. Antoine and André, 1995; Umani, 1996; Zoppini et al., 1995).

We took into consideration the suggestions provided by the three anonymous referees and the associate editor in further revising the manuscript, which we believe is now improved. When comments were common to several reviewers (e.g. comment about the difference of PP between eastern and western coasts) we provided the same responses. The main changes that we have included in the revised manuscript are:

- We explain how the use of CMEMS data improved the estimation of coastal PP derived from satellite data.
- We discuss how the domestic and industrial wastewater inputs can affect PP in certain regions of the Mediterranean coast.
- A Table has been added (now Table 1) in *section 2.2. Primary production estimates*. In this Table we define and indicate the acronyms and units of all primary production values given in the manuscript.
- Previous Table 1 (now Table 2) has been modified in order to provide median values (with mean production values between brackets). This adjustment has been done as PP data do not follow a normal distribution in coastal regions and the numbers of pixels with values higher than 10-30 gC m$^{-3}$ was higher in the eastern basin. This additional information is provided to help understanding why integrated coastal primary production is higher in the eastern Mediterranean than in the western part.
- Supplementary Table 1 has been added in Supplementary data as proposed by the referee 1 in comment #1.39.
- Supplementary figures 1 and 2 have been added in Supplementary data. They are now referred to in *section 3.2. Long-term variability and trends*.
- Colobar of Figure 5 has been changed as suggested in comment #1.42.
- The conclusion section has been strengthened following the recommendations provided in the referee comments #1.6 and #1.30. Specifically, it is now highlighted that primary production variability in the Mediterranean coastal regions is dominated by interannual and sub-decadal variations. In addition, different along-shelf zones based on their temporal PP patterns are observed for the first time in the study region. It is pointed out that regional changes in PP could be of paramount importance to understand variations in higher trophic levels.
- Other minor changes and references suggested by the three referees have been thoroughly considered.

We believe that the manuscript is now improved relative to its previous version, and we hope that you will find it acceptable for publication in Biogeosciences.

Yours sincerely,

P.M. Salgado-Hernanz

Dear reviewer, we would like to thank you the interest you have shown in our study and therefore consider it as suitable for publication in Biogeosciences. Here there are our answers:

**Response to main comments Referee #1:**

*#1.1:* One of the things that struck me was the higher primary productivity (per m³) and chlorophyll data in the Eastern basin than Western basin which I found very surprising. Due to low river inputs in the Eastern Mediterranean compared to the Western Mediterranean I naturally would expect the Western Mediterranean coastal area to be more productive. I would like the authors to discuss this in more detail – is this due to the uncertainty/overestimation of chlorophyll in the Gulf of Gabes as mentioned by the authors or is there observational data to back up the high productivity here.

**Response to #1.1**: We understand that the reviewer would expect higher PP in the Western than in the Eastern basin. Although it is surprising to observe higher coastal PP for the eastern than for the western basin, our results are consistent for both integrated and volumetric values. Higher coastal integrated PP in the eastern basin can be explained by its great coastal area compared to the western basin (twice larger). For the volumetric PP values, considering the lack of large shallow and productive areas in the western Mediterranean, there were few mean annual PP values above 30 gC m$^{-3}$ in this basin whereas high volumetric PP were more frequent in the Adriatic Sea and in the eastern Mediterranean. Pixels with PP values >30 gC m$^{-3}$ were located in shallow waters of the Gulf of Gabes and in the Nile Delta, and less so, in the northern Adriatic. Coastal values are highly dependent on local enrichment processes. Apart from the influence of rivers (mainly the Rhone in the western side, the Po in the Adriatic and the Nile in the eastern side), major influence in shelf production generally comes from other sources such as the inputs of the Black Sea in the northern Aegean, and from local processes in the Gulf of Gabes. This would highly increase the productivity in the eastern coastal areas. While some overestimation of PP may occur in these waters due to the distinct optical conditions of these waters (Bosc et al., 2004), the Gulf of Gabes is considered one of the most productive coastal areas of the Mediterranean (e.g. D'Ortenzio and Alcalà, 2009). Its shallowness (< 50 m <at 110 km off the coast), unique tidal range (maximum >2 m) and the lack of summer nutrient exhaustion undoubtedly contribute to the high productivity found in the coastal areas of the eastern basin (Béjaoui et al., 2019).

Averages are in this case also misleading because distributions are not normal. For a better understanding, we present to the reviewer the distribution of volumetric and integrated PP in function of the number of pixels (see above Figure 1) and a map showing only pixels with volumetric PP higher than 30gC m$^{-3}$ (see above Figure 2). In some shallow and highly productive regions this is particularly notable. In the case of integrated PP (gC m$^{-2}$, see herein Fig. 1a) this compensated by the integration depth and, thus, the weight of these pixel, although relevant, is less critical than in volumetric PP (gC m$^{-3}$, see herein Fig. 1b) where the influence of vertically averaging just over few surface values exacerbates the differences with overall values. As shown in herein Fig. 1b, due to the lack of large shallow and productive areas, there are few values above 30 (gC m³) in the western Mediterranean, whereas high PP is more frequent in the Adriatic Sea (red) and in the eastern Mediterranean (blue). If the pixels with values >30 gC m³ are plotted (see next Fig. 2) it becomes evident that most of them are located in shallow waters of the Gulf of

Gabes and in the Nile Delta, and less so, in the northern Adriatic. To avoid these problems, we refer now to median values in Table 2, yet mean values are still provided as a reference.

[Figure]

Figure 1. Frequency histograms for a) Integrated and b) vertically averaged PP estimations. Blue (East Med.) green (West Med), red (Adriatic). Note that Y-axis in Fig 1b is logarithmic.

[Figure]

Figure 2. Map showing the location of the pixels with values >30 gC m$^{-3}$ in blue.

**Actions to #1.1**: We changed Table 2 showing median PP values rather than mean PP values. Furthermore, we modified the text. In discussion, it now reads:

"Contrary to expectation, we observed that the eastern basin contributes more than the western basin to overall coastal production (51% and 25% respectively; Table 2). Its great extension (twice higher than the western basin) and the increased productivity in regions like Gabes, the Nile and the northern Aegean Sea may explain greater coastal PP in the eastern basin than in the western basin. Additionally, due to the lack of large shallow and productive areas in the western basin, we observed few volumetric PP values above 30 gC m$^{-3}$ in the western Mediterranean whereas high PP is more frequent in the Adriatic Sea and in the eastern Mediterranean in shallow waters of the Gulf of Gabes and in the Nile Delta."

**Table 2.** Surface area, chlorophyll mean (Chl) ± standard deviation (SD), ΣPP, the correspondent % to $\Sigma PP_{Coast}$ (% $\Sigma PP_{Coast}$), $PP_{annual}$ median ± SD ($PP_{annual}$ mean) and $PP_{VOL}$ median ± SD ($PP_{VOL}$

mean) for the Mediterranean Sea, open ocean waters, and coastal waters during the period 2002–2016.

| | Surface area $(10^3 \text{ km}^2)$ | (%) | Chl $(\text{mg m}^{-3})$ | $\Sigma PP$ (Gt C) | % of $\Sigma PP_{\text{Coast}}$ | $PP_{\text{annual}}$ $(\text{g C m}^{-2})$ | $PP_{\text{VOL}}$ $(\text{g C m}^{-3})$ |
|---|---|---|---|---|---|---|---|
| Mediterranean Sea | 2,504 | | 0.19±0.78* | 0.349±0.118*** | | 140±40** | |
| Open ocean waters | 1,975 | | 0.11±0.18* | 0.308±0.118 | | 136±40** | |
| Coastal waters | 529 | 100 | 0.30±0.17 | 0.041±0.004 | 100 | 83±75 (100) | 1.16±9.60 (2.93) |
| Western coast | 141 | 27 | 0.21±0.14 | 0.011±0.001 | 25 | 90±39 (98) | 1.23±2.61 (1.59) |
| Eastern coast | 287 | 54 | 0.30±0.16 | 0.021±0.002 | 51 | 73±86 (93) | 1.01±11.8 (3.34) |
| Adriatic coast | 101 | 19 | 0.39±0.23 | 0.010±0.001 | 24 | 99±76 (124) | 1.50±7.23 (3.27) |

* Mean surface Chl values obtained by averaging the 8-days and 4-km resolution of surface satellite Chl values obtained from CMEMS (Salgado-Hernanz et al., 2019).

** PP estimated by averaging published satellite data shown in Table 3.

*** ΣPP estimated adding coastal waters data from this study to open ocean waters data obtained from Table 3.

*#1.2:* Can the authors put any error estimates on this or give a lower bound on the Eastern value.

**Response to #1.2**: Considering that most published PP in the Mediterranean Sea use standard deviation as error estimates descriptor in their studies, we have maintained the standard deviation (S.D.) instead of the standard error (S.E.) because. The reviewer can find the SD of all median values in Table 2.

*#1.3:* Likewise, can you really give a contribution of total primary production to the Mediterranean for coastal areas if the studies that estimated these total Mediterranean values exclude the highly productive coastal areas (i.e. North Adriatic, Gulf of Gabes) as mentioned by the authors on Line 76 due to the high turbidity and thus inaccurate values.

**Response to #1.3:** We understand the reviewer concern. However, we use an improved Chl algorithm that is regionally tuned to consider the characteristics of the different Mediterranean regions. Certainly, this is not exempt from inaccuracies due to turbidity, but it is definitely an improvement from previous algorithms used for PP estimations. Furthermore, as explained in the manuscript (section 4.1. Costal primary production), "*the coastal PP obtained here for highly turbid regions such as Nile river delta or North Adriatic Sea were in accordance with previous published works that estimated in situ PP. Indeed, $PP_{annual}$ values off the Nile river delta, >100 g C m$^{-2}$ estimated here, are only slightly higher than those reported by Antoine et al. (1995) (80-100 g C m$^{-2}$). In the case of the Adriatic Sea, Umani (1996) reported values of PP from 50 to 200 g C m$^{-2}$ y$^{-1}$, while Zoppini et al. (1995) estimated PP rates from 210 to 260 g C m$^{-2}$ y$^{-1}$ in the northern coastal areas. Our estimations range between 100 and >350 (with mean values of 123±106 g C m$^{-2}$)*".

In addition, we can give a contribution of total PP to the Mediterranean for coastal areas since previous studies did not include coastal areas. We specify in the manuscript that *"Several studies have assessed PP at the scale of the entire Mediterranean Sea from satellite remote sensing data (Bosc et al., 2004; Bricaud et al., 2002b; Lazzari et al., 2012). However, coastal areas were generally ignored in such studies, so that their contribution to basin scale budgets is still largely unknown"*. Indeed, we emphasize the need to our study since in those previous studies they did not include high productive waters in their analysis. We specify here details about those previous studies:

- In Bricaud 2002, the authors mentioned that they exclude the highly productive coastal waters when providing mean spatial values. They believed that they introduced an overestimation of the Chl means so they decided to create a mask to remove them from their analysis: *"usual chlorophyll algorithms are generally not valid and lead to erroneous (overestimated) values of biomass and primary production……… In spite of the reduced spatial coverage of these turbid waters, the high (artifactual) chlorophyll concentrations may influence significantly the spatial means, so that they have to be discarded from these means. With this aim, a simplified, constant ''mask'' for turbid Case 2 waters was defined"*. Specifically, the authors considered that turbid Case 2 waters are *"In the Mediterranean Basin, these waters are essentially located in some coastal areas (Northern Adriatic Sea, Kerkenna shelf, gulf of Gabes, etc.) and within the plumes of the major rivers (Rhone, Po , Ebra, Nile, etc.)"*.

- In Bosc et al. 2004, the authors mentioned they used the same masks used in Bricaud et al 2002: *"masks corresponding to the whole Mediterranean Basin (except the Black Sea, which was excluded) or to one of its provinces were applied. Finally, in each region, turbid- Case 2 waters were discarded by applying a constant mask, defined as described by Bricaud et al. [2002]"*.

- In Lazzari et al. 2012, the authors pointed out that the results they provide in terms of pelagic NPP and Chl concentrations exclude the continental platform: *"Shallow areas (depth<200 m) and marginal seas were excluded from the statistics because the model was designed for pelagic areas"*.

   **#1.4:** Alternatively, is the CMEMS chlorophyll data corrected for these high turbidity areas, reducing the uncertainty in your estimates compared to previous studies like Bosc et al. 2004 and Bricaud et al., 2002 where these areas were excluded?

**Response to #1.4**: In the present study, we used the most adequate regional Chl product available up to date for the Mediterranean Sea. The use of the CMEMS Chl data allows having more accurate PP results because specific regional algorithms are used contrary to the Chl data generated by a single algorithm for the open and coastal areas together (Bosc et al., 2004; Bricaud et al., 2002a). The improved Chl algorithm that we used is regionally tuned to consider the characteristics of the different Mediterranean regions. Certainly, this is not exempt from inaccuracies due to turbidity, but it is a definitive improvement as compared to algorithms used for PP estimations. Additionally, the coastal PP obtained here for highly turbid regions such as Nile river delta or North Adriatic Sea were in accordance with previous published works that estimated PP (e.g. Antoine and André, 1995; Umani, 1996; Zoppini et al., 1995). In the studies

mentioned by the reviewer, no specific regional Chl algorithm was used. Both Bricaud et al. (2002) and Bosc et al. (2004) used Chl resulting from reprocessing #4, provided in July 2002 (see http://seawifs.gsfc.nasa.gov/SEAWIFS/RECAL/Repro4). The bio-optical algorithm was the ''OC4v4''algorithm proposed by O'Reilly et al. (1998).

**Action**: In the manuscript, it reads:

"We used the Mediterranean Sea Level-3 reprocessed surface chlorophyll concentration product (Chl L3) obtained from the EU Copernicus Marine Environment Monitoring Service (CMEMS). This product merges multi satellite observations,a nd is available at 1-day and 1-km resolution(http://marine.copernicus.eu/OCEANCOLOUR_MED_CHL_L3_REP_OBSERVATI ONS_009_073). Specifically, the dataset used is 'dataset-oc-med-chl-multi-l3-chl_1km_daily-rep-v02' and the variable name used is 'mass_concentration_of_chlorophyll_a_in_sea_water (Chl)' obtainable in a NetCDF-4 file format. This Chl L3 dataset is derived with an updated version of the regional algorithm MedOC4 (Mediterranean Ocean-Colour 4 bands MedOC4, Volpe et al., 2019) for pelagic deep Case-1 waters and the AD4 algorithm (ADriatic 4 band; Berthon and Zibordi, 2004; D'Alimonte and Zibordi, 2003) for Case-2 coastal waters (generally shallow and turbid waters)".

*#1.5:* Following on from this I would like to ask the authors whether they have considered doing the analysis (with small adjustments) for the whole Mediterranean Sea so that comparison for coastal primary against the whole Mediterranean is coherent using data that has been prepared in the same way. This would enhance their conclusions on the contribution of the coastal zone to primary productivity in the Mediterranean.

**Response to #1.5**: The focus of the present study is the coastal zone of the Mediterranean Sea at a reasonable resolution to be able to identify the main coastal PP features and with the aim of defining different coastal regions that are oversaw in more general PP estimations. Therefore, we decided to exclude open ocean waters. We agree that running the entire Mediterranean Sea would be more coherent but, since there are plenty of studies providing this information (we do in fact review all of them in Table 2), we do not feel that this is a major drawback. We also used the same PP model than in other studies dealing with the non-coastal waters.

*#1.6:* Generally, the manuscript is well written and English is good. I do feel that the conclusions can be strengthened and it would be nice if the authors could specifically say how this dataset/analysis will be useful to the Mediterranean science community. If the authors address the comments I have made, I think this manuscript can be considered for publication in Biogeosciences.

**Response to #1.6:** We are thankful for the positive comment of the reviewer. We have now enriched the Conclusion section with several sentences explaining the importance of understanding coastal production and its long-term variability in the Mediterranean Sea.

**Action**: We improved the text in the conclusion section. It now reads:

"In summary, pelagic PP in coastal shelves of the Mediterranean Sea during the period 2002-2016 was estimated in this study for the first time using available satellite ocean colour products. We estimated that 12% of PP of the Mediterranean Sea is attributable to coastal pelagic

production and from that, about 80% of this carbon fixation is sustained by regenerated pathways. High PP spatial variations were observed among the different regions, as mainly driven by major river effluents, exchanges with nearby seas (i.e. Black Sea and the Atlantic Ocean) and by local processes. Our study shows that some coastal areas are indeed highly productive (>400 g C m$^{-2}$) and sustain a large percentage of overall coastal production. Indeed, their temporal variability could be of paramount importance to understand variations in higher trophic levels (e.g. Piroddi et al., 2017). Despite that temporal variability is dominated by interannual and sub-decadal variations, our analysis reveals a weak global negative PP trend in the Mediterranean Sea related to climate driven patterns (i.e temperature increase). Nevertheless, long-term effects can be regionally variable (i.e. PP trends in the Adriatic Sea are positive) and variations in fluvial nutrient inputs, together with other processes such as ocean warming in coastal regions, including heat waves, deserve a closer look as longer ocean colour databases become available. Finally, we identify 18 along-shelf zones based on their temporal PP patterns. Two main PP groups were observed: zones with strong cross-shore gradients, typically found in wider estuarine regions and homogeneous zones within narrow continental shelf areas. These two types of coastal waters clearly characterize the coastal area of a sea where coastal waters are otherwise strongly influenced by ocean conditions.".

**Response to attached detailed comments from Reviewer # 1:**

The attached supplement provides my detailed comments on the manuscript.

*#1.7:* Title: I suggest removing pelagic as I currently feel the title is an oxymoron. I don't consider areas>5m deep to be pelagic?

**Response to #1.7:** We intended to clarify that we are not estimating the contribution of the benthic PP in the coast. We agree that, in some contexts, the term may be confusing in this context since pelagic often refers to open waters. Nevertheless, pelagic is also commonly used as opposed to benthic in coastal studies (ie, pelagic, bentho-pelagic and benthic fish classification). Some other authors (i.e. Macias et al., 2017) also use the term pelagic as opposed to benthic in coastal PP studies. In our study, we refer to the "water-column- primary production.

*#1.8:* Line 76: 'Coastal areas were generally ignored in such studies'. Following my statement above, both Bricaud et al. and Bosc et al. masked areas of high turbidity where data is uncertain. Please comment on what improvements have been made to the CMEMS data to make it relevant in this study (if improvements have indeed been made).

**Response to #1.8:** As explained previously, the Chl product available from CMEMS has been tailored to the Mediterranean region by using the regional algorithm MedOC4 for Case-1 waters (i.e. waters with low inorganic particles concentrations, low turbidity) and the AD4 algorithm for Case-2 waters (i.e. waters with high inorganic particles concentrations, high turbidity). Considering that the CMEMS data combined those two algorithms, its resulting chlorophyll concentration must be more precise for areas with high turbidity and thus, more relevant than chlorophyll concentration derived from no specific regional Chl algorithm as Bricaud et al. and Bosc et al. did use.

**Action**: We modified the text. Now, read in M&Ms section:

"Chl L3 dataset is derived by means of the Mediterranean Ocean Colour regional algorithms, that uses an updated version of the regional algorithm MedOC4 (Mediterranean Ocean-Colour 4 bands MedOC4, Volpe et al., 2019) for Case-1 waters (deep pelagic waters, low turbidity) and the AD4 algorithm (ADriatic 4 band; Berthon and Zibordi, 2004; D'Alimonte and Zibordi, 2003) for Case-2 waters (coastal shallow waters, high turbidity). Considering the great range of turbidity along the Mediterranean coastal regions, we believed that the use of CMEMS data with its regional algorithms for Case-1 (waters with low inorganic particles concentration) and Case-2 (waters with high inorganic particles concentration) waters is more relevant for our study."

*#1.9:* Line 152: How does this assumption impact your results? Are waters in the Mediterranean well mixed to 200m deep?

**Response to #1.9:** There is no, to our knowledge, parameterization of the Chl vertical profile from the Chl value at surface that would be valid for coastal waters. Therefore, it is not really feasible to assess whether ignoring possible deep-chlorophyll maxima in coastal areas is significantly affecting our PP estimates. Given the variability in coastal waters, we considered that using a homogenous profile is a better assumption than using global parameterizations of the shape of the vertical profile as a function of the surface Chl that are valid only for deep open ocean waters.

*#1.10:* Line 224: As already mentioned, considering that some of these authors exclude the productive areas that you are including is this a fair comparison? The analysis would be a lot stronger if the same dataset was used to compare coastal production vs total production in the Mediterranean.

**Response to #1.10:** We understand the reviewer statement and we agree that adding total production data in the Mediterranean Sea would have improve the analysis. However, our present concern was to fill a gap: the lack in published works of coastal PP estimations in the Mediterranean Sea.

Several papers already published estimations of PP for the whole Mediterranean Sea. For that reason, the focus of the present study is the coastal zone of the Mediterranean Sea at a reasonable resolution to be able to identify the main coastal PP features and with the aim of defining different coastal regions that are oversaw in more general PP estimations. In addition, we use here L3 CMEMS dataset and CMEMS Chl for the open ocean do not differ significantly from the Chl data used in previous studies such as Bricaud et al. (2002b).

*#1.11:* Equation 1: Considering you assign a uniform chlorophyll concentration it is not really dependent on depth?

**Response to #1.11:** The chlorophyll concentration is indeed uniform with depth, however irradiance is varying with depth by virtue of how light propagates in the water column. So, yes, t PP is dependent on depth.

*#1.12:* Line 236-238: The authors mention that the Eastern Mediterranean has twice the amount of coastal primary productivity than the western basin due to its size. However, primary productivity per unit volume is also twice the amount of western shelf and is also higher

than that observed in the Adriatic? Why is this? This is not what I would expect, especially given the little river inputs along the coast of the Eastern Mediterranean.

**Response to #1.12:** As explained in **#1.1**, mean primary production per unit volume exaggerates the production in shallow areas. We consider that mean values are highly affected by the production in these areas and, therefore, we now refer to median values which reveal that coastal median PP per unit volume is 16 % lower in the Eastern that in the western Mediterranean.

**Action**: We modified Table 2 (Table 1 in previous version of ms)

**Table 2.** Surface area, chlorophyll mean (Chl) ± standard deviation (SD), $\Sigma PP$, the correspondent % to $\Sigma PP_{Coast}$ (% $\Sigma PP_{Coast}$), $PP_{annual}$ median ± SD ($PP_{annual}$ mean) and $PP_{VOL}$ median ± SD ($PP_{VOL}$ mean) for the Mediterranean Sea, open ocean waters, and coastal waters during the period 2002–2016.

| | Surface area ($10^3$ km$^2$) | (%) | Chl (mg m$^{-3}$) | $\Sigma PP$ (Gt C) | % of $\Sigma PP_{Coast}$ | $PP_{annual}$ (g C m$^{-2}$) | $PP_{VOL}$ (g C m$^{-3}$) |
|---|---|---|---|---|---|---|---|
| Mediterranean Sea | 2,504 | | 0.19±0.78* | 0.349±0.118*** | | 140±40** | |
| Open ocean waters | 1,975 | | 0.11±0.18* | 0.308±0.118 | | 136±40** | |
| Coastal waters | 529 | 100 | 0.30±0.17 | 0.041±0.004 | 100 | 83±75 (100) | 1.16±9.60 (2.93) |
| Western coast | 141 | 27 | 0.21±0.14 | 0.011±0.001 | 25 | 90±39 (98) | 1.23±2.61 (1.59) |
| Eastern coast | 287 | 54 | 0.30±0.16 | 0.021±0.002 | 51 | 73±86 (93) | 1.01±11.8 (3.34) |
| Adriatic coast | 101 | 19 | 0.39±0.23 | 0.010±0.001 | 24 | 99±76 (124) | 1.50±7.23 (3.27) |

* Mean surface Chl values obtained by averaging the 8-days and 4-km resolution of surface satellite Chl values obtained from CMEMS (Salgado-Hernanz et al., 2019).
** PP estimated by averaging published satellite data shown in Table 3.
*** $\Sigma PP$ estimated adding coastal waters data from this study to open ocean waters data obtained from Table 3.

**#1.13:** Line 241-242: What about the Nile delta and Gulf of Gabes – these stand out to me as high areas of primary production based on Figure 1.

**Response to #1.13:** We agree with the reviewer.

**Action**: We added a sentence. Now reads in Results section (3.1.)
"*However, in some coastal regions of the eastern basin like the Gulf of Gabes and the Nile Estuary primary production is outstandingly high (>300 g C m$^{-2}$)*"

**#1.14:** Table 1: What are the uncertainties? Standard deviation? Please state this in the caption

**Response to #1.14:** The uncertainties indicate the Standard Deviation (S.D). It is now specified in the caption.

**Action**: We modified Table 2 caption (Table 1 in previous version of ms). Now read:

"**Table 2.** Surface area, chlorophyll mean (Chl) ± standard deviation (SD), ΣPP, the correspondent % to $\Sigma PP_{Coast}$ (% $\Sigma PP_{Coast}$), $PP_{annual}$ median ± SD ($PP_{annual}$ mean) and $PP_{VOL}$ median ± SD ($PP_{VOL}$ mean) for the Mediterranean Sea, open ocean waters, and coastal waters during the period 2002–2016."

*#1.15:* Table1: Why did you use a different product to estimate chlorophyll in the whole Mediterranean Sea or open ocean water rather than the same one as coastal waters? Why couldn't you also estimate primary productivity using the whole dataset? Then it is a coherent analysis and you are comparing like for like. It would then enable comparison of the coastal ocean vs the entire Med Sea in the temporal trend analysis too.

**Response to #1.15:** We used a different product because the ones used in previous studies (for the open areas of the Mediterranean Sea) are not valid in coastal waters. We used the CMEMS Chl data rather that Chl data from Bricaud et al. or Bosc et al. because, in coastal regions, turbid waters are expected. Nevertheless, when turbidity was low in some coastal waters, we observed that the CMEMS and other Chl data were consistent.

*#1.16:* Figure2/3. What is the difference between Fig 2c and Fig 3b?

**Response to #1.16:** Figure 2c shows the coefficient of variation (CV) of the mean primary production per unit area, in g C m$^{-2}$, of the surface waters. Figure 3b shows the CV of the mean productivity per unit volume, in g C m$^{-3}$.

*#1.17:* Figure 5/Line 303: The authors say there is no significant trend in primary productivity in the Adriatic based on Figure 4. Why then does the Adriatic actually show the largest trend in Figure 5 with almost the entire 'coastal' Adriatic showing a positive trend? Likewise, I can't really see any trends in the Western basin despite the authors saying there was a slight significant negative trend in the Western basin based on Figure4.

**Response to #1.17:** The reviewer is right, Figure 4 and Figure 5 could bring misleading. Figure 4 showed a regional trend resulting from 15 points (one mean value per year). Moreover, from 2012 a reduction in PP is shown in every region but the Adriatic. The Adriatic region presented positive PP values for years 2013 and 2014 (see Supplementary figure 1) and this could change the PP trend when only 15 points are considered (i.e Fig. 4).

**Action**: In order to avoid misunderstanding, we now only provide trends obtained with the complete time series (Fig 5).

[Figure]

Figure 5: Trends in primary production and sea surface temperature. Values correspond to the change per decade. a) Theil-Sen trend in pelagic primary production estimated from daily values for the 2002-2016 period. b) Trend in SST temperature. Only significant trends (p < 0.05) are shown.

**#1.18:** Figure 7: Are the alongshore (Z areas) also based on the temporal patterns as indicated by the main caption to the figure?

**Response to #1.18:** The reviewer is right. SOM aggregates the characteristic temporal patterns according to their similarities. In section 2.3 Coastal regionalization we quote, line 18-190 "Then, 18 alongshore marine ecoregions were obtained considering the most relevant cross-shore limits of the SOM-derived regions (Z1 to Z18)."

**#1.19:** Line 358: The authors suggest enhanced production occurs in regions of freshwater influence. I would argue R7 is not. What other factors lead to high R7? Possibly domestic and industrial wastewater inputs?

**Action**: We have rephrased this sentence. Now it reads,

'An exception is the R7 pattern, which is exclusively located in the shallowest inner shelf of the Gulf of Gabes,..' .

In the discussion section, now read:

" Indeed, the Gulf of Gabes is a region displaying consistently high Chl and PP in most studies (e.g. Bosc et al., 2004; Barale et al., 2008). Drira et al. (2008) reported high biomass and toxic dinoflagellate blooms in the inner shelf of the Gulf of Gabes where surface nitrate concentration often exceeded 1µM. This enrichment is associated with degradation of the water quality attributed to industrial and urban activities (Hamza-Chaffai et al., 1997; Zairi and Rouis, 1999). However, even though these waters may suffer from eutrophication, satellite-borne data overestimates Chl within these waters, as revealed by Katlane et al. (2011) who observed constant high turbidity and suspended matter of industrial origin affecting these waters but also, reflection from the bottom affecting MODIS data. This suggests that general Chl algorithms may be particularly inaccurate in this region."

*#1.20:* Line 369-372: Interestingly Macias et al. (2018) use model simulations to show that primary production in the coastal region of the Western basin (including Gulf of Lions) is mostly influenced by circulation patterns, not river inputs. I suggest the authors include this reference somewhere in this manuscript.

**Response to #1.20:** We thank the reviewer for his/her suggestion.

**Action:** We added the reference in the text. We now make reference to the paper of Macias et al (2017) in the discussion section 4.1.

"Mediterranean coastal production is also supported by other sources such as local mesoscale processes (Macias et al., 2017)."

*#1.21:* Line 400: But the eastern Mediterranean also had higher values m3 that the western basin so it is not purely due to the bigger surface are of the eastern basin?

**Response to #1.21:** The reviewer is right. In order to avoid any confusion, we modified the text.

**Action**: We modified the text. It now reads in the discussion section:

"Contrary to what we would have expected, we observed that the eastern basin contributes more than the western basin to overall coastal production (51% and 25% respectively; Table 2). Its great extension (twice higher than the western basin) and the increased productivity in regions like Gabes, the Nile and the northern Aegean Sea may explain greater coastal PP in the eastern basin than in the western basin. Additionally, due to the lack of large shallow and productive areas in the western basin, we observed few volumetric PP values above 30 gC m$^{-3}$ in the western Mediterranean whereas high PP is more frequent in the Adriatic Sea and in the eastern Mediterranean in shallow waters of the Gulf of Gabes and in the Nile Delta."

*#1.22:* Lines 400-410: What about the influence of wastewater inputs (Powley et al., 2016) and submarine groundwater discharge (Rodellas et al., 2015)? It is mentioned again later in the discussion but I think it should be introduced earlier.

**Response to #1.22:** In the introduction, we do mention the influence of groundwater and human activities (line 52-54 "Terrestrial uploads of nutrients and organic matter originating from groundwater discharges, flash floods or river runoff as well as exchanges with seafloor strongly control the productivity of these waters (Basterretxea et al., 2010; Woodson and Litvin, 2014)"; line 72-73 "Moreover, intensive agricultural practices and urbanization have brought unprecedented use and contamination of coastal groundwater (Basterretxea et al., 2010; Tovar-Sánchez et al., 2014)." We thank the reviewer for his/her comment and we added these references. We have also emphasized the importance of groundwater and nutrient-rich effluents from human activities in the introduction.

**Action**: we modified the introduction and the discussion.

Now in the introduction, we can read:

"Nutrient-rich effluents from human activities in the coast (domestic wastewater, fertilizers, industrial, etc.) and natural river discharges affect continental shelf productivity in this sea, sustaining locally enhanced pelagic and benthic biomass."

Now read in the discussion:

"Variations in atmospheric deposition, groundwater and river outflows together with the influence of human activities through changes in landscape use and domestic wastewater management are important sources of nutrient in the ecosystem and thus, act as major drivers of PP in these waters (e.g. Paerl et al., 1999; Powley et al., 2016)."

*#1.23:* Line 431: What method did Barale et al., use? Is this also from satellites?

**Response to R#1:** We now precise in the text the method used by Barale et al.

**Action:** We modified the text. Now read in the discussion section.

 "Barale et al. (2008), using Chl anomalies derived from SeaWiFS data, observed a general decrease in Chl biomass in the Mediterranean Sea over the period 1998–2003".

*#1.24:* Line 443-445: Are you referring to the Biomodal Osciallation System (BIOS; i.e., Civitarese et al. 2010) here? If yes, I suggest you refer to it explicitly.

**Response to #1.24:** We do indeed refer to the Bimodal Oscillating System**.**

**Action:** We added this specification in the text. Now read in the discussion section:

"Alternatively, the Bimodal Oscillating System (BiOS), i.e. the feedback mechanism between the Adriatic and Ionian (Civitarese et al., 2010) peaking between 2004 and 2006 could have affected mass and nutrient exchanges between the Adriatic and the north Ionian Sea (Font et al., 2007; Schroeder et al., 2008; Šolić et al., 2008; Viličić et al., 2012)."

*#1.25:* Lines 475-485: What about domestic and industrial wastewater inputs into the sea? Powley et al. (2016) show they may be significant and certainly are likely to contribute to primary production in some areas of the Mediterranean coastline.

**Response to #1.25:** We thank the reviewer for his/her suggestion.

**Action:** We have included an explicit reference to domestic wastewater (Powley et al. 2016) in this paragraph now.

Now read in the discussion:

"Variations in atmospheric deposition, groundwater and river outflows together with the influence of human activities through changes in landscape use and domestic wastewater management are important sources of nutrient in the ecosystem and thus, act as major drivers of PP in these waters.

*#1.26:* Table 4: Please state how the errors are calculated.

**Response to #1.26:** We thank the reviewer for his/her comment.

**Action:** We modified the table 4 caption and we added standard deviation to the annual mean PP (PP$_{annual}$) and annual integrated PP ($\Sigma$PP).

Now read in table 4:

**Table 4.** Surface, river discharge flow ($Q$), annual mean PP ($PP_{annual}$), annual integrated PP ($\Sigma PP$) and its contribution respect to the total coastal Mediterranean Sea PP for each of the 18 alongshore zones characterized in the Mediterranean Sea. Mean and standard deviation (S.D.) are calculated from 14 year averages is calculated from 15-year averages (2002-2016).

| | Area | | $Q$ | $PP_{annual} \pm$S.D. | $\Sigma PP \pm$S.D. | |
|---|---|---|---|---|---|---|
| | ($km^2$) | (%) | ($km^3\ y^{-1}$) | ($g\ C\ m^{-2}$) | ($10^{-3}$ Gt C) | (%) |
| Z1 | 1,869 | 0.4 | 0.5 | 215 ±124 | 0.22±0.05 | 0.4 |
| Z2 | 7,226 | 1.4 | 1.2 | 107±58 | 0.62±0.08 | 1.4 |
| Z3 | 18,870 | 3.6 | 21.4 | 104±47 | 1.71±0.16 | 3.6 |
| Z4 | 15,196 | 2.9 | 57.7 | 128±72 | 1.61±0.12 | 2.9 |
| Z5 | 878 | 0.2 | 1.9 | 84±33 | 0.04±0.02 | 0.2 |
| Z6 | 20,392 | 3.9 | 14.6 | 101±64 | 1.62±0.20 | 3.9 |
| Z7 | 29,666 | 5.7 | 0.5 | 74±26 | 1.66±0.20 | 5.7 |
| Z8 | 8,178 | 1.6 | 3.7 | 81±34 | 0.40±0.09 | 1.6 |
| Z9 | 64,780 | 12.4 | 70.5 | 140±124 | 7.63±0.66 | 12.4 |
| Z10 | 40,997 | 7.8 | 35.8 | 89±37 | 2.81±0.25 | 7.8 |
| Z11 | 58,252 | 11.1 | 21.5 | 81±59 | 2.95±0.67 | 11.1 |
| Z12 | 25,720 | 4.9 | 21.2 | 123±76 | 2.25±0.33 | 4.9 |
| Z13 | 30,71 | 0.6 | 0 | 53±18 | 0.09±0.02 | 0.6 |
| Z14 | 16,814 | 3.2 | 21.3 | 97±61 | 1.21±0.14 | 3.2 |
| Z15 | 28,544 | 5.5 | 17 | 170±182 | 4.02±0.50 | 5.5 |
| Z16 | 46,065 | 8.8 | 0 | 48±17 | 1.85±0.12 | 8.8 |
| Z17 | 123,071 | 23.5 | 1.1 | 90±87 | 9.72±0.80 | 23.5 |
| Z18 | 13,411 | 2.6 | 6.1 | 125±56 | 1.24±0.18 | 2.6 |

*#1.27:* Figure 8: The figure caption and figure do not seem to match to me. There appears to be nothing about seasonality in the figure

**Response to #1.27:** We thank the reviewer for highlighting this mistake. There was indeed an error in the caption.

**Action**: We corrected the figure 8 caption. Now read in the Figure 8 caption:

" Figure 8. a) *ef*-ratio in coastal waters (<200 m) of the Mediterranean Sea and estimated values of b) new ($PP_{new}$) and c) regenerated production ($PP_{reg}$). Mean values for the period 2002-2016."

*#1.28:* Figure 9: What unit is annual PP in? Does it make a difference if you use m-2 vs m-3 vs total?

**Response to #1.28:** The annual PP unit is in $g\ C\ m^{-2}$.

**Action**: we modified the Figure 9 legend and caption. Now read:

"Figure 9: Relation between primary production, shelf width and river discharge flow ($Q$). Bubble colours indicate the PPannual ($g\ C\ m^{-2}$) for each of the 18 defined zones (see Fig. 8 and Table 4)."

***#1.29:*** Line 546-547– "Our analysis also reveals a weak negative PP trend which cannot be classed as climate driven" – but on lines 456 you say "we observed an influence of climate scale variability on coastal productivity as suggested by the inverse correlations between ΣPP and SST and, more loosely, with NAO and MO?" I don't agree/understand this conclusion based upon

**Response to #1.29:** We are sorry for the inconsistency. Changes and corrections had been made to avoid any confusion.

**Action**: We modified the text. Now read in the conclusion section:

"Despite that temporal variability is dominated by interannual and sub-decadal variations, our analysis reveals a weak global negative PP trend in the Mediterranean Sea related to climate-driven patterns (i.e., temperature increase). Nevertheless, long-term effects can be regionally variable (i.e. PP trends in the Adriatic Sea are positive) and variations in fluvial nutrient inputs, together with other processes such as ocean warming in coastal regions, including heat waves, deserve a closer look as longer ocean colour databases become available."

***#1.30:*** Conclusion: It would be nice if the authors could speculate how a dataset like this could be useful to the Mediterranean/scientific community. For example, could it be used to highlight coastal areas where additional monitoring should take place (Note the authors don't have to use this particular example)

**Response to #1.30:** We are thankful for the positive comment of the reviewer. We have now enriched the Conclusion section with several sentences explaining the importance of understanding coastal production and its long-term variability in the Mediterranean Sea.

**Action**: We modify the text. Now read in conclusion section:

"In summary, pelagic PP in coastal shelves of the Mediterranean Sea during the period 2002-2016 was estimated in this study for the first time using available satellite ocean colour product. We estimated that 12% of PP of the Mediterranean Sea is attributable to coastal pelagic production and from that, about 80% of this carbon fixation is sustained by regenerated pathways. High PP spatial variations were observed among the different regions, as mainly driven by major river effluents, exchanges with nearby seas (i.e. Black Sea and the Atlantic Ocean) and by local processes. Our study shows that some coastal areas are indeed highly productive (>400 g C m$^{-2}$) and sustain a large percentage of overall coastal production. Indeed, their temporal variability could be of paramount importance to understand variations in higher trophic levels (e.g. Piroddi et al., 2017). Despite that temporal variability is dominated by interannual and sub-decadal variations, our analysis reveals a weak global negative PP trend in the Mediterranean Sea related to climate drive patters (i.e temperature increase). Nevertheless, long-terms effects can be regionally variable (i.e. PP trends in the Adriatic Sea are positive) and variations in fluvial nutrient inputs, together with other processes such as ocean warming in coastal regions, including heat waves, deserve a closer look as longer ocean colour database becomes available. Finally, we identify 18 along-shelf zones based on their temporal PP patterns. Two main PP groups were observed: zones with strong cross-shore gradients, typically found in wider estuarine regions and homogeneous zones within narrow continental shelf areas. These two types of coastal waters clearly characterize the coastal area of a sea where coastal waters are otherwise strongly influenced by ocean conditions".

*#1.31:* Line 84: rather than basin scale budgets I suggest the authors be specific and either say basin scale PP or basin scale carbon fixation.

**Action:** We have changed it accordingly. Now read in the introduction:

"First, we provide global estimations of PP in coastal waters and we assess their contribution to basin scale PP, their interannual variability and long-term trends"

*#1.32:* Line 104: 'whenever they exceeded about 3-times the mean'. Using "about" in this sentence makes it seem not very precise. Do you really mean to include this here?

**Response to R#1:** We agree. The term "about" has been deleted. Thank you for the advice.

**Action:** We modified the corresponding sentence. Now read in the M&Ms section 2.1.:

"For each parameter, outliers were removed whenever they exceeded 3-times the mean ± SD of the time series"

*#1.33:* Line 131: when you say day length do you mean hours of daylight?

**Response to R#1:** Yes indeed**.**

**Action:** It has been specified in the text. Now read in the M&Ms section:

"D is the day length or hours of daylight (h)".

*#1.34:* Line 170: For clarity I suggest adding coastline before Western, Eastern and Adriatic.

**Action:** We modified the corresponding sentence. Now read in M&Ms section:

"As shown in Table 1, we report PP either as vertically integrated values (PP, in g C m$^{-2}$), spatially integrated estimates for certain basins or regions (ΣPP, in Gt C) or as mean volumetric values (PP$_{VOL,}$ in g C m$^{-3}$)."

*#1.35:* Line 240 Gulf of Sirte – I suggest if places are mentioned, they are included in the map in figure 1.

**Action:** The location of Gulf of Sirte has been added in Figure 1.

[Figure]

Figure 1: Map of the Mediterranean Sea showing the main basins, sea regions, surrounding countries and major rivers. Bathymetric data were obtained from ETOPO1 (Amante and Eakins, 2009). The black contour indicates the 200m isobath, the limit of coastal waters as defined in the present study.

*#1.36:* Line 241: add north before western African.

**Action:** We modified the corresponding sentence. Now read in the results section:

"Along the western North African coast,…"

*#1.37:* Table 2: Suggest using 'Mediterranean' rather than 'Global' .

**Action:** We modified the corresponding sentence. Now read in Table 2 caption:

"**Table 2.** Surface area, chlorophyll mean (Chl) ± standard deviation (SD), $\Sigma PP$, the correspondent % to $\Sigma PP_{Coast}$ (% $\Sigma PP_{Coast}$), $PP_{annual}$ median ± SD ($PP_{annual}$ mean) and $PP_{VOL}$ median ± SD ($PP_{VOL}$ mean) for the Mediterranean Sea, open ocean waters, and coastal waters during the period 2002–2016."

*#1.38:* Line 316: Please rephrase as I don't understand what you are trying to say,

**Action:** It has been rephrased. Now read in the results section:

"A significant negative correlation was observed between coastal $\Sigma PP$ and SST (r=-0.63, p< 0.001; Fig. 6a) revealing a decrease in phytoplankton biomass as the sea warms up."

*#1.39:* Lines 318-323: Are these results shown anywhere: Perhaps they can be included in supplementary material?

**Response to #1.39:** In the manuscript we mentioned the results that were more relevant and significant for the discussion. However, a new supplementary table 1 showing all the results is added to the manuscript.

**Action:** A supplementary table 1 has been added with the correlations between the anomalies of the annual and seasonal $\Sigma PP$ and the corresponding climatic indices NAO and MOI.

Supp. Table 1: Correlations between the yearly total carbon fixation anomalies for the coastal Mediterranean waters and its sub basins (ΣPP) and its corresponding sea surface temperature (SST).

| | | NAO index | | | | | MOI index | | | | |
|---|---|---|---|---|---|---|---|---|---|---|---|
| | | Annual | Spring | Summer | Fall | Winter | Annual | Spring | Summer | Fall | Winter |
| Coastal waters | r | **-0,45** | 0,02 | 0,02 | 0,00 | 0,00 | **-0,22** | **0,28** | 0,02 | 0,02 | 0,02 |
| | *Pvalue* | ***< 0.001*** | *0,60* | *0,58* | *0,96* | *0,97* | ***0,00*** | ***0,04*** | *0,62* | *0,63* | *0,60* |
| Western coast | r | -0,40 | 0,00 | **0,25** | 0,12 | 0,02 | -0,22 | 0,08 | 0,00 | 0,00 | 0,00 |
| | *Pvalue* | *< 0.001* | *0,88* | ***0,06*** | *0,20* | *0,59* | *0,01* | *0,31* | *0,80* | *0,85* | *0,96* |
| Eastern coast | r | -0,42 | 0,02 | **0,22** | 0,02 | 0,00 | -0,11 | 0,02 | 0,02 | 0,01 | 0,04 |
| | *Pvalue* | *< 0.001* | *0,63* | ***0,08*** | *0,59* | *0,89* | *0,15* | *0,62* | *0,65* | *0,71* | *0,46* |
| Adriatic coast | r | -0,31 | 0,03 | 0,08 | 0,00 | 0,00 | -0,38 | **0,37** | 0,01 | 0,00 | 0,01 |
| | *Pvalue* | *< 0.001* | *0,56* | *0,30* | *0,89* | *0,89* | *< 0.001* | ***0,02*** | *0,74* | *0,93* | *0,70* |

*#1.40:* Line 550: MAW – This acronym is not defined in the text so please use full term.

**Response to R#1:** We thank the reviewer for highlighting this oversight.

**Action**: It has been added to the text. Now read in the discussion section:

"Also, localized patterns of relatively high primary production were found in persistent deep water density fronts resulting from the interaction of Modified Atlantic Water (MAW) and Mediterranean water by Lohrenz et al. (1988)."

*#1.41:* Line 610 Bricaud reference – please provide full reference/link that works

**Response to #1.41:** This reference has been deleted in the new ms version.

*#1.42:* Figures: I suggest to avoid using the rainbow colour scheme as it can emphasize unrealistic patterns.

**Response to #1.42:** We thank the reviewer for his/her advice. We took the advice and changed figure 5. In addition, we also used different colour patterns for figures 2, 3 and 8 (shown next) although we would prefer better to maintain the original figures.

**Action**: Figure 5 has been changed.

[Figure]

Figure 5: Trends in primary production and sea surface temperature. Values correspond to the change per decade. a) Theil-Sen trend in pelagic primary production estimated from daily values for the 2002-2016 period. b) Trend in SST temperature. Only significant trends ($p < 0.05$) are shown.

[Figure]

Figure 2: Mean distribution of a) chlorophyll ($Chl_{mean}$, in mg m$^{-3}$), b) annual PP ($PP_{annual}$, in g C m$^{-2}$) and, c) coefficient of variation of PP values ($CV_{PP}$).

[Figure]

Figure 3: Mean distribution of a) volumetric PP ($PP_{VOL}$, in g C m$^{-3}$) and b) coefficient of variation of volumetric PP values ($CV_{PP}$).

[Figure]

Figure 8. a) *ef*-ratio in coastal waters (<200 m) of the Mediterranean Sea and estimated values of b) new ($PP_{new}$) and c) regenerated production ($PP_{reg}$). Mean values for the period 2002-2016.

**Response to Referee #2**

Dear referee, we would like to thank you the interest you have shown in our study and we appreciate the point you made about the importance of studying coastal areas. We thank you for considering our manuscript as suitable for publication in Biogeosciences with *minor changes*. Below are our replies to your comments:

**Response to supplement minor comments from Referee #2:**

*#2.1:* PG 2 Line 41 missing full stop: 2007). The productivity.

**Action**: We corrected the corresponding sentence. Now read in the introduction section:

"In coastal waters, physical and biological processes enhance the carbon transport out of the continental margins into the deep layers of the oceans, thus connecting terrestrial with deep oceanic systems (Cai, 2011; Carlson et al., 2001; Cole et al., 2007). The productivity…."

*#2.2:* PG 5 line 125 I would rephrase "*Note that neither Chl, a\* and \phi are made variable with time.*" With "Note that Chl, a\* and \phi are considered time independent parameters."

We did not modify the sentence as proposed because we do not mean to claim anything here about whether these parameters are or are not variable with time. There are actually good reasons for them to be. No reliable information exists, however, to model this variability and that is why they are maintained constant in the model.

**Action**: The sentence reads in the M&Ms section:

"Note that neither Chl, $a^*$ and $\Phi$ are not made variable with time."

*#2.3:* PG 5 In Equation 4, in order to compute light attenuation is it necessary to consider the normalization on cosines to account for Solar Declination?

**Response to #2.3**: The $\chi$ factor and the exponent "e" were derived from a large dataset of in situ measurements taken over a range of solar zenith angles. As such, the sun zenith angle is not explicitly considered in this formalism, yet it is taken into account on average.

*#2.4:* PG 5 lines 148,149 the empirical formula, Morel (1991) and Morel et al. (1996), are valid also for coastal waters, the modelled primary production corresponds to Gross Primary Production or Net Primary Production?

**Response to #2.4**: The empirical formula by Morel (1991) and Morel et al. (1996) is indeed valid for coastal waters. The modelled PP corresponds to Net Primary Production.

**Action**: We specified in the text that it corresponds to NPP. Now read in M&Ms section:

"Details about how values are assigned to the parameters $a^*$ and $\Phi$, their dependence on temperature, and other features of this Net Primary Production model (NPP), are to be found in Morel (1991) and Morel et al. (1996)."

*#2.5:* PG 6 lines 156,159 The studies by Laws 2000,2011 to derive *ef*-ratio are calibrated on open ocean conditions, could Authors comments on the applicability of such empirical relations in the coastal areas?

**Response to #2.5**: The temperature dependent *ef*-ratio is certainly an empirical approximation and a global algorithm that may present local biases. According to Laws et al., it provides a good approximation to experiments of N-15 labeled uptake, explaining 87% of the variance in the observed ratios. As shown in the original paper (Laws et al., 2000), values from highly different systems were considered in this relationship. As pointed out by the reviewer, most of these regions were oceanic, but also included coastal upwelling areas such as Peru. Even though the algorithm is a coarse generalization of the relationship between total production, export production, and environmental variables (T) that may not be particular to systems like river discharge regions, we believe it yields reasonable results in the Mediterranean and provides a good approximation of export production.

*#2.6:* PG 6 lines 169,170 "*We report annual PP estimates (Gt C) for the entire Mediterranean coastal areas (ΣPPcoast) and separately for the Western, Eastern and Adriatic basins (ΣPPbasin).*" Here Authors mean Western and Eastern coastal basins or open ocean Basins?

**Response to #2.6**: We refer to the coastal basins.

**Action**: It has been now specified in the manuscript. Now read in M&Ms section:

"As shown in Table 1, we report PP either as vertically integrated values (PP, in g C m$^{-2}$), spatially integrated estimates for certain basins or regions (ΣPP, in Gt C) or as mean volumetric values (PP$_{VOL,}$ in g C m$^{-3}$)."

*#2.7:* PG 7 Table 1: The total values of PP for the Mediterranean Sea are obtained combining literature data for the open ocean water summed to the coastal estimates derived in this manuscript? Please explain.

**Response to #2.7:** Our coastal values are added to open ocean estimations to estimate total basin PP.

**Action:** To avoid any confusion, we now specified it in the Table 2 (Table 1 in previous ms version) caption. Now read:

"**Table 2.** Surface area, chlorophyll mean (Chl) ± standard deviation (SD), ΣPP, the correspondent % to ΣPP$_{Coast}$ (% ΣPP$_{Coast}$), PP$_{annual}$ median ± SD (PP$_{annual}$ mean) and PP$_{VOL}$ median ± SD (PP$_{VOL}$ mean) for the Mediterranean Sea, open ocean waters, and coastal waters during the period 2002–2016. For ΣPP and PP$_{annual,}$ Mediterranean Sea values were obtained summing open ocean waters values to coastal waters values"

*#2.8:* PG11: Does Figure 3 show surface PP values or vertically averaged values or they coincide because chlorophyll vertical distribution is constant?

**Response to #2.8**: Figure 3 shows the column-integrated PP values. PP$_{VOL}$ is simply the integrated PP divided by whichever is shallower of the bottom depth or the productive layer.

*#2.9:* PG12 Figure 4 In the caption I would specify "whole Mediterranean coast, b) western coast basin, c) eastern coast basin" otherwise it can be confusing.

**Action**: We decided to specify "whole Mediterranean Sea" in the figure caption. Now it reads:

 "Figure 4: PP variability and trends for coastal waters in a) the whole Mediterranean Sea, b) western basin, c) eastern basin and d) the Adriatic coast. Black solid lines indicate the original monthly $\Sigma PP_{Coastal}$ anomalies and the filtered low frequency signal is overlaid in blue. Green solid lines indicate the filtered low frequency signal for Chl anomalies (mg m$^{-3}$). The red line indicates the PP trend during the analysed period (2002–2016) and the grey band indicates year 2012."

*#2.10:* PG 13 lines 315,316"*A significant negative correlation was observed between coastal ΣPP and SST (r=-0.63, p< 0.001; Fig. 6a) showing that the important decrease of Chl over the years was able to compensate the effect of temperature increase."* Could authors elaborate a bit more the expected correlation between ΣPP, SST and Chl and the corresponding compensation?

**Response to #2.10:** We agree with the reviewer that the sentence was a bit confusing.

**Action**: The sentence has been rephrased. Now read in Results section:

"A significant negative correlation was observed between coastal monthly anomalies of $\Sigma PP$ and SST (r=-0.63, p< 0.001; Fig. 6a) revealing a decrease in phytoplankton biomass as the sea warms up."

*#2.11:* PG 14 Figure 6: It would be nice to see also the chlorophyll trend and how it correlates with SST, NAO and MOI.

**Response to #2.11:** We agree with the reviewer that it would have been interesting to add figures of chl trend with SST, NAO and MOI. Considering that we have already quite a number of figures and tables in the actual ms, we propose to add these correlations to the Supplement Material.

**Action**: We added a figure showing Chl anomalies with NAO, SST and MOI in the supplement material. Now read in suppl. mat.:

[Figure]

Supp Figure 3. Relationship between coastal pelagic chlorophyll (Chl anomalies, green lines) and a) SST anomalies, b) NAO index and c) MOI index (red lines).

**#2.12:** Pg 18 lines 391-392 "*Indeed, Case-1 waters are largely predominant in the coastal Mediterranean regions whereas Case-2 waters are reduced to less than 5% of the whole basin.*" The 5% is related to the coastal basin or to the total Basin? It would be important to report the Case-2 water fraction of the coastal basin to evaluate the relative importance.

**Response to #2.12:** The 5 % refers to the whole basin.

**Action**: In order to clarify this point, we modified the text. Now read in the Results section:

"Indeed, previous studies agreed that Case-1 waters are largely predominant in the coastal Mediterranean regions whereas Case-2 waters are reduced to less than 5% of the whole basin (Antoine and André, 1995; Bosc et al., 2004; Bricaud et al., 2002a). This constitutes some 23% of the coastal waters with prevalence in the north Adriatic Sea, Gulf of Gabes and around Nile delta."

**#2.13:** Pg 20 Lines 439,441 "While *negative tendencies seem to fit with the assumed model of PP limitation associated with increasing temperatures, the origin of the positive trend in the Adriatic basin is more uncertain*". Also chlorophyll exhibits a reduction starting from 2012 and being an independent variable it could be the responsible, or a concurring responsible, for such trend.

**Action**: Now we add this in the manuscript as a plausible explanation (see 4.2. section in the Discussion):

"While negative tendencies seem to fit with the assumed model of PP limitation associated with increasing temperatures, the origin of the positive trend in the Adriatic basin is more uncertain. A

plausible explanation is the variation in the flux and loads of the northern Adriatic rivers. For example, Giani et al. (2012) observed an increase of the Po River flow with increasing phosphate and dissolved nitrogen concentrations in the Po's delta and its surrounding shelf waters. Alternatively, the Bimodal Oscillating System (BiOS), i.e. the feedback mechanism between the Adriatic and Ionian peaking between 2004 and 2006 could have affected mass and nutrient exchanges between the Adriatic and the north Ionian Sea (Font et al., 2007; Schroeder et al., 2008; Šolić et al., 2008; Viličić et al., 2012). In addition, Chl exhibits a strong reduction starting from 2012 that could be the responsible for such trend (Fig. 4-d)."

*#2.14:* Pg 22 line 511. *"Our data does not display a general relationship between shelf width (Q) and PPannual"* from this sentence it seems that Q is the symbol to indicate shelf width instead in figure 9 Q refers to river discharge.

**Action**: This error has been corrected (Discussion section):

"Our data do not display a general relationship between shelf width and $PP_{annual}$ (Table 4 and Fig. 9).".

*#2.15:* Pg 22 Figure 9. The bubble are a bit superimposed and it is not easy to understand what's going on especially near the origin axis. Would it be possible to use a color bar with fixed size bubbles to reduce overlapping, use a log scale for x and y axis, or to arrange the plot to increase readability.

**Action:** Figure 9 has been improved following reviewer's suggestion.

[Figure]

Figure 9: Relation between primary production, shelf width and river discharge flow (*Q*). Bubble colours indicate the $PP_{annual}$ (g C m$^{-2}$) for each of the 18 defined zones (see Fig. 8 and Table 4).

Dear reviewer, we would like to thank you for the interest you have shown in our study and therefore consider it as suitable for publication in Biogeosciences with some modifications. Here are our answers to your comments:

**Response to comments Referee #3:**

*#3.1:* Lines 89-102: Are satellite data (chl, sst etc.) used at daily frequency? In the paper the authors cited monthly or 8 days means but in this paragraph there is not any reference. Could you clarify?

**Response to #3.1:** We recognize the text was misleading regarding the resolution of the different products.

**Action**: We have added further details about the remote sensing data and their processing methodology in section 2.1 Remote sensing data in the Materials and Methods section. Now the M&Ms section reads:

"We used the Mediterranean Sea Level-3 reprocessed surface chlorophyll concentration product (Chl L3) obtained from the EU Copernicus Marine Environment Monitoring Service (CMEMS). This product merges multi satellite observations, and is available at 1-day and 1-km resolution (http://marine.copernicus.eu/OCEANCOLOUR_MED_CHL_L3_REP_OBSERVATIONS_009 _073). Specifically, the dataset used is 'dataset-oc-med-chl-multi-l3-chl_1km_daily-rep-v02' and the variable name used is 'mass_concentration_of_chlorophyll_a_in_sea_water (Chl)' obtainable in a NetCDF-4 file format. This Chl L3 dataset is derived with an updated version of the regional algorithm MedOC4 (Mediterranean Ocean-Colour 4 bands MedOC4, Volpe et al., 2019) for pelagic deep Case-1 waters and the AD4 algorithm (ADriatic 4 band; Berthon and Zibordi, 2004; D'Alimonte and Zibordi, 2003) for Case-2 coastal waters (generally shallow and turbid waters)".

Level-2 Sea Surface Temperature (SST, ℃) at 1-day and 1-km was obtained from every available orbit from Moderate Resolution Imaging Spectroradiometer (MODIS) aboard Terra and Aqua satellites. Data were downloaded from the National Aeronautics and Space Administration (NASA) archive website (http://oceancolor.gsfc.nasa.gov/). Only night-time orbits were selected to avoid problems with skin temperature during daylight. Orbits with quality flags 0 (best), 1 (good) and 2 (questionable) 2 in SST were included after checking their validity and accuracy in order to have a more complete dataset. Daily (24-hour averaged) Photosynthetically Active Radiation (PAR, in E m$^{-2}$) was obtained as a Level 3 product at 9 km and 1-day resolution. This is the best available resolution at NASA archive of MODIS and Medium Resolution Imaging Spectrometer (MERIS) data (https://oceancolor.gsfc.nasa.gov/l3/)."

*#3.2:* Line 106: "*Only values at depths exceeding 5 m depth were considered…*" is there any reference for this assumption? As you know the layer that could influence remote sensing measurements depends on the sea water bio-chemical conditions. Based on my experience I believe that satellite measurements could be influenced by the bottom seagrass also for depths greater than 5 m. Maybe the authors could investigate, in some way, in order to give to the reader an idea of how much final results could be influenced by this issue.

**Response to #3.2:** We considered that 5m depth was a reasonable value to avoid interference by bottom seagrasses reflectance or extreme coastal regions. We recognize that some shallow water

pixels may be affected by seafloor reflectance when the water is clear but, in most cases, this represents a small fraction of the 1x1 km pixel signal.

**#3.3:** Line 130: Is PP estimated on daily satellite images? Graphics and images in the paper show monthly data. Could the authors describe the exact technique used? Did they average input satellite data (i.e. CHL, sst, PAR etc.) and then compute PP? Or did they computed PP on daily satellite data and then averaged PP data?

**Response to #3.3:** We acknowledge that this was not clear. PP calculation was performed every 7 days (starting at day 4), using the averaged Chl for a 7-day window from day-3 to day+3 and for 1 grid point out of 3. Therefore, PP model was run at a 8-day resolution and for 1 pixel out of 3 pixels. Later, PP data was interpolated to 1-day 1-km grid. In some cases -e.g. Fig 6-, monthly PP data and its anomalies were derived from the above mentioned dataset to be able to correlate PP values with the climatic indices NAO and MOI, and SST values.

**Action**: In order to avoid any misunderstanding, we added more details in the text. Now read in M&Ms section (section 2.2):

"Daily PP calculation was performed every 7 days (starting at day 4), using the averaged Chl for a 7-day window from day-3 to day+3 and for 1 grid point out of 3. Therefore, PP model was run with 8-days resolution and for 1 pixel out of 3 pixels. Later, daily PP data was interpolated to 1-day 1-km grid. Monthly PP data and its anomalies were derived from the above-mentioned dataset."

**#3.4:** Line 152-155: this point could also be investigated analyzing the mixing layer depth of the study area. Probably the chlorophyll uniform profile assumption may not be wrong in many cases, but having an idea of where this assumption is wrong could help to better understand the results of the study.

**Response to #3.4:** There is no, to our knowledge, parameterization of the Chl vertical profile from the Chl value at surface that would be valid for coastal waters. Therefore, it is not really feasible to assess whether ignoring possible deep-chlorophyll maxima in coastal areas is significantly affecting our PP estimates. Given the variability in coastal waters, we considered that using a homogenous profile is a better assumption than using global parameterizations of the shape of the vertical profile as a function of the surface Chl that are valid only for deep open ocean waters.

**#3.5:** Line 169-170: Please define exactly how you computed annual PP (PP). Afterwards, in the text, the authors analyze the results for ΣPP, PPannual, PPVOLannual, etc. but I cannot find any definition of these parameters. I think it is crucial to define exactly the quantities used in the analysis.

**Response to #3.5:** We agree with the reviewer that definitions of these parameters are missing in the ms.

**Action**: A new Table 1 has been added clarifying the different definitions of PP that we use.

**Table 1**. Primary production acronyms used in this study, their units and definitions.

| Variable | Units | Definition |
|---|---|---|
| P | $gC\ m^{-3}\ s^{-1}$ | Instantaneous production at each depth $(z)$ of the water column. |
| PP | $gC\ m^{-2}$ | Daily primary production per surface unit. Integration of P over depth and daylength. |
| $PP_{annual}$ | $gC\ m^{-2}$ | Annual mean production per surface unit. |
| $\Sigma PP$ | GtC | Total carbon fixation per year within a basin or specific region. |
| $PP_{VOL}$ | $gC\ m^{-3}$ | Mean volumetric PP. Averaged form the surface to the bottom depth or productive layer |
| PPnew | $gC\ m^{-2}$ | New production (i.e., from allochthonous sources) |
| PPreg | $gC\ m^{-2}$ | Regenerated production. |

*#3.6:* Line 179-180: has this alongshore regionalization been done with SOM (or other technique), or has it been done by the authors observing the results of the SOM regionalization?

**Response to #3.6:** This alongshore regionalization in 18 zones (Z1-Z18) has been performed observing the results of the temporal patterns observed after the SOM regionalization (R1-R9). SOM aggregates the characteristic temporal patterns according to their similarities. In section 2.3 Coastal regionalization we quote, line 18-190 "Then, 18 alongshore marine ecoregions were obtained considering the most relevant cross-shore limits of the SOM-derived regions (Z1 to Z18)."

*#3.7:* Line 217-218: How did the authors deseasonalize the data?

**Response to #3.7:** Time series were de-seasonalized by removing the 8-day means for the original 8-day time series. It has been added to the manuscript section 2.5. Statistical analysis as

**Action**: We now precise it in the main text. Now read in M&Ms section:

"Time series were de-seasonalized by removing the 8-day climatological mean for the original time series".

*#3.8:* Line 226: Is annual carbon fixation per surface area the PP daily average multiplied by 365? If no, how is it estimated?

**Response to #3.8:** The annual value is the sum of all daily values.

*#3.9:* Line 237-239: from tab 1, annual carbon fixation is quite similar between eastern and western subbasins. Since the area of east shelf is about twice of west shelf, it is obvious that the eastern annual integrated PP is approximately double of the west sub-basin. On the other hand, it is not absolutely obvious why the "productivity per unit volume" of eastern compart is more than double of the western one (and even higher than the Adriatic Sea). I'm a little bit surprised…

**Response to #3.9:** We understand this might be somewhat counterintuitive. Although it is surprising to observe higher coastal PP for the eastern than for the western basin, our results are consistent for both integrated and volumetric values. Higher coastal integrated PP in the eastern

basin can be explained by its great coastal area compared to the western basin (twice larger). For the volumetric PP values, considering the lack of large shallow and productive areas in the western Mediterranean, there were few mean annual PP values above 30 gC m$^{-3}$ in this basin whereas high volumetric PP were more frequent in the Adriatic Sea and in the eastern Mediterranean. Pixels with PP values >30 gC m$^{-3}$ were located in shallow waters of the Gulf of Gabes and in the Nile Delta, and less so, in the northern Adriatic. Coastal values are highly dependent on local enrichment processes. Apart from the influence of rivers (mainly the Rhone in the western side, the Po in the Adriatic and the Nile in the eastern side), major influence in shelf production generally comes from other sources such as the inputs of the Black Sea in the northern Aegean, and from local processes in the Gulf of Gabes. This would highly increase the productivity in the eastern coastal areas. While some overestimation of PP may occur in these waters due to the distinct optical conditions of these waters (Bosc et al., 2004), the Gulf of Gabes is considered one of the most productive coastal areas of the Mediterranean (e.g. D'Ortenzio and Alcalà, 2009). Its shallowness (< 50 m <at 110 km off the coast), unique tidal range (maximum >2 m) and the lack of summer nutrient exhaustion undoubtedly contribute to the high productivity found in the coastal areas of the eastern basin (Béjaoui et al., 2019).

Averages distributions are not normal. In some shallow and highly productive regions this is particularly notable. In order to better understand the distribution to integrated PP and volumetric PP for each pixel, we show here frequency histograms for both of them (Fig. 1). In the case of PP (gC m$^{-2}$; see Fig. 1a) this is compensated by the integration depth and, thus, the weight of these pixel, although relevant, is less critical than in PP (gC m$^{-3}$, Fig 1b) where the influence of vertically averaging just over few surface values exacerbates the differences with overall values. As shown in Fig. 1b, due to the lack of large shallow and productive areas, there are few values above 30 (gC m$^3$) in the Western Med, whereas high PP is more frequent in the Adriatic (red) and in the Eastern Med (blue). If the pixels with values >30 gC m$^3$ are plotted (Fig 2.) it becomes evident that most of them are located in shallow waters of the Gulf of Gabes and in the Nile delta, and less so, in the northern Adriatic. To avoid these problems, we refer now to median values in Table 2 (Table 1 in the previous version of ms), yet mean values are still provided as a reference.

[Figure]

Fig. 1 Frequency histograms for a) Integrated and b) vertically averaged PP estimations. Blue (East Med.) green (West Med), red (Adriatic). Y-axis in fig 1b is logarithmic.

[Figure]

Figure 2. Map showing the location of the pixels with values >30 gC m$^{-3}$ in blue.

**Action**: We modified Table 2 (Table 1 in the previous version of the ms). Now read in Results section:

**Table 2.** Surface area, chlorophyll mean (Chl) ± standard deviation (SD), ΣPP, the correspondent % to ΣPP$_{Coast}$ (% ΣPP$_{Coast}$), PP$_{annual}$ median ± SD (PP$_{annual}$ mean) and PP$_{VOL}$ median ± SD (PP$_{VOL}$ mean) for the Mediterranean Sea, open ocean waters, and coastal waters during the period 2002–2016. For ΣPP and PP$_{annual}$, Mediterranean Sea values were obtained summing open ocean waters values to coastal waters values.

| | Surface area | | Chl | ΣPP | % of | PP$_{annual}$ | PP$_{VOL}$ |
|---|---|---|---|---|---|---|---|
| | ($10^3$ km$^2$) | (%) | (mg m$^{-3}$) | (Gt C) | ΣPP$_{Coast}$ | (g C m$^{-2}$) | (g C m$^{-3}$) |
| Mediterranean Sea | 2,504 | | 0.19±0.78* | 0.349±0.118*** | | 140±40** | |
| Open ocean waters | 1,975 | | 0.11±0.18* | 0.308±0.118 | | 136±40** | |
| Coastal waters | 529 | 100 | 0.30±0.17 | 0.041±0.004 | 100 | 83±75 (100) | 1.16±9.60 (2.93) |
| Western coast | 141 | 27 | 0.21±0.14 | 0.011±0.001 | 25 | 90±39 (98) | 1.23±2.61 (1.59) |
| Eastern coast | 287 | 54 | 0.30±0.16 | 0.021±0.002 | 51 | 73±86 (93) | 1.01±11.8 (3.34) |
| Adriatic coast | 101 | 19 | 0.39±0.23 | 0.010±0.001 | 24 | 99±76 (124) | 1.50±7.23 (3.27) |

* Mean surface Chl values obtained by averaging the 8-days and 4-km resolution of surface satellite Chl values obtained from CMEMS (Salgado-Hernanz et al., 2019).

** PP estimated by averaging published satellite data shown in Table 3.

*** ΣPP estimated adding coastal waters data from this study to open ocean waters data obtained from Table 3.

**#3.10:** Line 239-241: I suggest starting the sentence by citing the figure you are referring to instead of citing it at the end.

**Action**: This change has been performed. Now read in Results section:

"Figure 2 reveals the differences PP$_{annual}$ between the more productive shelf waters in the western basin and those in the eastern basin (90±39 g C m$^{-2}$ and 73±86 g C m$^{-2}$, p<0.001)."

***#3.11:*** Line 243. "…*the coefficient of variation of primary production (CVPP)…*" how did you calculate this coefficient? Defining this "coefficient of variation of primary production" would also clarify why there are 2 different "coefficient of variation of primary production", one in figure 2c and another in figure 3b.

**Response to #3.11:** The Coefficient of Variation is a dispersion measurement defined as Standard Deviation / Mean.

**Action**: We specified in the text how CV was estimated. Now read in M&Ms section:

"The coefficient of variation (CV) has been estimated for PP as the ratio of the standard deviation to the mean."

***#3.12:*** Tab 1: Could you please insert the exact reference (product ID as for satellite daily data described above) for the 8-days e 4km resolution data taken from CMEMS? I cannot find them. Why is *** only for "Mediterranean Sea"? shouldn't it also be on "Open ocean waters"?

**Response to #3.12:** We used ** for data that we took from Table 2. We used *** for data that we estimated using our coastal results added to the open waters results that we obtained in the literature from Table 3.

**Action**: We added now in the manuscript the specific product ID and variable used for the analysis and we modified Table 2 (Table 1 in the previous version of ms).

Now read in M&Ms section:

"We used the Mediterranean Sea Level-3 reprocessed surface chlorophyll concentration product (Chl L3) from multi satellite observations, obtained from the EU Copernicus Marine Environment Monitoring Service (CMEMS) available at 1-day and 1-km resolution (http://marine.copernicus.eu/OCEANCOLOUR_MED_CHL_L3_REP_OBSERVATIONS_009 _073). The specific dataset used is 'dataset-oc-med-chl-multi-l3-chl_1km_daily-rep-v02' and the variable name used is 'mass_concentration_of_chlorophyll_a_in_sea_water (Chl)', obtainable in a NetCDF-4 file format"

Now read in Table 2:

**Table 2.** Surface area, chlorophyll mean (Chl) ± standard deviation (SD), ΣPP, the correspondent % to ΣPP$_{Coast}$ (% ΣPP$_{Coast}$), PP$_{annual}$ median ± SD (PP$_{annual}$ mean) and PP$_{VOL}$ median ± SD (PP$_{VOL}$ mean) for the Mediterranean Sea, open ocean waters, and coastal waters during the period 2002–2016. For ΣPP and PP$_{annual}$, Mediterranean Sea values were obtained summing open ocean waters values to coastal waters values.

| | Surface area | | Chl | ΣPP | % of | PP$_{annual}$ | PP$_{VOL}$ |
|---|---|---|---|---|---|---|---|
| | ($10^3$ km$^2$) | (%) | (mg m$^{-3}$) | (Gt C) | ΣPP$_{Coast}$ | (g C m$^{-2}$) | (g C m$^{-3}$) |
| Mediterranean Sea | 2,504 | | 0.19±0.78* | 0.349±0.118*** | | 140±40** | |

| | | | | | | | |
|---|---|---|---|---|---|---|---|
| Open ocean waters | 1,975 | | 0.11±0.18* | 0.308±0.118 | | 136±40** | |
| Coastal waters | 529 | 100 | 0.30±0.17 | 0.041±0.004 | 100 | 83±75 (100) | 1.16±9.60 (2.93) |
| Western coast | 141 | 27 | 0.21±0.14 | 0.011±0.001 | 25 | 90±39 (98) | 1.23±2.61 (1.59) |
| Eastern coast | 287 | 54 | 0.30±0.16 | 0.021±0.002 | 51 | 73±86 (93) | 1.01±11.8 (3.34) |
| Adriatic coast | 101 | 19 | 0.39±0.23 | 0.010±0.001 | 24 | 99±76 (124) | 1.50±7.23 (3.27) |

* Mean surface Chl values obtained by averaging the 8-days and 4-km resolution of surface satellite Chl values obtained from CMEMS (Salgado-Hernanz et al., 2019).

** PP estimated by averaging published satellite data shown in Table 3.

*** ΣPP estimated adding coastal waters data from this study to open ocean waters data obtained from Table 3.

*#3.13:* Line 290-291: How did you estimate this interannual variability?

**Response to #3.13:** This value of interannual variability was calculated using the formula (PP max – PP min/ PP average) * 100. A PPtot value was calculated for every year. Then, the minimum value is subtracted from the maximum value and divided by the average.

**Action**: Section 2.3 now reads:

"ΣPP$_{Coastal}$ exhibits moderate interannual variability (up to 25%) whereas basin scale interannual variations range from 26% in the Adriatic basin, up to 28% in the western basin and 29% in the eastern basin. This value of interannual variability was calculated subtracting the year with the minimum annual PP to the year with the maximum annual PP and then dividing this value by the mean annual PP".

*#3.14:* Line 298-299: How did you calculate the "the filtered low frequency signal" for PP and CHL?

**Response to R#3.14:** First, we calculated the anomalies of the total monthly PP (ΣPP) for each of the 180 months between January 2002 and December 2015. Then, we used the *smooth* function of Matlab applying the *sgolay* filter with a span degree of 17 to the anomalies of the total PP. This sgolay filter uses the Savitzky-Golay method with the polynomial degree specified by degree. In the case of Figure 4, we used monthly data (in total 180 time steps) for each region (a-whole Mediterranean coast, b-western coast, c-eastern coast, d- Adriatic coast) to visualize PP variability and trends. To filter the low frequency signal, we used a span degree of 17. This degree was therefore filtering about 8 months before and after every time step (about 1.5 years). Some proofs were doing using a degree of 29 (2.5 years low signal) or 59 (5 years low signal).

**Action**: Now read in the text (section 3.2):

"As shown in Fig. 4, variability in annual PP is dominated by short-scale variations (i.e. subdecadal). The interannual variability is indicated by the low frequency signal of the of the monthly mean anomalies. The filtered low frequency signal to the anomalies of ΣPP and Chl has been calculated using the Matlab smooth function and applying the sgolay filter, which uses the Savitzky-Golay method, with a polynomial span degree of 17. This degree was therefore filtering about 8 months before and after every time step (about 1.5 years), showing then interannual variability".

***#3.15:*** Line 306-307: "Most of these regions presented declining PP trends…". This sentence does not seem so evident observing fig 5a. The only evident negative trend is in the Gulf of Gabes as underlined by the authors. Moreover, I believe that fig 5a and graphs in fig4 are quite inconsistent. In fig 4 trends are negative for Mediterranean Sea, west and east sub-basin, while for Adriatic Sea there is no evident trend. From fig 5a I'd say that on average Mediterranean Sea trend is quite positive (red areas are greater than blue ones). For west and east sub-basin the negative trend shown in fig 4a is not so evident, especially for the eastern compart. About Adriatic Sea fig 5a shows a clear positive trend. Could the authors explain this apparently discrepancy and how a reader should interpret it?

**Response to #3.15:** We recognize that, despite being different estimations, Figure 4 and Figure 5 could be misleading. Figure 4 showed a regional trend resulting from 15 points (one mean value per year). Moreover, from 2012 a reduction in PP is shown in every region but the Adriatic. The Adriatic region presented positive PP values for years 2013 and 2014 (see Supplementary figure 1) and this could change the PP trend when only 15 points are considered (i.e Figure 4). Figure 5 showed trend for each pixel based on daily data.

**Action**: in order to avoid any misunderstanding, we now only provide trends obtained with the complete time series (as shown in fig 5).

[Figure]

Figure 5: Trends in primary production and sea surface temperature. Values correspond to the change per decade.  a) Theil-Sen trend in pelagic primary production estimated from daily values for the 2002-2016 period. b) Trend in SST temperature. Only significant trends (p < 0.05) are shown.

***#3.16:*** Fig 6: line 324: it is not specified (here or in the text) what blue lines meaning. Are they annual PP anomalies?

**Response to #3.16:** This figure shows the relationship between coastal pelagic PP and SST, NAO and MOI. Blue lines indicate the global PP monthly anomalies for the whole Mediterranean Sea, whereas red lines indicate the SST monthly anomalies, NAO or MOI indices respectively. Blue lines are the same in Fig 6a-c.

**Action**: We modified Figure 6a caption. Now read in Figure 6 caption:

 "*Figure 6: Relationship between coastal pelagic primary production (ΣPP anomalies, blue lines) and a) SST anomalies, b) NAO index and c) MOI index (red lines*")

*#3.17:* Line 327-336: R1 to R9 are represented in fig 7 as PP in gC m$^{-2}$ d$^{-1}$, but in tab 3 there are mean annual PP values. Again, it should be defined how you estimate mean annual PP starting from daily PP.

**Response to #3.17:** The SOM analysis was performed using the original PP time series in gC m$^{-2}$ d$^{-1}$. The patterns that we obtained are shown in Fig. 7. In order to provide mean values for the different regions, we give information of the mean annual values in gC m$^{-2}$ y$^{-1}$ as data in the manuscript is provided in year unit.

We estimate mean annual PP values from daily values as follows: Firstly, mean value were calculated using daily original data. Secondly, their mean value was obtained. Thirdly, data was converted to annual units by multiplying it by 365.

**Action:** We now defined in the text how we estimate mean annual PP. Now read in title Table 3:

"Mean annual PP is estimating by averaging mean daily PP and then multiply it by the number of days of the year; i. e., 365."

*#3.18:* Line 392: "…whereas Case-2 waters are reduced to less than 5% of the whole basin.". Is there any reference for this statement?

**Response to #3.18:** In the manuscript we provided some references "*Indeed, previous studies agreed that Indeed, Case-1 waters are largely predominant in the coastal Mediterranean regions whereas Case-2 waters are reduced to less than 5% of the whole basin (Antoine and André, 1995; Bosc et al., 2004; Bricaud et al., 2002). In particular, they are confined to the north Adriatic Sea, Gulf of Gabes and around Nile delta where our PP estimations may present larger uncertainties (Antoine and André, 1995).*".

We obtained that value of about 5% from the literature. Bricaud et al. (2002) explained in section 2.1. Computation of PP: *"On these maps, Case 2 waters were identified by discarding the pixels where R(555) was > 0.025 (see Bricaud & Morel, 1987), and an ''average mask'' was selected. The corresponding area was 5% of the total area of the Basin, which is close to the estimate (4%) provided by Antoine et al. (1995). Although it is acknowledged that the extension of Case 2 waters may vary throughout the year, the use of such a constant mask (instead of a temporally variable mask) allows the spatial means of chlorophyll concentration (or primary production) to be computed over a fixed area, and therefore to be comparable from month to month. The same average mask was used for".*

Bosc et al. (2004) also use this value: *"The Mediterranean is particularly well adapted to ocean color studies as the conditions of observation are optimal (low cloudiness), and also because turbid-case 2 waters, where the interpretation of ocean color is complex, are here of marginal areal extent, covering less than 5% of the whole basin [Antoine et al., 1995]."*

Antoine and André (1995) specify *"Turbid case 2 waters are particularly extensive in the Adriatic Sea, covering about 25% of the basin area. Moreover"*. Also: *"This procedure also allows turbid case 2 waters to be identified and thereafter discarded (black mask in images). They cover about 3.5% of the entire eastern Mediterranean, mainly located in the north Adriatic Sea, in the Gulf of Gabes, and around the Nile delta."*. In this article, they published a Table where they provide information about the area excluding case 2 waters for the subprovinces of the Adriatic, Aegean, Ionian, North Levantine, South Levantine and Total Mediterranean Sea.

**Table 2.** Sea Surface Covered by Each of the Five Provinces Defined in the Present Work

| | Subprovince | | | | | |
| --- | --- | --- | --- | --- | --- | --- |
| | Adriatic | Aegean | Ionian | North Levantine | South Levantine | Total |
| Area, $10^{12}$ m$^2$ | 0.134 | 0.193 | 0.773 | 0.244 | 0.336 | 1.68 |
| Percentage of whole eastern Mediterranean | 8 | 11.5 | 46 | 14.5 | 20 | 100* |
| Area excluding case 2 waters, $10^{12}$ m$^2$ | 0.100 | 0.192 | 0.760 | 0.241 | 0.330 | 1.62 |

*The eastern basin contributes to 67.4% of the whole Mediterranean area.

*#3.19:* Fig 8: Caption of the figure refers to another type of figure (seasonal PP). Fig 8a has a different color palette (and range) with respect to other 2 (b and c) images.

**Response to #3.19:** We apologize for the mistake.

**Action**: The figure caption has been corrected. Now read in Figure 8 caption:

"*Figure 8. a) ef-ratio in coastal waters (<200 m) of the Mediterranean Sea and estimated values of b) new (PPnew) and c) regenerated production (PPreg). Mean values for the period 2002-2016.*"

**References**

[revised manuscript text omitted]

---

## Author Response (AR2)

**Manuscript Reference No**.: bg-2020-457
**Biogeosciences Discuss**., https://doi.org/10.5194/bg-2020-457-RC1, 2021
**Title:** "Pelagic primary production in the coastal Mediterranean Sea: variability, trends and contribution to basin scale budgets" by Paula Maria Salgado-Hernanz et al. 2021.

==============================

**Response to Associate Editor Decision**

We deeply thank the editor and the referees for their comments and the final acceptance of our manuscript for publication in Biogeosciences.

**Response to minor comments from Referee #1:**

*#1.1:* Table 4: Where did the river discharge data come from? I suggest you provide a reference for it.
**Action**: We thank the referee for notifying this lack. River discharge data (Q) was taken from the book "*Tockner, K., Uehlinger, U. and Robinson, C.: Rivers of Europe, 1st ed., Academic Press, Elsevier., 2008*". This reference has been now provided in the manuscript.

In addition, header of Table 4 now reads: "**Table 4.** *Surface, river discharge flow (Q), annual mean PP (PP_annual), annual integrated PP (ΣPP) and its contribution respect to the total coastal Mediterranean Sea PP for each of the 18 alongshore zones characterized in the Mediterranean Sea. Mean and standard deviation (S.D.) are calculated from 14 year averages is calculated from 15-year averages (2002-2016). River discharge flow was extracted from Tockner et al. 2008*"

*#1.2:* Line 467-468: "In addition, Chl exhibits a strong reduction starting from 2012 that could be the responsible for such trend". Please clarify this sentence as I don't follow what you are trying to say.
**Action**: We thank the referee for the demand of this clarification. We agree with the referee that that sentence did not provide new information. The sentence then has been removed since PP trends in the Adriatic Sea are previously discussed (lines 460-466) and that sentence mislead the information.

*#1.3:* Figures: I'm still concerned by the use of the rainbow colour scale but will not insist that the author change the figures that use it. However, I do wonder why the authors prefer the rainbow colour scale? If it is because it shows trends clearer, then I urge the authors to consider that the rainbow colour scale can emphasize patterns that are not actually significant and disappear in a linear colour scale such as the scale used in the figures presented in the response to authors.
**Action**: Figures 2 and 3 has been changed and the *rainbow* colour scale has been rejected. Now for the figures, we used instead *parula* colour scale.

*#1.4:* Line 396-397: I suggest change "might have alter very weakly" to "might very weakly alter".
**Action**: Thank you for the correction. The sentence has been modified.

*#1.5:* Line 431: "from" rather than "rom"

**Action**: Thank you for the typo. It has been now corrected.

*#1.6:* Line 446-447: suggest changing "so that temporal trends derived from their analysis are highly depending on decadal variability" to "so that temporal trends derived from their analysis highly depend on decadal variability"

**Action**: Thank you for the correction. The sentence has been refreshed.